# A Decrease in Psycho-Emotional Health in Middle-Aged Russian Women Associated with Their Lifestyle

**DOI:** 10.3390/ijerph18020388

**Published:** 2021-01-06

**Authors:** Maria V. Saporovskaia, Tatiana L. Kryukova, Maria E. Voronina, Elena V. Tikhomirova, Anna G. Samokhvalova, Svetlana A. Khazova

**Affiliations:** 1Department of General and Social Psychology, Kostroma State University, 156005 Kostroma, Russia; saporov35@mail.ru (M.V.S.); mariaantonvoron@mail.ru (M.E.V.); tichomirowa82@mail.ru (E.V.T.); 2Department of Education and Acmeology, Kostroma State University, 156005 Kostroma, Russia; a_samohvalova@ksu.edu.ru; 3Department of Special Education and Psychology, Kostroma State University, 156005 Kostroma, Russia; hazova_svetlana@mail.ru

**Keywords:** women, middle-aged, psycho-emotional health, well-being, relationships with parents, attachment, separation, guilt

## Abstract

(1) Background: The present study is aimed to determine the predicting role of objective (lifestyle) and subjective factors of middle-aged women’s psycho-emotional health such as their relations with parents, attachment and separation types. Women who are overloaded with professional and family roles have high stress level, their indicators of psychological well-being and emotional level decrease when they have to give everyday care to their elderly parents. (2) Methods: The research sample has two empirical groups. Sample of Study 1 includes middle-aged women (*n* = 61) aged 38–56 (M = 48.1, SD = 3.5); sample of Study 2 includes middle-aged women (*n* = 85) aged 33–52 (M = 40.6, SD = 3.1): married (70.5%) and divorced (29.5%), having children of 14–28 years old; giving everyday care to elderly parents for more than 1.5 yrs. Some live separately (62.3 %), or have to cohabitate with parents (37.7%). All women evaluate their life situation as difficult and manifest signs of high psycho-emotional stress. We used methods adapted for the Russian-speaking sample: getting socio-demographic information, an interview; The scales of psychological well-being; Attachment style and Interpersonal Guilt Questionnaires (study 1); Psychological Separation Inventory, Purpose-in-Life Test, projective methods (study 2), mathematical statistics. (3) Results: A number of factors and indicators of women’s psycho-emotional health decrease in the situation of role overload have been identified. Among the factors there are four main types of women’s relationships with parents: *Anxious closeness; Ambivalence of feelings; Secondary relationship with parents; Alienation,* predicting of psycho-emotional health that are reducing or enhancing their personal resources. Besides, a type of separation of an adult woman from her mother predicts her level of well-being. (4) Conclusions: The study confirms that middle-aged Russian women’s psycho-emotional health depends on contextual factors (difficult role-overloaded lifestyle) and factors integrating women‘s relations towards parents, attachment, guilt and separation. Types of middle-aged women’s relationships with parents contribute to their psychoemotional health in a different way.

## 1. Introduction

The change in the usual routine and in economic relations, stratification in society, and gender inequality, as well as political, economic, sociocultural, and ecological threats, result in psychological overloads, changes in the lifestyle, and affect the mentality of contemporary females, who remain a vulnerable social group even in the 21st century. Antonio Guterres, the UN Secretary-General, emphasized that the coronavirus-related crisis made the life situation for many women worse in a variety of countries, and asked world governments to make protection from gender-based violence the priority in their plans to cope with the consequences of the COVID-19 pandemic [1].

Women’s vulnerability in Russia is obvious. Women’s routine is the history of coping with difficulties on their own, survival skills, coping with the situation, and, at the same time, of self-development. A typical Russian woman is oriented at a full-time professional job and a promotion, together with her family duties and child-rearing [2]. Despite the fact that women are involved in providing for their families financially, according to the latest sociological research, men limit their own social male roles to those of a breadwinner and moneymaker. The role related to childcare and child-rearing remains traditionally female [3,4]. At the same time, financially, women are in the inferior position in comparison with men. The average income of females in January 2020 was only 67.9% of the average male income. The industrial segregation will maintain the inequality in the future [2].

Women nowadays, including Russia women, have a high level of education and professional and career interests. Moreover, child-rearing and childcare are also female duties, which make it hard for women to combine family and professional duties [4,5]. Describing a professionally successful woman, researchers use such terms as double load, role conflict, double career, etc., [6,7]. Research in gender psychology shows that gender conflicts, related to the need to combine professional and family roles, lead to the increase in stress levels in women, whereas the indicators of psychological well-being decrease together with the emotional tone [5,8]. The increase in stress levels leads to the low satisfaction with marriage and to difficulties in self-actualization. In turn, it leads to the deterioration of psychological health that, according to the WHO, is determined by mental and psychological well-being [9]. Evidence shows that the psychological and physical condition of females in the situation of a chronic family stress leads to subjectively felt helplessness and pointlessness of effort [10,11]. The accumulating exhaustion stemming from low results and hard efforts and the inability to satisfy own needs can gradually result in the female’s depletion of adaptation resources and may lead to the deteriorated psycho-emotional health and well-being [12,13].

However, not only the need to combine professional, spousal, and parental roles can lead to lack of well-being. The period between 35 and 55 years of age, i.e., the middle age of a woman, is the time of systemic changes in family relations (mainly, the attitude to growing children and ageing parents) [14]. These changes often cause stress and take a high toll on the woman’s resources [15,16]. The roles related to care for elderly parents are still highly feminized in Russia. As a rule, this challenge is met by middle-age women. This life situation is usually described as the vertical change –adult children become “parents” toward their own elderly parents. Psychologically, the situation is hard for both ex-children and ex-parents. As the system of care for the elderly is poorly developed in Russia and the low level of income prevents families from hiring nurses, women are often left to face the problem on their own.

Besides this, Russian middle-aged women are typical representatives of the collectivistic culture and holders of its core values. They feel the necessity to be involved in the life of their nearest people, to help in a difficult situation, to manifest attachment and care. Traditional collectivistic values include sticking to traditions, obedience, and duty. These values keep the family as a cohesive entity with stable interconnections between its members [17]. Therefore, the involvement into caregiving for elderly parents is not only an objective necessity, but also an important need of a person who adheres to collectivistic values. Upon reaching the middle age, women start to experience new forms of closeness to elderly parents, due to the deterioration of the parents’ physical and mental health. They often feel the highest possible level of care and closeness in their relationships with parents. At the same time, they experience intrusion, as parents are often possessive and jealous of daughters [18]. Women feel the need to draw interpersonal borders in their relationships with elderly parents. To do it successfully, it is important to understand the specifics of the attachment of an adult daughter to her parents and completeness or incompleteness of psychological separation. This is especially vital for the positive (or negative) interpretation and re-interpretation of the relationships with parents. At this age, women are in a transition between the life’s scenarios “inherited” from parents to the independent life planning. However, the subjective factor (the style of attachment, specifics of separation, and guilt toward parents) has a significant influence on the specifics and effectiveness of the process [18,19].

We define psycho-emotional health as the basis of personal functionality and stability against the influence of external factors. Among its indicators are psychological and emotional well-being, the emotional state, and the meaning-value regulation (namely, life goals, meaningfulness, and control of life) [20,21].

It should be noted that the attachment style, the type of separation, and the feeling of guilt are components of any child’s/daughter’s relationship to her parents. The combination of their specific characteristics determines the main types of relations, as generalized models [22]. Since the type of relationship between a daughter and her parents is a link between her inner world and a social environment, the identification and description of the types of relationships are important for understanding the role of the subjective factor in the psycho-emotional health of women included in daily caregiving.

Recent years have seen a number of research studies on psychological problems of caregivers for their elderly relatives [23,24]. There is evidence that women who care for their aged parents with health problems risk their own health: Caregivers to Alzheimer patients lose from four to eight years of life, and caregivers for cancer patients have an impaired immune system [25].

Therefore, middle-aged women face the need to simultaneously solve three major life problems (triple load): professional career; care for their own family and the change in relations with growing children; and daily care for elderly relatives. The unity and complexity of these tasks are the reflection of the specific female lifestyle that inevitably influences their psycho-emotional health and well-being [26,27]. However, we realize that the objectively difficult life situation can be aggravated with a number of subjective psychological factors, closely related to the specifics of family relations.

We see psycho-emotional health as the basis of personal functionality and stability against the influence of external factors. Among its indicators are psychological and emotional well-being, the emotional state, and the meaning-value regulation (namely, life goals, meaningfulness, and control of life) [9,19].

Thus, initially, we assumed that the psycho-emotional health of middle-aged women with triple role load was conditioned by both objective (life situation) and subjective (relationships) factors.

Research questions are as follows: What are the types of relations with their parents among middle-aged women included in the daily care of their parents? What is the role of the subjective factor—their attachment style, the feeling of guilt toward parents, and the separation from parents’ type in their psycho-emotional health and well-being?

## 2. Materials and Methods

### 2.1. Research Design

The research was conducted in small towns in the Kostroma region, which is ranked 62nd out of 85 in the Russian rating of quality of life in its regions. In the past year, the region has lost two positions in the rating, indicating the deterioration of life quality. All the respondents in our research work in educational, medical, and commercial institutions. Their income does not exceed the average regional level; that is why they look after their parents themselves, as they cannot afford a nurse [2,28].

The research sample includes two empirical groups. The sample of Study 1 includes middle-aged women (*n* = 61) aged 38–56 (M = 48.1, SD = 3.5). In total, 67.5% are married, 29% divorced, and 3.5% are widowed. All of them have children aged 19–28; in 38% of cases, children live separately, but depend on their parents financially (fully or partly). In 50.2% of cases, children live together with the parents. In 11.8% cases, children live separately with their own families, but involve their mothers into care for grandchildren. The sample of Study 2 includes middle-aged women (*n* = 85) aged 33–52 (M = 40.6, SD = 3.1). In total, 70.5% of those are married, and 29.5% are divorced. All have children aged 14–21. In 41% of cases, children live separately from their parents, but depend on them financially (fully or partly). A total of 59% of children live together with their parents.

All the respondents have been looking after one or two elderly parents on a daily basis for 1.5 years and longer. Those respondents who live separately from their parents (62.3%) state that they live “in two homes”; 37.7% of the respondents live with their parents. The preliminary interview showed that 100% of the respondents see their life situation as difficult and experience signs of high psychological and emotional tension.

### 2.2. Data Collection and Measurement

The empirical data were collected from December 2017 to June 2020. The study included two stages: Study 1 was carried out from December 2017 to June 2018, with the sample of 61 women; Study 2 was conducted from August 2019 to June 2020, with 85 women.

At the first stage, both studies involved the questionnaire, to collect social and demographic data (age, gender, and level of education; marital status and the age of children; professional status, experience, and length of care for elderly parents). Then, in a face-to-face interview, each respondent was asked to describe her real-life situation and subjectively evaluate everyday stress overload (high, medium, and low levels of stress).

The content analysis of these replies enabled the authors to identify a number of empirical references (indicators) of deteriorating psycho-emotional health and well-being of women in a situation of role overload, namely low indicators of psychological well-being, dependence on the opinion of others, and on the evaluation of others while taking important decisions, the lack of meaningfulness in life, the lack of particular aims in life, lower control over the situation around (Study 1). The replies are characterized by the respondents’ anxiety about their future and neurotic symptoms, such as high demands set to oneself, easy excitement, irritation, and low emotional tone (Study 2).

The analysis of the contemporary understanding of the problem enabled the authors to consider additional factors of deteriorating psycho-emotional health, namely the attitude to parents and the type of attachment, including irrational guilt (Study 1) and the style of separation from parents (Study 2).

The research was conducted with the following tools (Table A2):The questionnaire to collect social and demographic information.Study 1The semi-structured interview on the topic “The image of parents and my life situation”.The scales of psychological well-being, Ryff, 1995 (84 items; 6 points Likert scale (totally disagree–totally agree); 7 main and 3 subscales; Cronbach alpha 0.71–0.91, Russian language sample *n* = 510 [29,30].Semantic differential, SD, Osgood, 1952/1964, modified for the study of family—conditioned states [31].Attachment Style Questionnaire (ASQ), Feeney, Noller, 1994 (40 items; 6 point Likert scale (totally disagree–totally agree); 5 subscales; Cronbach alpha 0.73–0.80 for 2 samples [32].The Interpersonal Guilt Questionnaire-67 (IGQ-67), O’Connor et al., 1987 (67 items, 5 point Likert scale, “very untrue or strongly disagree” versus “very true or strongly agree”, with some items reverse scored, 4 subscales. In the sample of 1979 participants, Cronbach alpha of the full-scale score is 0.91, and 0.78, 0.78, 0.75, and 0.86 for each of the 4 subscales [33].Study 2Psychological Separation Inventory (PSI), Hoffman, 1984 [34]; Dzukaeva, 2014 (124 items, 4 subscales, Cronbach alpha is 0.84–0.92, in a Russian language sample, *n* = 196; 5-point Likert scale: 1—completely not about me; 5—completely about me [35];Purpose-in-Life Test (PIL), Crumbaugh, Maholic, 1969/1981 [21]; Leontiev, 2013 (20 pairs of opposite items; 5 subscales and general life-meaningfulness indicator, Cronbach alpha is 0.74–0.87 for a Russian language sample [36].The projective method of incomplete sentences, based on the principles of projective research and content analysis (Holaday, Smith, and Sherry, 2000; McAdams and Zeldow, 1993) [37,38].The projective method of metaphors’ analysis “My lifeline”, Solomin, 2002 [39].

All the methods were adapted for the Russian-speaking sample.

### 2.3. Statistical Analysis

The collected data were processed with the software pack SPSS Statistics 22.0.

The preliminary stage involved the procedure of assessing the normality of criteria distribution through calculations in descriptive statistics by using the Kolmogorov-–Smirnov consent criteria (Kolmogorov–Smirnov Test). To distinguish mid-aged women’s attitude toward their parents an exploratory factor analysis was used. The prediction of adult daughters’ psycho-emotional health was set up via simple linear regression. The differences between the groups were assessed with Mann–Whitney U test.

### 2.4. Ethical Considerations

Pursuant to Federal Act On personal data No. 152 of 27 July 2006, all subjects gave their informed consent to take part in the research before they participated in it. The respondents were informed of the research aim and gave permission to use the obtained data (Appendix A
Table A1). The rights of persons participating in the research were protected. The research was conducted in accordance with the 1975 Helsinki Declaration, revised in 2013, under the supervision of the regional department of the Russian Psychological Society. The protocol was approved by the Ethics Committee of Yaroslavl State Medical University, Russia (No. 41, 10.22.2020).

## 3. Results

### 3.1. Study 1

This study is based on problem questions—what are the types of relations with parents’ characteristic of middle-aged women who are included in the daily care of elderly parents? And how are these types related/conjugated to their psycho-emotional health?

Relationship types with parents (closeness, independence of parents/self-sufficiency, and freedom), attachment style, and feeling of guilt were studied as independent variables or a subjective factor of women’s psycho-emotional health.

The components of psychological well-being (autonomy and environmental management; personal growth, positive relations, life goals, and self-acceptance) are considered as dependent variable and indicators of psycho-emotional health.

Descriptive statistics are presented in Appendix A
Table A3. According to the data obtained by the Kolmogorov–Smirnov test, a normal distribution is observed for all variables, with the exception of indicators of closeness and separation of parents (*p* < 0.05). The indicator of closeness in the sample is biased toward high values, and separation is biased toward low values.

The major task at the first stage was to describe types of relations middle-aged women had to their parents, whom they provided everyday care for; and to define the place and role of guilt toward parents in the structure of relations between children and parents. The types were identified on the grounds of the results obtained with the help of the following measures: modified semantic differential/SD (Osgood) for family-conditioned states (15 bipolar subscales, combined in three scales: closeness, independence of parents/self-sufficiency, and freedom), Attachment Style Questionnaire (ASQ), and the Interpersonal Guilt Questionnaire-67 (IGQ).

The factor analysis enabled us to identify four relationship types middle-aged women could have with parents (four factors, 69.7% of the explained variance, principle component analysis, Varimax rotation of normalized data) (Table 1).

Type 1. *Anxious closeness* (factor loading is 2.9, 24.5% of the explained variance). This type is characterized by a high degree of closeness with parents (0.79), but at the same time the woman experiences a set of irrational forms of guilt: survivor’s guilt (0.8), separation guilt (0.89), and responsibility guilt (0.77).

Type 2. *Ambivalence of feelings* (factor loading is 2.6, 21.4% of the explained variance). This type is characterized by immersion in relationships with parents (0.79) and expressed need for their approval (0.72). At the same time, a woman sees parents as dependent, not separated, with a low autonomy level (−0.64), and experiences self-hatred, which destroys her (0.62).

Type 3. *Secondary relationship toward parents* (factor loading is 1.5, 12.5% of the explained variance). This type is characterized by freedom in relationships (0.64), relationships are perceived as secondary to personal achievements and own life (0.82).

Type 4. *Alienation* (factor loading is 1.3, 11.2% of the explained variance). When having this type of attitude toward parents, a woman experiences discomfort from closeness with a parent (0.42) and she feels insecure in a relationship with them (−0.93).

Then, to understand the role of relationship types in predicting the indicators of psycho-emotional health of middle-aged women included in the daily care of their parents, a simple linear regression was carried out. The independent variables were the four types of relations with parents, and the dependent variables were the components of women’s psychological well-being. The independent variables were obtained by the regression method. Simple linear regression results are shown in Table 2.

Thus, the type of relationship with parents, characterized by certain attachment style parameters and guilt is a predictor of psycho-emotional health indicators in middle-aged women and is related to their psychological well-being. The most vulnerable among women with role overload associated with the need to combine family and professional activities with the daily care of elderly parents are those women with alienated, ambivalent, and anxious types of attitudes toward their parents. It is accompanied with the feeling of guilt and decreasing female self-acceptance, it blurs the personal boundaries, and it depletes personality resources of middle-aged women.

### 3.2. Study 2

The problem question of this study is what is the role of the separation from mother type (a subjective factor) in psycho-emotional health of middle-aged women included in the daily care of elderly parents? The problem question of this study is what is the role of the separation from mother type (a subjective factor) in psycho-emotional health of middle-aged women included in the daily care of elderly parents? The independent variable was the separation from mother type. The indicators of psycho-emotional health (dependent variables) were meaning-value regulation (life goals, meaningfulness, me as locus of control, and control of life), emotional state (neurotic symptoms), aggression (intra-punitive vs. extra-punitive), and resources.

The indicators of psycho-emotional health included the variables from the formalized measure Purpose-in-Life (PIL) Test and semi-formalized projective instrument (the method of incomplete sentences and metaphors’ analysis, “My lifeline”). The drawing “My lifeline” and “the projective method of incomplete sentences” were processed by identifying and counting the formal and content-related indicators of the category under study that reflected psycho-emotional state of women. “My lifeline” identified the following categories: manifestations of neurotic symptoms in life-path drawing, manifestations of defensive aggression, and manifestations of resources and possibilities. The method of incomplete sentences enabled to identify the following basic criteria to describe the image of a woman’s future: definite image of future and anxiety in the description of future. It is worth mentioning that the content analysis of the data obtained through qualitative methods made it possible to use statistical analysis. Descriptive statistics are presented in Appendix A
Table A4.

The median filter (Me = Mo = 4) was used to identify the differences in the indicators of psycho-emotional health of middle-aged women connected with a separation from mother type, as a more general criterion; as a result, 20 people were excluded from this stage of the research. We had groups with higher values of separation on the scale separation type (*n* = 39; M = 4.4, SD = 0.35), which demonstrated a successful separation type. Groups with lower values (*n* = 26, M = 3.2, SD = 0.53) demonstrated a conflictive separation type. Then a number of significant differences in the indicators of psycho-emotional health in middle-aged women with successful and conflictive separation styles were identified (Table 3).

The conflictive separation type was dominating by the indicators of *a**nxiety in the description of future* (U = 295.5, *p* = 0.003), *neurotic symptoms in life-path drawing* (U = 306.5, *p* = 0.002), and *d**efinite image of future* (U = 340; *p* = 0.02).

Then, to identify the prediction of the indicators of psycho-emotional health of an adult daughter connected with a separation from mother type, simple linear regression analysis was carried out. The analysis involved a series of simple linear regressions with the continuous independent variable of “separation type”. The results shown in Table 4, with the separation from her mother type of a middle-aged woman served as the independent variable.

It was found that the type of separation from mothers predicted such indicators of psycho-emotional health in middle-aged women as meaning-value regulation (life goals, life process, “me” as locus of control, life as control locus, and meaningfulness of life), emotional state (neurotic symptoms), and aggression (intra-punitive vs. extra-punitive). The unfinished conflictive separation of an adult woman from her mother, especially in the vertical change, when an adult daughter performs parental functions toward her elderly mother may worsen her psycho-emotional health. It can possibly lead to an intrapersonal conflict that hinders new life prospects and becomes a source of difficult emotional states: high tension, anxiety, dissatisfaction, and inability to control own life.

## 4. Discussion

### 4.1. Discussion of Study 1

One of the important research results was the identification and description of four types of relations with parents of middle-aged women with a “triple” load.

Type 1. *Anxious closeness* is characterized by the experience of strong objective and subjective closeness with parents, as well as openness and trust toward them. However, these positive feelings are accompanied by a deep feeling of guilt toward parents—Survivor Guilt, Separation Guilt, Responsibility Guilt. A woman feels guilty toward parents, as they are not the only people who are important to her; her life is filled with other meaning (relationships with a partner, her own children, friends, professional activities, hobbies, etc.). She irrationally believes that her achievements negatively reflect on her parents, and her independence makes them feel lonely and abandoned. At the same time, she is convinced that she alone is responsible for her parents’ happiness and well-being, and she blames herself for not being able to give them even more than now.

Type 2. *Ambivalence of feelings*, on the one hand, is characterized by excessive immersion in relationships with parents, psychological dependence on them, and great need for their approval. At the same time, a woman realizes that parents are dependent on her and need her attention, support, help, love, and care. They strive to ensure that the daughter completely belongs to them, so that they are the main and only people for her. On the other hand, the inconsistency of the situation (except for the parents there is a married family, friends, professional activity, etc.), a conflict (parents denounce their daughter for insufficient involvement in their life, underestimate all her efforts) leads to the fact that she begins to feel the Guilt of self-hatred, negatively evaluates itself or depreciates altogether. This subjectively allows her to maintain a strong connection with the loved ones at the expense of her own well-being.

Type 3. *Secondary* to personal achievement and other life *relationship toward parents* is characterized by a sufficient distance in relations with parents, moderate satisfaction of a woman with these relationships. Relationships with parents are perceived as secondary to other important life spheres. A woman cares for and takes care of her parents, but at the same time, she is focused on her own life tasks. This may be due to her achievement of emotional independence from her parents. Her own life (marriage, family, profession, hobbies, and a social net) takes priority.

Type 4. *Alienation* takes place when a woman experiences discomfort from closeness and intimacy with parents and feels insecure in a relationship with them. Relationships are characterized by detachment, formalization, and general dissatisfaction with the situation of the relationship. She fulfills her obligations to them, but does not receive any positive feedback, recognition, and gratitude.

Psychological well-being and psycho-emotional health of a woman are strongly conjugated with the type of attitude she has toward her parents [19,40,41]. These relationships could become a relevant personal resource, enhancing the woman’s family identity and generational integrity, developing her understanding of life value and the inevitability of its end, and promoting the feeling of control over her own life. However, when major relationships with aging parents come to a crisis and new relationships are formed, women can develop a strengthening internal personal conflict. It ruins processes of self-identification and leads to roles’ misbalance. This lack of psychological well-being is aggravated by the life situation related to the role overload in middle-aged women, which leads to the deterioration of their psycho-emotional health.

Clarifying the role of the relationship types to one’s parents in predicting the psycho-emotional health indicators in middle-aged women who took care for elderly parents the following facts were revealed: The most dangerous for the psycho-emotional health of middle-aged women were the types *Alienation* and *Ambivalence of feelings*. The alienated type predicts a decline in most indicators of a woman’s psychological well-being: positive attitude toward others, a sense of autonomy, self-acceptance, personal growth, level of life goals, and a general well-being. Detachment, emotional coldness in relations with parents, and dissatisfaction with the needs for security and acceptance, undermine a woman’s confidence not only in relations with parents, but also to herself, the close ones, and reduce the quality of her life.

The conjugation of the ambivalent type with the most maladaptive form of guilt, guilt of self-hatred, also leads to negative consequences for a woman’s psycho-emotional health, reducing the indices of autonomy, self-acceptance, and general well-being.

The type *Anxious closeness* is also associated with irrational types of guilt, but, at the same time, less destructive, due to the fact that this type does not significantly affect the general indicator of the woman’s well-being. However, it still predicts a decrease in self-acceptance and personal autonomy, independence, and mastership of her life. It can be assumed that closeness within this type can contribute to maintaining positive relationships between women and others, including close people. Thanks to this, the risk of a woman’s psychological ill-being is reduced.

The safest and soundest type for the psycho-emotional health of a woman with a triple load is *Secondary relationship with parents*. This type predicts good skills in environmental management, which means an increase in the feeling of life control, pursuing own goals and needs. A woman feels like the subject of her own life. Emotionally, this type of relationship is the most constructive and resourceful, since it is not burdened with guilt and psychological dependence on aging parents.

An important characteristic of the relationship of a middle-aged woman with her elderly parents is the feelings of guilt toward them and the attachment style. Feeling guilty in adulthood is a psychological mechanism that motivates a woman to adhere to the eudemonic lifestyle, i.e., to actualize oneself via care for the “other”—in this case, for ageing parents and growing children [19,42]. On the other hand, the role of guilt in a female personal development is not so straightforward. High values of guilt can indicate the deterioration of personal autonomy, violated personal borders, the weakening of self-acceptance, difficulties in everyday life management, and lack of control over the situation. Thus, taking into account the context of the “triple” load, the type of relationship a woman has with her parents (which also includes the type of guilt), for whom she carries out daily care, is a predictor of her psychological well-being and psycho-emotional health.

### 4.2. Discussion of Study 2

The study of differences in the indicators of psycho-emotional health in women with a different type of separation from mothers showed the following important results. Women with a complete successful separation have a certain sense of life (meaningfulness of life and vivid life goals), believe in their abilities (life as locus of control), and are able to function successfully and independently, in the present and in the future (life process). Such women prefer active strategies aimed to solve problems and overcome difficulties, take the responsibility, and actualize internal and external resources (indicators of resources, possibilities and means).

Middle-aged women who are experiencing an incomplete and conflictive separation from mothers can demonstrate defensive aggressive reactions and expressly display the lack of emotional well-being. It was found with the method of drawing metaphors content analysis (indicators of neurotic symptoms in life-path drawing). The high level of anxiety, defensive and aggressive intentions, and fear can be regarded as a way to reduce a conflict and to get rid of responsibility for their own life in the future. The research demonstrated that all this could result from an improper process of separation from a mother.

At this stage, a woman’s life dependence on her mother may become a barrier for her own existential needs. It hinders planning and makes it more difficult for a woman to adapt to the situation in which she has to combine several roles—employee/professional, mother, wife, and caregiver/guardian. Describing their future, women see particular tasks (e.g., to have a child, to live at the seaside, etc.), but this future is associated with anxiety: “when I think of the future, I am afraid”, “I hope that the future will be peaceful, without problems”, “I am anxious when I think about my future”, “I am concerned about my future” (definite image of the future and anxiety in the description of the future). We consider it as a compensation for the undesired past and present and the immersion in the desirable future that will be free from the anxiety of separation (Purpose-in-Life Test; the projective method of incomplete sentences).

The results of the simple linear regression analysis showed that the style of separation from mothers in the adulthood was manifested by women via such psycho-emotional indicators as the meaningfulness of the present and the future, the ability of self-determination, and control over their own lives. The plans for the future are not illusions, but supported with taking responsibility for these plans. Emotionally, these women are more mature, and they are ready to show empathy and achieve the internal and external balance with themselves and the world around them.

The decrease in the separation indicators is accompanied by the lack of freedom from guilt, anxiety, lack of trust, responsibility, indignation, and anger toward mothers. The anxiety related to the perception of the personal life path increases. The lack of separation from the maternal image causes protective behavior that is manifested through auto- and hetero-aggression (neurotic symptoms, defensive aggressive, and a method of metaphor analysis) [19,40,42].

The increase in the anxiety indicators, while reflecting on the own life line and oneself as a subject of the own life under the conflictive separation, is regarded as being caused by the lack of independence and actual uncertainty of a woman in her own maturity.

## 5. Conclusions

The psychological research shows that the objectively difficult life situation of middle-aged women associated with their role overload, the need to combine various social roles—professional, wife, mother, and daughter, in relation to aging parents—reflects the specifics of the lifestyle of a modern woman [14,25]. We do not claim that this characterizes only Russian women’s way of life. Throughout the world, women in different cultures perform these functions. However, in collectivistic cultures, where the interests and values of the group prevail over personal interests, the consequences of such a lifestyle are specific. This specificity is associated with the action of both objective and subjective factors. In the actual study, the subjective factors included parameters of attachment style, the characteristics of the relationship with parents, the feeling of guilt, and separation from mother type. These factors are interrelated with each other, and the combination of their components defines four types of relationships of middle-aged women with their parents. It is important that all types of relationships are based on attachment style, combined with various forms of guilt. Each type of relationship is a predictor of a decrease or preservation of the psycho-emotional health of women.

The psycho-emotional health of overloaded middle-aged Russian women as a consequence of their hard life situation is conjugated by a subjective factor that integrates women’s relationships with their parents, attachment style, and separation from mother type.

Reliable indicators of distress and a decrease in the psycho-emotional health of middle-aged women with a “triple” load are as follows: states of high anxiety, tension, and dissatisfaction; negative super-critical attitude toward oneself; lack of confidence; difficulties or impossibility of life mastery; and anxiety in relation to their own future. An important positive contribution to the psycho-emotional health of a middle-aged woman is made by maintaining a high degree of emotional and cognitive autonomy, in combination with active participation in the life of parents and children, and the implementation of caring for them.

Study limitations, first and foremost, concern generalization of the conclusions made. The relative sample/group heterogeneity in the first and second studies, the excessive number of variables used, the absence of a control group (e.g., middle-aged women without a “triple” load), and cross-cultural comparison lack also prevent from wider conclusions.

Understanding of objective and subjective factors in the psycho-emotional health of middle-aged women by minimizing these limitations, focusing on the role of men in maintaining the psycho-emotional health of women facing triple load, and studying resources are among important research perspectives.

## Figures and Tables

**Table 1 ijerph-18-00388-t001:** Exploratory factor analysis results. Middle-aged women’s types of relations with their parents (*n* = 61).

Independent Variables ***	Factors ****			
	1	2	3	4
Separation guilt (2)	0.881			
Survivor’s guilt (2)	0.797			
Closeness (3)	0.785			
Responsibility guilt (2)	0.770			
Immersion in the relationship (1)		0.793		
Need of approval (1)		0.720		
Independence of parents/self-sufficiency (3)		−0.639		
Guilt of hatred (2)		0.622		
Closeness-related discomfort (1)				0.419
Relationships as secondary to achievements (1)			0.819	
Freedom (3)			0.643	
Confidence in self and others (1)				−0.928

* Notes: (1) Attachment Style Questionnaire (ASQ); (2) The Interpersonal Guilt Questionnaire-67 (IGQ); (3) modified semantic differential/SD (Osgood) for family-conditioned states. **** KMO measure of the adequacy of sample selection = 0.72; Bartlett’s sphericity test results—χ^2^ = 272.39; df = 66, *p* < 0.00. Principle components analysis, Varimax Rotation. Total % of variance explained: 69.7. Loadings below 0.4 are not presented.

**Table 2 ijerph-18-00388-t002:** Middle-aged women with a triple load’s types of relations with parents as predictors of their psycho-emotional health (*n* = 61).

Dependent Variables	Independent Variables											
	Anxious Closeness			Ambivalence of Feelings			Secondary Relationship toward Parents			Alienation		
	Β ***	*p*	R2	*β*	*p*	R2	*β*	*p*	R2	Β	*p*	R2
Positive relationships (1) ****	0.26	0.04	0.07							−0.4	0.00	0.16
Autonomy (1)	−0.29	0.03	0.08	−0.28	0.03	0.08				−0.41	0.00	0.17
Self-acceptance (1)	−0.32	0.01	0.1	−0.35	0.01	0.12				−0.38	0.00	0.14
Environmental management (1)							0.29	0.02	0.09			
Personal growth (1)										−0.27	0.04	0.07
Life goals (1)										−0.26	0.05	0.07
General psychological well-being (1)				−0.31	0.01	0.1				−0.45	0.00	0.2

* Notes: *β* value; *p*-value; % of explained variance R2. **** (1) The scales of psychological well-being.

**Table 3 ijerph-18-00388-t003:** Differences in the indicators of psycho-emotional health in middle-aged women with a successful (*n*1 = 39) and a conflictive (*n*2 = 26) type of separation from mothers.

Indicators of Women’s Psycho-Emotional Health	Mann–Whitney U Test	M Group with Successful Separation*n*1 = 39	M Group with Conflictive Separation *n*2 = 26
Life goals (2)	361.5 *	36.73	27.4
Life process (2)	361.5 *	36.73	27.4
Life as locus of control (2)	335.0 *	37.41	26.38
Meaningfulness of life (2)	336.5 *	37.37	26.44
Anxiety in the description of the future (3)	295.5 **	27.58	41.13
Definite image of future (3)	340.0 *	28.72	39.42
Indicators of neurotic symptoms in life-path drawing (4)	306.5 **	27.86	40.71
Indicators of resources, possibilities, and means (4)	349.0 *	37.05	26.92

Notes: * *p* ≤ 0.05, ** *p* ≤ 0.01, the grouping variable: Separation Style (2) Purpose-in-Life Test, (3) the projective method of incomplete sentences, and (4) the method of metaphors’ analysis.

**Table 4 ijerph-18-00388-t004:** Data on predicting the indicators of psycho-emotional health by separation type in middle-aged women (*n* = 85).

Dependent Variables	Regression Coefficient β	*p*-Value	Percent of Variance Explained R^2^
Life goals (2)	0.27	0.03	0.076
Life process (2)	0.32	0.00	0.105
Me as locus of control (2)	0.36	0.00	0.129
Life as locus of control (2)	0.24	0.05	0.059
Meaningfulness of life (2)	0.37	0,00	0.135
Anxiety in the description of the future (3)	−0.33	0.00	0.109
Indicators of neurotic symptoms in life-path drawing (4)	−0.35	0.00	0.122
Defensive aggression (4)	−0.24	0.05	0.057

Note: Independent variable—Separation style, (2) Purpose-in-Life Test; (3) the projective method of incomplete sentences; (4) the method of metaphors’ analysis.

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
