# Peer review of "A Decrease in Psycho-Emotional Health in Middle-Aged Russian Women Associated with Their Lifestyle"

_ijerph, 2021, doi:10.3390/ijerph18020388_

Round 1

Reviewer 1 Report

I am very happy to say that the manuscript "A decrease in psycho-emotional health in middle-aged Russian women associated with their lifestyle: factors and indicators" can be accepted for publication in your journal. A good job has been done by the authors in their last revision, so I have no further comments, except for this one: "Factors and indicators" should be removed from the title since they have no specific meanings.

Reviewer 2 Report

The reviewer's questions have been addressed.

Reviewer 3 Report

Thank you for letting me review the revised version of the manuscript.

The whole quality of the paper has been improved. No further concerns are noted by this Reviewer.

This manuscript is a resubmission of an earlier submission. The following is a list of the peer review reports and author responses from that submission.

Round 1

Reviewer 1 Report

This is a potentially interesting mixed method (qualitative and quantitative) study of 146 mid-aged women from a small town in Russia who were all looking after aging parents. A number of tests and interviews were done to determine the quality of the relationship with parents and the well being of the women, with an attempt to match the two.

I use the word "potentially" because I found the article difficult to understand. I believe that shortening it and making it both clearer and more concise would improve the paper considerably.

Questions

How were the women recruited? Were there two waves of recruitment? Did the participants know what was being studied? Attaching the consent form in an appendix would help to understand how the women were "primed."

Were all the women interviewed?

The staging was not clear to me. I think a diagram might help. What I understood is that the first 61 women were divided according to attachment categories and the second stage 85 (all different women?) were divided according to their well being and then attachment categories and well being categories were matched. Is that correct? Were they matched in the same or in different women?

What was the reason for not matching with women who were not looking after parents to see if parental attachment and burden was a significant factor in the women's well being?

The interpretation of findings needs to be more modest. Nothing was "proven."

Author Response

Response to Reviewer 1 Comments

This is a potentially interesting mixed method (qualitative and quantitative) study of 146 mid-aged women from a small town in Russia who were all looking after aging parents. A number of tests and interviews were done to determine the quality of the relationship with parents and the well being of the women, with an attempt to match the two.

Point1. I use the word "potentially" because I found the article difficult to understand. I believe that shortening it and making it both clearer and more concise would improve the paper considerably.

Response 1: The manuscript has undergone professional English editing in the Journal.

Questions

Point2. How were the women recruited? Were there two waves of recruitment? Did the participants know what was being studied? Attaching the consent form in an appendix would help to understand how the women were "primed."

Response 2: Thank you for the question. Yes, there have been two waves of recruitment.  The changes were made (see 2.1): it has been indicated that the sample consisted of two empirical groups of middle-aged women who gave a informed consent (see Appendix 1) and knew about the aim of the study. The first group took part in Study 1, the second group in Study 2.

Point 3. Were all the women interviewed?

The staging was not clear to me. I think a diagram might help. What I understood is that the first 61 women were divided according to attachment categories and the second stage 85 (all different women?) were divided according to their well being and then attachment categories and well being categories were matched. Is that correct? Were they matched in the same or in different women?

Response 3: Yes, all women were interviewed. The first group of women (N = 61) participated at the first stage, the research of women's attitudes towards their parents. Women were divided according to the attachment style. The criterion for dividing the second group of respondents (N = 85) in study 2 was the style of separation from Mother. These are two independent groups of respondents.

What was the reason for not matching with women who were not looking after parents to see if parental attachment and burden was a significant factor in the women's well being?

Response 4: We understand the limitations of this study: we did not research those middle-aged women who were not included in the care of the elderly parents. In order to obtain more complete and accurate data, it is planned to introduce a control group in the nearst study.

The interpretation of findings needs to be more modest. Nothing was "proven."

Response 5: We partially agree, we tried to be more moderate in interpretations. We consider it more adequate to speak not about the influence of the attachment style and separation type on women`s psychoemotional health, meaning prediction. We assume that a triple load situation under certain conditions can be a personal resource. Consideration of this issue requires further research.

Submission Date

03 November 2020

Date of this review

10 Nov 2020 13:17:56

Response to Reviewer 1 Comments

This is a potentially interesting mixed method (qualitative and quantitative) study of 146 mid-aged women from a small town in Russia who were all looking after aging parents. A number of tests and interviews were done to determine the quality of the relationship with parents and the well being of the women, with an attempt to match the two.

Point1. I use the word "potentially" because I found the article difficult to understand. I believe that shortening it and making it both clearer and more concise would improve the paper considerably.

Response 1: The manuscript has undergone professional English editing in the Journal.

Questions

Point2. How were the women recruited? Were there two waves of recruitment? Did the participants know what was being studied? Attaching the consent form in an appendix would help to understand how the women were "primed."

Response 2: Thank you for the question. Yes, there have been two waves of recruitment.  The changes were made (see 2.1): it has been indicated that the sample consisted of two empirical groups of middle-aged women who gave a informed consent (see Appendix 1) and knew about the aim of the study. The first group took part in Study 1, the second group in Study 2.

Point 3. Were all the women interviewed?

The staging was not clear to me. I think a diagram might help. What I understood is that the first 61 women were divided according to attachment categories and the second stage 85 (all different women?) were divided according to their well being and then attachment categories and well being categories were matched. Is that correct? Were they matched in the same or in different women?

Response 3: Yes, all women were interviewed. The first group of women (N = 61) participated at the first stage, the research of women's attitudes towards their parents. Women were divided according to the attachment style. The criterion for dividing the second group of respondents (N = 85) in study 2 was the style of separation from Mother. These are two independent groups of respondents.

What was the reason for not matching with women who were not looking after parents to see if parental attachment and burden was a significant factor in the women's well being?

Response 4: We understand the limitations of this study: we did not research those middle-aged women who were not included in the care of the elderly parents. In order to obtain more complete and accurate data, it is planned to introduce a control group in the nearst study.

The interpretation of findings needs to be more modest. Nothing was "proven."

Response 5: We partially agree, we tried to be more moderate in interpretations. We consider it more adequate to speak not about the influence of the attachment style and separation type on women`s psychoemotional health, meaning prediction. We assume that a triple load situation under certain conditions can be a personal resource. Consideration of this issue requires further research.

Submission Date

03 November 2020

Date of this review

10 Nov 2020 13:17:56

Response to Reviewer 1 Comments

This is a potentially interesting mixed method (qualitative and quantitative) study of 146 mid-aged women from a small town in Russia who were all looking after aging parents. A number of tests and interviews were done to determine the quality of the relationship with parents and the well being of the women, with an attempt to match the two.

Point1. I use the word "potentially" because I found the article difficult to understand. I believe that shortening it and making it both clearer and more concise would improve the paper considerably.

Response 1: The manuscript has undergone professional English editing in the Journal.

Questions

Point2. How were the women recruited? Were there two waves of recruitment? Did the participants know what was being studied? Attaching the consent form in an appendix would help to understand how the women were "primed."

Response 2: Thank you for the question. Yes, there have been two waves of recruitment.  The changes were made (see 2.1): it has been indicated that the sample consisted of two empirical groups of middle-aged women who gave a informed consent (see Appendix 1) and knew about the aim of the study. The first group took part in Study 1, the second group in Study 2.

Point 3. Were all the women interviewed?

The staging was not clear to me. I think a diagram might help. What I understood is that the first 61 women were divided according to attachment categories and the second stage 85 (all different women?) were divided according to their well being and then attachment categories and well being categories were matched. Is that correct? Were they matched in the same or in different women?

Response 3: Yes, all women were interviewed. The first group of women (N = 61) participated at the first stage, the research of women's attitudes towards their parents. Women were divided according to the attachment style. The criterion for dividing the second group of respondents (N = 85) in study 2 was the style of separation from Mother. These are two independent groups of respondents.

What was the reason for not matching with women who were not looking after parents to see if parental attachment and burden was a significant factor in the women's well being?

Response 4: We understand the limitations of this study: we did not research those middle-aged women who were not included in the care of the elderly parents. In order to obtain more complete and accurate data, it is planned to introduce a control group in the nearst study.

The interpretation of findings needs to be more modest. Nothing was "proven."

Response 5: We partially agree, we tried to be more moderate in interpretations. We consider it more adequate to speak not about the influence of the attachment style and separation type on women`s psychoemotional health, meaning prediction. We assume that a triple load situation under certain conditions can be a personal resource. Consideration of this issue requires further research.

Submission Date

03 November 2020

Date of this review

10 Nov 2020 13:17:56

Response to Reviewer 1 Comments

This is a potentially interesting mixed method (qualitative and quantitative) study of 146 mid-aged women from a small town in Russia who were all looking after aging parents. A number of tests and interviews were done to determine the quality of the relationship with parents and the well being of the women, with an attempt to match the two.

Point1. I use the word "potentially" because I found the article difficult to understand. I believe that shortening it and making it both clearer and more concise would improve the paper considerably.

Response 1: The manuscript has undergone professional English editing in the Journal.

Questions

Point2. How were the women recruited? Were there two waves of recruitment? Did the participants know what was being studied? Attaching the consent form in an appendix would help to understand how the women were "primed."

Response 2: Thank you for the question. Yes, there have been two waves of recruitment.  The changes were made (see 2.1): it has been indicated that the sample consisted of two empirical groups of middle-aged women who gave a informed consent (see Appendix 1) and knew about the aim of the study. The first group took part in Study 1, the second group in Study 2.

Point 3. Were all the women interviewed?

The staging was not clear to me. I think a diagram might help. What I understood is that the first 61 women were divided according to attachment categories and the second stage 85 (all different women?) were divided according to their well being and then attachment categories and well being categories were matched. Is that correct? Were they matched in the same or in different women?

Response 3: Yes, all women were interviewed. The first group of women (N = 61) participated at the first stage, the research of women's attitudes towards their parents. Women were divided according to the attachment style. The criterion for dividing the second group of respondents (N = 85) in study 2 was the style of separation from Mother. These are two independent groups of respondents.

What was the reason for not matching with women who were not looking after parents to see if parental attachment and burden was a significant factor in the women's well being?

Response 4: We understand the limitations of this study: we did not research those middle-aged women who were not included in the care of the elderly parents. In order to obtain more complete and accurate data, it is planned to introduce a control group in the nearst study.

The interpretation of findings needs to be more modest. Nothing was "proven."

Response 5: We partially agree, we tried to be more moderate in interpretations. We consider it more adequate to speak not about the influence of the attachment style and separation type on women`s psychoemotional health, meaning prediction. We assume that a triple load situation under certain conditions can be a personal resource. Consideration of this issue requires further research.

Submission Date

03 November 2020

Date of this review

10 Nov 2020 13:17:56

Response to Reviewer 1 Comments

This is a potentially interesting mixed method (qualitative and quantitative) study of 146 mid-aged women from a small town in Russia who were all looking after aging parents. A number of tests and interviews were done to determine the quality of the relationship with parents and the well being of the women, with an attempt to match the two.

Point1. I use the word "potentially" because I found the article difficult to understand. I believe that shortening it and making it both clearer and more concise would improve the paper considerably.

Response 1: The manuscript has undergone professional English editing in the Journal.

Questions

Point2. How were the women recruited? Were there two waves of recruitment? Did the participants know what was being studied? Attaching the consent form in an appendix would help to understand how the women were "primed."

Response 2: Thank you for the question. Yes, there have been two waves of recruitment.  The changes were made (see 2.1): it has been indicated that the sample consisted of two empirical groups of middle-aged women who gave a informed consent (see Appendix 1) and knew about the aim of the study. The first group took part in Study 1, the second group in Study 2.

Point 3. Were all the women interviewed?

The staging was not clear to me. I think a diagram might help. What I understood is that the first 61 women were divided according to attachment categories and the second stage 85 (all different women?) were divided according to their well being and then attachment categories and well being categories were matched. Is that correct? Were they matched in the same or in different women?

Response 3: Yes, all women were interviewed. The first group of women (N = 61) participated at the first stage, the research of women's attitudes towards their parents. Women were divided according to the attachment style. The criterion for dividing the second group of respondents (N = 85) in study 2 was the style of separation from Mother. These are two independent groups of respondents.

What was the reason for not matching with women who were not looking after parents to see if parental attachment and burden was a significant factor in the women's well being?

Response 4: We understand the limitations of this study: we did not research those middle-aged women who were not included in the care of the elderly parents. In order to obtain more complete and accurate data, it is planned to introduce a control group in the nearst study.

The interpretation of findings needs to be more modest. Nothing was "proven."

Response 5: We partially agree, we tried to be more moderate in interpretations. We consider it more adequate to speak not about the influence of the attachment style and separation type on women`s psychoemotional health, meaning prediction. We assume that a triple load situation under certain conditions can be a personal resource. Consideration of this issue requires further research.

Submission Date

03 November 2020

Date of this review

10 Nov 2020 13:17:56

Response to Reviewer 1 Comments

This is a potentially interesting mixed method (qualitative and quantitative) study of 146 mid-aged women from a small town in Russia who were all looking after aging parents. A number of tests and interviews were done to determine the quality of the relationship with parents and the well being of the women, with an attempt to match the two.

Point1. I use the word "potentially" because I found the article difficult to understand. I believe that shortening it and making it both clearer and more concise would improve the paper considerably.

Response 1: The manuscript has undergone professional English editing in the Journal.

Questions

Point2. How were the women recruited? Were there two waves of recruitment? Did the participants know what was being studied? Attaching the consent form in an appendix would help to understand how the women were "primed."

Response 2: Thank you for the question. Yes, there have been two waves of recruitment.  The changes were made (see 2.1): it has been indicated that the sample consisted of two empirical groups of middle-aged women who gave a informed consent (see Appendix 1) and knew about the aim of the study. The first group took part in Study 1, the second group in Study 2.

Point 3. Were all the women interviewed?

The staging was not clear to me. I think a diagram might help. What I understood is that the first 61 women were divided according to attachment categories and the second stage 85 (all different women?) were divided according to their well being and then attachment categories and well being categories were matched. Is that correct? Were they matched in the same or in different women?

Response 3: Yes, all women were interviewed. The first group of women (N = 61) participated at the first stage, the research of women's attitudes towards their parents. Women were divided according to the attachment style. The criterion for dividing the second group of respondents (N = 85) in study 2 was the style of separation from Mother. These are two independent groups of respondents.

What was the reason for not matching with women who were not looking after parents to see if parental attachment and burden was a significant factor in the women's well being?

Response 4: We understand the limitations of this study: we did not research those middle-aged women who were not included in the care of the elderly parents. In order to obtain more complete and accurate data, it is planned to introduce a control group in the nearst study.

The interpretation of findings needs to be more modest. Nothing was "proven."

Response 5: We partially agree, we tried to be more moderate in interpretations. We consider it more adequate to speak not about the influence of the attachment style and separation type on women`s psychoemotional health, meaning prediction. We assume that a triple load situation under certain conditions can be a personal resource. Consideration of this issue requires further research.

Submission Date

03 November 2020

Date of this review

10 Nov 2020 13:17:56

Response to Reviewer 1 Comments

This is a potentially interesting mixed method (qualitative and quantitative) study of 146 mid-aged women from a small town in Russia who were all looking after aging parents. A number of tests and interviews were done to determine the quality of the relationship with parents and the well being of the women, with an attempt to match the two.

Point1. I use the word "potentially" because I found the article difficult to understand. I believe that shortening it and making it both clearer and more concise would improve the paper considerably.

Response 1: The manuscript has undergone professional English editing in the Journal.

Questions

Point2. How were the women recruited? Were there two waves of recruitment? Did the participants know what was being studied? Attaching the consent form in an appendix would help to understand how the women were "primed."

Response 2: Thank you for the question. Yes, there have been two waves of recruitment.  The changes were made (see 2.1): it has been indicated that the sample consisted of two empirical groups of middle-aged women who gave a informed consent (see Appendix 1) and knew about the aim of the study. The first group took part in Study 1, the second group in Study 2.

Point 3. Were all the women interviewed?

The staging was not clear to me. I think a diagram might help. What I understood is that the first 61 women were divided according to attachment categories and the second stage 85 (all different women?) were divided according to their well being and then attachment categories and well being categories were matched. Is that correct? Were they matched in the same or in different women?

Response 3: Yes, all women were interviewed. The first group of women (N = 61) participated at the first stage, the research of women's attitudes towards their parents. Women were divided according to the attachment style. The criterion for dividing the second group of respondents (N = 85) in study 2 was the style of separation from Mother. These are two independent groups of respondents.

What was the reason for not matching with women who were not looking after parents to see if parental attachment and burden was a significant factor in the women's well being?

Response 4: We understand the limitations of this study: we did not research those middle-aged women who were not included in the care of the elderly parents. In order to obtain more complete and accurate data, it is planned to introduce a control group in the nearst study.

The interpretation of findings needs to be more modest. Nothing was "proven."

Response 5: We partially agree, we tried to be more moderate in interpretations. We consider it more adequate to speak not about the influence of the attachment style and separation type on women`s psychoemotional health, meaning prediction. We assume that a triple load situation under certain conditions can be a personal resource. Consideration of this issue requires further research.

Submission Date

03 November 2020

Date of this review

10 Nov 2020 13:17:56

Response to Reviewer 1 Comments

This is a potentially interesting mixed method (qualitative and quantitative) study of 146 mid-aged women from a small town in Russia who were all looking after aging parents. A number of tests and interviews were done to determine the quality of the relationship with parents and the well being of the women, with an attempt to match the two.

Point1. I use the word "potentially" because I found the article difficult to understand. I believe that shortening it and making it both clearer and more concise would improve the paper considerably.

Response 1: The manuscript has undergone professional English editing in the Journal.

Questions

Point2. How were the women recruited? Were there two waves of recruitment? Did the participants know what was being studied? Attaching the consent form in an appendix would help to understand how the women were "primed."

Response 2: Thank you for the question. Yes, there have been two waves of recruitment.  The changes were made (see 2.1): it has been indicated that the sample consisted of two empirical groups of middle-aged women who gave a informed consent (see Appendix 1) and knew about the aim of the study. The first group took part in Study 1, the second group in Study 2.

Point 3. Were all the women interviewed?

The staging was not clear to me. I think a diagram might help. What I understood is that the first 61 women were divided according to attachment categories and the second stage 85 (all different women?) were divided according to their well being and then attachment categories and well being categories were matched. Is that correct? Were they matched in the same or in different women?

Response 3: Yes, all women were interviewed. The first group of women (N = 61) participated at the first stage, the research of women's attitudes towards their parents. Women were divided according to the attachment style. The criterion for dividing the second group of respondents (N = 85) in study 2 was the style of separation from Mother. These are two independent groups of respondents.

What was the reason for not matching with women who were not looking after parents to see if parental attachment and burden was a significant factor in the women's well being?

Response 4: We understand the limitations of this study: we did not research those middle-aged women who were not included in the care of the elderly parents. In order to obtain more complete and accurate data, it is planned to introduce a control group in the nearst study.

The interpretation of findings needs to be more modest. Nothing was "proven."

Response 5: We partially agree, we tried to be more moderate in interpretations. We consider it more adequate to speak not about the influence of the attachment style and separation type on women`s psychoemotional health, meaning prediction. We assume that a triple load situation under certain conditions can be a personal resource. Consideration of this issue requires further research.

Submission Date

03 November 2020

Date of this review

10 Nov 2020 13:17:56

Response to Reviewer 1 Comments

This is a potentially interesting mixed method (qualitative and quantitative) study of 146 mid-aged women from a small town in Russia who were all looking after aging parents. A number of tests and interviews were done to determine the quality of the relationship with parents and the well being of the women, with an attempt to match the two.

Point1. I use the word "potentially" because I found the article difficult to understand. I believe that shortening it and making it both clearer and more concise would improve the paper considerably.

Response 1: The manuscript has undergone professional English editing in the Journal.

Questions

Point2. How were the women recruited? Were there two waves of recruitment? Did the participants know what was being studied? Attaching the consent form in an appendix would help to understand how the women were "primed."

Response 2: Thank you for the question. Yes, there have been two waves of recruitment.  The changes were made (see 2.1): it has been indicated that the sample consisted of two empirical groups of middle-aged women who gave a informed consent (see Appendix 1) and knew about the aim of the study. The first group took part in Study 1, the second group in Study 2.

Point 3. Were all the women interviewed?

The staging was not clear to me. I think a diagram might help. What I understood is that the first 61 women were divided according to attachment categories and the second stage 85 (all different women?) were divided according to their well being and then attachment categories and well being categories were matched. Is that correct? Were they matched in the same or in different women?

Response 3: Yes, all women were interviewed. The first group of women (N = 61) participated at the first stage, the research of women's attitudes towards their parents. Women were divided according to the attachment style. The criterion for dividing the second group of respondents (N = 85) in study 2 was the style of separation from Mother. These are two independent groups of respondents.

What was the reason for not matching with women who were not looking after parents to see if parental attachment and burden was a significant factor in the women's well being?

Response 4: We understand the limitations of this study: we did not research those middle-aged women who were not included in the care of the elderly parents. In order to obtain more complete and accurate data, it is planned to introduce a control group in the nearst study.

The interpretation of findings needs to be more modest. Nothing was "proven."

Response 5: We partially agree, we tried to be more moderate in interpretations. We consider it more adequate to speak not about the influence of the attachment style and separation type on women`s psychoemotional health, meaning prediction. We assume that a triple load situation under certain conditions can be a personal resource. Consideration of this issue requires further research.

Submission Date

03 November 2020

Date of this review

10 Nov 2020 13:17:56

Response to Reviewer 1 Comments

This is a potentially interesting mixed method (qualitative and quantitative) study of 146 mid-aged women from a small town in Russia who were all looking after aging parents. A number of tests and interviews were done to determine the quality of the relationship with parents and the well being of the women, with an attempt to match the two.

Point1. I use the word "potentially" because I found the article difficult to understand. I believe that shortening it and making it both clearer and more concise would improve the paper considerably.

Response 1: The manuscript has undergone professional English editing in the Journal.

Questions

Point2. How were the women recruited? Were there two waves of recruitment? Did the participants know what was being studied? Attaching the consent form in an appendix would help to understand how the women were "primed."

Response 2: Thank you for the question. Yes, there have been two waves of recruitment.  The changes were made (see 2.1): it has been indicated that the sample consisted of two empirical groups of middle-aged women who gave a informed consent (see Appendix 1) and knew about the aim of the study. The first group took part in Study 1, the second group in Study 2.

Point 3. Were all the women interviewed?

The staging was not clear to me. I think a diagram might help. What I understood is that the first 61 women were divided according to attachment categories and the second stage 85 (all different women?) were divided according to their well being and then attachment categories and well being categories were matched. Is that correct? Were they matched in the same or in different women?

Response 3: Yes, all women were interviewed. The first group of women (N = 61) participated at the first stage, the research of women's attitudes towards their parents. Women were divided according to the attachment style. The criterion for dividing the second group of respondents (N = 85) in study 2 was the style of separation from Mother. These are two independent groups of respondents.

What was the reason for not matching with women who were not looking after parents to see if parental attachment and burden was a significant factor in the women's well being?

Response 4: We understand the limitations of this study: we did not research those middle-aged women who were not included in the care of the elderly parents. In order to obtain more complete and accurate data, it is planned to introduce a control group in the nearst study.

The interpretation of findings needs to be more modest. Nothing was "proven."

Response 5: We partially agree, we tried to be more moderate in interpretations. We consider it more adequate to speak not about the influence of the attachment style and separation type on women`s psychoemotional health, meaning prediction. We assume that a triple load situation under certain conditions can be a personal resource. Consideration of this issue requires further research.

Submission Date

03 November 2020

Date of this review

10 Nov 2020 13:17:56

Response to Reviewer 1 Comments

This is a potentially interesting mixed method (qualitative and quantitative) study of 146 mid-aged women from a small town in Russia who were all looking after aging parents. A number of tests and interviews were done to determine the quality of the relationship with parents and the well being of the women, with an attempt to match the two.

Point1. I use the word "potentially" because I found the article difficult to understand. I believe that shortening it and making it both clearer and more concise would improve the paper considerably.

Response 1: The manuscript has undergone professional English editing in the Journal.

Questions

Point2. How were the women recruited? Were there two waves of recruitment? Did the participants know what was being studied? Attaching the consent form in an appendix would help to understand how the women were "primed."

Response 2: Thank you for the question. Yes, there have been two waves of recruitment.  The changes were made (see 2.1): it has been indicated that the sample consisted of two empirical groups of middle-aged women who gave a informed consent (see Appendix 1) and knew about the aim of the study. The first group took part in Study 1, the second group in Study 2.

Point 3. Were all the women interviewed?

The staging was not clear to me. I think a diagram might help. What I understood is that the first 61 women were divided according to attachment categories and the second stage 85 (all different women?) were divided according to their well being and then attachment categories and well being categories were matched. Is that correct? Were they matched in the same or in different women?

Response 3: Yes, all women were interviewed. The first group of women (N = 61) participated at the first stage, the research of women's attitudes towards their parents. Women were divided according to the attachment style. The criterion for dividing the second group of respondents (N = 85) in study 2 was the style of separation from Mother. These are two independent groups of respondents.

What was the reason for not matching with women who were not looking after parents to see if parental attachment and burden was a significant factor in the women's well being?

Response 4: We understand the limitations of this study: we did not research those middle-aged women who were not included in the care of the elderly parents. In order to obtain more complete and accurate data, it is planned to introduce a control group in the nearst study.

The interpretation of findings needs to be more modest. Nothing was "proven."

Response 5: We partially agree, we tried to be more moderate in interpretations. We consider it more adequate to speak not about the influence of the attachment style and separation type on women`s psychoemotional health, meaning prediction. We assume that a triple load situation under certain conditions can be a personal resource. Consideration of this issue requires further research.

Submission Date

03 November 2020

Date of this review

10 Nov 2020 13:17:56

Response to Reviewer 1 Comments

This is a potentially interesting mixed method (qualitative and quantitative) study of 146 mid-aged women from a small town in Russia who were all looking after aging parents. A number of tests and interviews were done to determine the quality of the relationship with parents and the well being of the women, with an attempt to match the two.

Point1. I use the word "potentially" because I found the article difficult to understand. I believe that shortening it and making it both clearer and more concise would improve the paper considerably.

Response 1: The manuscript has undergone professional English editing in the Journal.

Questions

Point2. How were the women recruited? Were there two waves of recruitment? Did the participants know what was being studied? Attaching the consent form in an appendix would help to understand how the women were "primed."

Response 2: Thank you for the question. Yes, there have been two waves of recruitment.  The changes were made (see 2.1): it has been indicated that the sample consisted of two empirical groups of middle-aged women who gave a informed consent (see Appendix 1) and knew about the aim of the study. The first group took part in Study 1, the second group in Study 2.

Point 3. Were all the women interviewed?

The staging was not clear to me. I think a diagram might help. What I understood is that the first 61 women were divided according to attachment categories and the second stage 85 (all different women?) were divided according to their well being and then attachment categories and well being categories were matched. Is that correct? Were they matched in the same or in different women?

Response 3: Yes, all women were interviewed. The first group of women (N = 61) participated at the first stage, the research of women's attitudes towards their parents. Women were divided according to the attachment style. The criterion for dividing the second group of respondents (N = 85) in study 2 was the style of separation from Mother. These are two independent groups of respondents.

What was the reason for not matching with women who were not looking after parents to see if parental attachment and burden was a significant factor in the women's well being?

Response 4: We understand the limitations of this study: we did not research those middle-aged women who were not included in the care of the elderly parents. In order to obtain more complete and accurate data, it is planned to introduce a control group in the nearst study.

The interpretation of findings needs to be more modest. Nothing was "proven."

Response 5: We partially agree, we tried to be more moderate in interpretations. We consider it more adequate to speak not about the influence of the attachment style and separation type on women`s psychoemotional health, meaning prediction. We assume that a triple load situation under certain conditions can be a personal resource. Consideration of this issue requires further research.

Submission Date

03 November 2020

Date of this review

10 Nov 2020 13:17:56

Response to Reviewer 1 Comments

This is a potentially interesting mixed method (qualitative and quantitative) study of 146 mid-aged women from a small town in Russia who were all looking after aging parents. A number of tests and interviews were done to determine the quality of the relationship with parents and the well being of the women, with an attempt to match the two.

Point1. I use the word "potentially" because I found the article difficult to understand. I believe that shortening it and making it both clearer and more concise would improve the paper considerably.

Response 1: The manuscript has undergone professional English editing in the Journal.

Questions

Point2. How were the women recruited? Were there two waves of recruitment? Did the participants know what was being studied? Attaching the consent form in an appendix would help to understand how the women were "primed."

Response 2: Thank you for the question. Yes, there have been two waves of recruitment.  The changes were made (see 2.1): it has been indicated that the sample consisted of two empirical groups of middle-aged women who gave a informed consent (see Appendix 1) and knew about the aim of the study. The first group took part in Study 1, the second group in Study 2.

Point 3. Were all the women interviewed?

The staging was not clear to me. I think a diagram might help. What I understood is that the first 61 women were divided according to attachment categories and the second stage 85 (all different women?) were divided according to their well being and then attachment categories and well being categories were matched. Is that correct? Were they matched in the same or in different women?

Response 3: Yes, all women were interviewed. The first group of women (N = 61) participated at the first stage, the research of women's attitudes towards their parents. Women were divided according to the attachment style. The criterion for dividing the second group of respondents (N = 85) in study 2 was the style of separation from Mother. These are two independent groups of respondents.

What was the reason for not matching with women who were not looking after parents to see if parental attachment and burden was a significant factor in the women's well being?

Response 4: We understand the limitations of this study: we did not research those middle-aged women who were not included in the care of the elderly parents. In order to obtain more complete and accurate data, it is planned to introduce a control group in the nearst study.

The interpretation of findings needs to be more modest. Nothing was "proven."

Response 5: We partially agree, we tried to be more moderate in interpretations. We consider it more adequate to speak not about the influence of the attachment style and separation type on women`s psychoemotional health, meaning prediction. We assume that a triple load situation under certain conditions can be a personal resource. Consideration of this issue requires further research.

Submission Date

03 November 2020

Date of this review

10 Nov 2020 13:17:56

Response to Reviewer 1 Comments

This is a potentially interesting mixed method (qualitative and quantitative) study of 146 mid-aged women from a small town in Russia who were all looking after aging parents. A number of tests and interviews were done to determine the quality of the relationship with parents and the well being of the women, with an attempt to match the two.

Point1. I use the word "potentially" because I found the article difficult to understand. I believe that shortening it and making it both clearer and more concise would improve the paper considerably.

Response 1: The manuscript has undergone professional English editing in the Journal.

Questions

Point2. How were the women recruited? Were there two waves of recruitment? Did the participants know what was being studied? Attaching the consent form in an appendix would help to understand how the women were "primed."

Response 2: Thank you for the question. Yes, there have been two waves of recruitment.  The changes were made (see 2.1): it has been indicated that the sample consisted of two empirical groups of middle-aged women who gave a informed consent (see Appendix 1) and knew about the aim of the study. The first group took part in Study 1, the second group in Study 2.

Point 3. Were all the women interviewed?

The staging was not clear to me. I think a diagram might help. What I understood is that the first 61 women were divided according to attachment categories and the second stage 85 (all different women?) were divided according to their well being and then attachment categories and well being categories were matched. Is that correct? Were they matched in the same or in different women?

Response 3: Yes, all women were interviewed. The first group of women (N = 61) participated at the first stage, the research of women's attitudes towards their parents. Women were divided according to the attachment style. The criterion for dividing the second group of respondents (N = 85) in study 2 was the style of separation from Mother. These are two independent groups of respondents.

What was the reason for not matching with women who were not looking after parents to see if parental attachment and burden was a significant factor in the women's well being?

Response 4: We understand the limitations of this study: we did not research those middle-aged women who were not included in the care of the elderly parents. In order to obtain more complete and accurate data, it is planned to introduce a control group in the nearst study.

The interpretation of findings needs to be more modest. Nothing was "proven."

Response 5: We partially agree, we tried to be more moderate in interpretations. We consider it more adequate to speak not about the influence of the attachment style and separation type on women`s psychoemotional health, meaning prediction. We assume that a triple load situation under certain conditions can be a personal resource. Consideration of this issue requires further research.

Submission Date

03 November 2020

Date of this review

10 Nov 2020 13:17:56

Response to Reviewer 1 Comments

This is a potentially interesting mixed method (qualitative and quantitative) study of 146 mid-aged women from a small town in Russia who were all looking after aging parents. A number of tests and interviews were done to determine the quality of the relationship with parents and the well being of the women, with an attempt to match the two.

Point1. I use the word "potentially" because I found the article difficult to understand. I believe that shortening it and making it both clearer and more concise would improve the paper considerably.

Response 1: The manuscript has undergone professional English editing in the Journal.

Questions

Point2. How were the women recruited? Were there two waves of recruitment? Did the participants know what was being studied? Attaching the consent form in an appendix would help to understand how the women were "primed."

Response 2: Thank you for the question. Yes, there have been two waves of recruitment.  The changes were made (see 2.1): it has been indicated that the sample consisted of two empirical groups of middle-aged women who gave a informed consent (see Appendix 1) and knew about the aim of the study. The first group took part in Study 1, the second group in Study 2.

Point 3. Were all the women interviewed?

The staging was not clear to me. I think a diagram might help. What I understood is that the first 61 women were divided according to attachment categories and the second stage 85 (all different women?) were divided according to their well being and then attachment categories and well being categories were matched. Is that correct? Were they matched in the same or in different women?

Response 3: Yes, all women were interviewed. The first group of women (N = 61) participated at the first stage, the research of women's attitudes towards their parents. Women were divided according to the attachment style. The criterion for dividing the second group of respondents (N = 85) in study 2 was the style of separation from Mother. These are two independent groups of respondents.

What was the reason for not matching with women who were not looking after parents to see if parental attachment and burden was a significant factor in the women's well being?

Response 4: We understand the limitations of this study: we did not research those middle-aged women who were not included in the care of the elderly parents. In order to obtain more complete and accurate data, it is planned to introduce a control group in the nearst study.

The interpretation of findings needs to be more modest. Nothing was "proven."

Response 5: We partially agree, we tried to be more moderate in interpretations. We consider it more adequate to speak not about the influence of the attachment style and separation type on women`s psychoemotional health, meaning prediction. We assume that a triple load situation under certain conditions can be a personal resource. Consideration of this issue requires further research.

Submission Date

03 November 2020

Date of this review

10 Nov 2020 13:17:56

Response to Reviewer 1 Comments

This is a potentially interesting mixed method (qualitative and quantitative) study of 146 mid-aged women from a small town in Russia who were all looking after aging parents. A number of tests and interviews were done to determine the quality of the relationship with parents and the well being of the women, with an attempt to match the two.

Point1. I use the word "potentially" because I found the article difficult to understand. I believe that shortening it and making it both clearer and more concise would improve the paper considerably.

Response 1: The manuscript has undergone professional English editing in the Journal.

Questions

Point2. How were the women recruited? Were there two waves of recruitment? Did the participants know what was being studied? Attaching the consent form in an appendix would help to understand how the women were "primed."

Response 2: Thank you for the question. Yes, there have been two waves of recruitment.  The changes were made (see 2.1): it has been indicated that the sample consisted of two empirical groups of middle-aged women who gave a informed consent (see Appendix 1) and knew about the aim of the study. The first group took part in Study 1, the second group in Study 2.

Point 3. Were all the women interviewed?

The staging was not clear to me. I think a diagram might help. What I understood is that the first 61 women were divided according to attachment categories and the second stage 85 (all different women?) were divided according to their well being and then attachment categories and well being categories were matched. Is that correct? Were they matched in the same or in different women?

Response 3: Yes, all women were interviewed. The first group of women (N = 61) participated at the first stage, the research of women's attitudes towards their parents. Women were divided according to the attachment style. The criterion for dividing the second group of respondents (N = 85) in study 2 was the style of separation from Mother. These are two independent groups of respondents.

What was the reason for not matching with women who were not looking after parents to see if parental attachment and burden was a significant factor in the women's well being?

Response 4: We understand the limitations of this study: we did not research those middle-aged women who were not included in the care of the elderly parents. In order to obtain more complete and accurate data, it is planned to introduce a control group in the nearst study.

The interpretation of findings needs to be more modest. Nothing was "proven."

Response 5: We partially agree, we tried to be more moderate in interpretations. We consider it more adequate to speak not about the influence of the attachment style and separation type on women`s psychoemotional health, meaning prediction. We assume that a triple load situation under certain conditions can be a personal resource. Consideration of this issue requires further research.

Submission Date

03 November 2020

Date of this review

10 Nov 2020 13:17:56

Response to Reviewer 1 Comments

This is a potentially interesting mixed method (qualitative and quantitative) study of 146 mid-aged women from a small town in Russia who were all looking after aging parents. A number of tests and interviews were done to determine the quality of the relationship with parents and the well being of the women, with an attempt to match the two.

Point1. I use the word "potentially" because I found the article difficult to understand. I believe that shortening it and making it both clearer and more concise would improve the paper considerably.

Response 1: The manuscript has undergone professional English editing in the Journal.

Questions

Point2. How were the women recruited? Were there two waves of recruitment? Did the participants know what was being studied? Attaching the consent form in an appendix would help to understand how the women were "primed."

Response 2: Thank you for the question. Yes, there have been two waves of recruitment.  The changes were made (see 2.1): it has been indicated that the sample consisted of two empirical groups of middle-aged women who gave a informed consent (see Appendix 1) and knew about the aim of the study. The first group took part in Study 1, the second group in Study 2.

Point 3. Were all the women interviewed?

The staging was not clear to me. I think a diagram might help. What I understood is that the first 61 women were divided according to attachment categories and the second stage 85 (all different women?) were divided according to their well being and then attachment categories and well being categories were matched. Is that correct? Were they matched in the same or in different women?

Response 3: Yes, all women were interviewed. The first group of women (N = 61) participated at the first stage, the research of women's attitudes towards their parents. Women were divided according to the attachment style. The criterion for dividing the second group of respondents (N = 85) in study 2 was the style of separation from Mother. These are two independent groups of respondents.

What was the reason for not matching with women who were not looking after parents to see if parental attachment and burden was a significant factor in the women's well being?

Response 4: We understand the limitations of this study: we did not research those middle-aged women who were not included in the care of the elderly parents. In order to obtain more complete and accurate data, it is planned to introduce a control group in the nearst study.

The interpretation of findings needs to be more modest. Nothing was "proven."

Response 5: We partially agree, we tried to be more moderate in interpretations. We consider it more adequate to speak not about the influence of the attachment style and separation type on women`s psychoemotional health, meaning prediction. We assume that a triple load situation under certain conditions can be a personal resource. Consideration of this issue requires further research.

Submission Date

03 November 2020

Date of this review

10 Nov 2020 13:17:56

Response to Reviewer 1 Comments

This is a potentially interesting mixed method (qualitative and quantitative) study of 146 mid-aged women from a small town in Russia who were all looking after aging parents. A number of tests and interviews were done to determine the quality of the relationship with parents and the well being of the women, with an attempt to match the two.

Point1. I use the word "potentially" because I found the article difficult to understand. I believe that shortening it and making it both clearer and more concise would improve the paper considerably.

Response 1: The manuscript has undergone professional English editing in the Journal.

Questions

Point2. How were the women recruited? Were there two waves of recruitment? Did the participants know what was being studied? Attaching the consent form in an appendix would help to understand how the women were "primed."

Response 2: Thank you for the question. Yes, there have been two waves of recruitment.  The changes were made (see 2.1): it has been indicated that the sample consisted of two empirical groups of middle-aged women who gave a informed consent (see Appendix 1) and knew about the aim of the study. The first group took part in Study 1, the second group in Study 2.

Point 3. Were all the women interviewed?

The staging was not clear to me. I think a diagram might help. What I understood is that the first 61 women were divided according to attachment categories and the second stage 85 (all different women?) were divided according to their well being and then attachment categories and well being categories were matched. Is that correct? Were they matched in the same or in different women?

Response 3: Yes, all women were interviewed. The first group of women (N = 61) participated at the first stage, the research of women's attitudes towards their parents. Women were divided according to the attachment style. The criterion for dividing the second group of respondents (N = 85) in study 2 was the style of separation from Mother. These are two independent groups of respondents.

What was the reason for not matching with women who were not looking after parents to see if parental attachment and burden was a significant factor in the women's well being?

Response 4: We understand the limitations of this study: we did not research those middle-aged women who were not included in the care of the elderly parents. In order to obtain more complete and accurate data, it is planned to introduce a control group in the nearst study.

The interpretation of findings needs to be more modest. Nothing was "proven."

Response 5: We partially agree, we tried to be more moderate in interpretations. We consider it more adequate to speak not about the influence of the attachment style and separation type on women`s psychoemotional health, meaning prediction. We assume that a triple load situation under certain conditions can be a personal resource. Consideration of this issue requires further research.

Submission Date

03 November 2020

Date of this review

10 Nov 2020 13:17:56

Response to Reviewer 1 Comments

This is a potentially interesting mixed method (qualitative and quantitative) study of 146 mid-aged women from a small town in Russia who were all looking after aging parents. A number of tests and interviews were done to determine the quality of the relationship with parents and the well being of the women, with an attempt to match the two.

Point1. I use the word "potentially" because I found the article difficult to understand. I believe that shortening it and making it both clearer and more concise would improve the paper considerably.

Response 1: The manuscript has undergone professional English editing in the Journal.

Questions

Point2. How were the women recruited? Were there two waves of recruitment? Did the participants know what was being studied? Attaching the consent form in an appendix would help to understand how the women were "primed."

Response 2: Thank you for the question. Yes, there have been two waves of recruitment.  The changes were made (see 2.1): it has been indicated that the sample consisted of two empirical groups of middle-aged women who gave a informed consent (see Appendix 1) and knew about the aim of the study. The first group took part in Study 1, the second group in Study 2.

Point 3. Were all the women interviewed?

The staging was not clear to me. I think a diagram might help. What I understood is that the first 61 women were divided according to attachment categories and the second stage 85 (all different women?) were divided according to their well being and then attachment categories and well being categories were matched. Is that correct? Were they matched in the same or in different women?

Response 3: Yes, all women were interviewed. The first group of women (N = 61) participated at the first stage, the research of women's attitudes towards their parents. Women were divided according to the attachment style. The criterion for dividing the second group of respondents (N = 85) in study 2 was the style of separation from Mother. These are two independent groups of respondents.

What was the reason for not matching with women who were not looking after parents to see if parental attachment and burden was a significant factor in the women's well being?

Response 4: We understand the limitations of this study: we did not research those middle-aged women who were not included in the care of the elderly parents. In order to obtain more complete and accurate data, it is planned to introduce a control group in the nearst study.

The interpretation of findings needs to be more modest. Nothing was "proven."

Response 5: We partially agree, we tried to be more moderate in interpretations. We consider it more adequate to speak not about the influence of the attachment style and separation type on women`s psychoemotional health, meaning prediction. We assume that a triple load situation under certain conditions can be a personal resource. Consideration of this issue requires further research.

Submission Date

03 November 2020

Date of this review

10 Nov 2020 13:17:56

Response to Reviewer 1 Comments

This is a potentially interesting mixed method (qualitative and quantitative) study of 146 mid-aged women from a small town in Russia who were all looking after aging parents. A number of tests and interviews were done to determine the quality of the relationship with parents and the well being of the women, with an attempt to match the two.

Point1. I use the word "potentially" because I found the article difficult to understand. I believe that shortening it and making it both clearer and more concise would improve the paper considerably.

Response 1: The manuscript has undergone professional English editing in the Journal.

Questions

Point2. How were the women recruited? Were there two waves of recruitment? Did the participants know what was being studied? Attaching the consent form in an appendix would help to understand how the women were "primed."

Response 2: Thank you for the question. Yes, there have been two waves of recruitment.  The changes were made (see 2.1): it has been indicated that the sample consisted of two empirical groups of middle-aged women who gave a informed consent (see Appendix 1) and knew about the aim of the study. The first group took part in Study 1, the second group in Study 2.

Point 3. Were all the women interviewed?

The staging was not clear to me. I think a diagram might help. What I understood is that the first 61 women were divided according to attachment categories and the second stage 85 (all different women?) were divided according to their well being and then attachment categories and well being categories were matched. Is that correct? Were they matched in the same or in different women?

Response 3: Yes, all women were interviewed. The first group of women (N = 61) participated at the first stage, the research of women's attitudes towards their parents. Women were divided according to the attachment style. The criterion for dividing the second group of respondents (N = 85) in study 2 was the style of separation from Mother. These are two independent groups of respondents.

What was the reason for not matching with women who were not looking after parents to see if parental attachment and burden was a significant factor in the women's well being?

Response 4: We understand the limitations of this study: we did not research those middle-aged women who were not included in the care of the elderly parents. In order to obtain more complete and accurate data, it is planned to introduce a control group in the nearst study.

The interpretation of findings needs to be more modest. Nothing was "proven."

Response 5: We partially agree, we tried to be more moderate in interpretations. We consider it more adequate to speak not about the influence of the attachment style and separation type on women`s psychoemotional health, meaning prediction. We assume that a triple load situation under certain conditions can be a personal resource. Consideration of this issue requires further research.

Submission Date

03 November 2020

Date of this review

10 Nov 2020 13:17:56

Response to Reviewer 1 Comments

This is a potentially interesting mixed method (qualitative and quantitative) study of 146 mid-aged women from a small town in Russia who were all looking after aging parents. A number of tests and interviews were done to determine the quality of the relationship with parents and the well being of the women, with an attempt to match the two.

Point1. I use the word "potentially" because I found the article difficult to understand. I believe that shortening it and making it both clearer and more concise would improve the paper considerably.

Response 1: The manuscript has undergone professional English editing in the Journal.

Questions

Point2. How were the women recruited? Were there two waves of recruitment? Did the participants know what was being studied? Attaching the consent form in an appendix would help to understand how the women were "primed."

Response 2: Thank you for the question. Yes, there have been two waves of recruitment.  The changes were made (see 2.1): it has been indicated that the sample consisted of two empirical groups of middle-aged women who gave a informed consent (see Appendix 1) and knew about the aim of the study. The first group took part in Study 1, the second group in Study 2.

Point 3. Were all the women interviewed?

The staging was not clear to me. I think a diagram might help. What I understood is that the first 61 women were divided according to attachment categories and the second stage 85 (all different women?) were divided according to their well being and then attachment categories and well being categories were matched. Is that correct? Were they matched in the same or in different women?

Response 3: Yes, all women were interviewed. The first group of women (N = 61) participated at the first stage, the research of women's attitudes towards their parents. Women were divided according to the attachment style. The criterion for dividing the second group of respondents (N = 85) in study 2 was the style of separation from Mother. These are two independent groups of respondents.

What was the reason for not matching with women who were not looking after parents to see if parental attachment and burden was a significant factor in the women's well being?

Response 4: We understand the limitations of this study: we did not research those middle-aged women who were not included in the care of the elderly parents. In order to obtain more complete and accurate data, it is planned to introduce a control group in the nearst study.

The interpretation of findings needs to be more modest. Nothing was "proven."

Response 5: We partially agree, we tried to be more moderate in interpretations. We consider it more adequate to speak not about the influence of the attachment style and separation type on women`s psychoemotional health, meaning prediction. We assume that a triple load situation under certain conditions can be a personal resource. Consideration of this issue requires further research.

Submission Date

03 November 2020

Date of this review

10 Nov 2020 13:17:56

Response to Reviewer 1 Comments

This is a potentially interesting mixed method (qualitative and quantitative) study of 146 mid-aged women from a small town in Russia who were all looking after aging parents. A number of tests and interviews were done to determine the quality of the relationship with parents and the well being of the women, with an attempt to match the two.

Point1. I use the word "potentially" because I found the article difficult to understand. I believe that shortening it and making it both clearer and more concise would improve the paper considerably.

Response 1: The manuscript has undergone professional English editing in the Journal.

Questions

Point2. How were the women recruited? Were there two waves of recruitment? Did the participants know what was being studied? Attaching the consent form in an appendix would help to understand how the women were "primed."

Response 2: Thank you for the question. Yes, there have been two waves of recruitment.  The changes were made (see 2.1): it has been indicated that the sample consisted of two empirical groups of middle-aged women who gave a informed consent (see Appendix 1) and knew about the aim of the study. The first group took part in Study 1, the second group in Study 2.

Point 3. Were all the women interviewed?

The staging was not clear to me. I think a diagram might help. What I understood is that the first 61 women were divided according to attachment categories and the second stage 85 (all different women?) were divided according to their well being and then attachment categories and well being categories were matched. Is that correct? Were they matched in the same or in different women?

Response 3: Yes, all women were interviewed. The first group of women (N = 61) participated at the first stage, the research of women's attitudes towards their parents. Women were divided according to the attachment style. The criterion for dividing the second group of respondents (N = 85) in study 2 was the style of separation from Mother. These are two independent groups of respondents.

What was the reason for not matching with women who were not looking after parents to see if parental attachment and burden was a significant factor in the women's well being?

Response 4: We understand the limitations of this study: we did not research those middle-aged women who were not included in the care of the elderly parents. In order to obtain more complete and accurate data, it is planned to introduce a control group in the nearst study.

The interpretation of findings needs to be more modest. Nothing was "proven."

Response 5: We partially agree, we tried to be more moderate in interpretations. We consider it more adequate to speak not about the influence of the attachment style and separation type on women`s psychoemotional health, meaning prediction. We assume that a triple load situation under certain conditions can be a personal resource. Consideration of this issue requires further research.

Submission Date

03 November 2020

Date of this review

10 Nov 2020 13:17:56

Response to Reviewer 1 Comments

This is a potentially interesting mixed method (qualitative and quantitative) study of 146 mid-aged women from a small town in Russia who were all looking after aging parents. A number of tests and interviews were done to determine the quality of the relationship with parents and the well being of the women, with an attempt to match the two.

Point1. I use the word "potentially" because I found the article difficult to understand. I believe that shortening it and making it both clearer and more concise would improve the paper considerably.

Response 1: The manuscript has undergone professional English editing in the Journal.

Questions

Point2. How were the women recruited? Were there two waves of recruitment? Did the participants know what was being studied? Attaching the consent form in an appendix would help to understand how the women were "primed."

Response 2: Thank you for the question. Yes, there have been two waves of recruitment.  The changes were made (see 2.1): it has been indicated that the sample consisted of two empirical groups of middle-aged women who gave a informed consent (see Appendix 1) and knew about the aim of the study. The first group took part in Study 1, the second group in Study 2.

Point 3. Were all the women interviewed?

The staging was not clear to me. I think a diagram might help. What I understood is that the first 61 women were divided according to attachment categories and the second stage 85 (all different women?) were divided according to their well being and then attachment categories and well being categories were matched. Is that correct? Were they matched in the same or in different women?

Response 3: Yes, all women were interviewed. The first group of women (N = 61) participated at the first stage, the research of women's attitudes towards their parents. Women were divided according to the attachment style. The criterion for dividing the second group of respondents (N = 85) in study 2 was the style of separation from Mother. These are two independent groups of respondents.

What was the reason for not matching with women who were not looking after parents to see if parental attachment and burden was a significant factor in the women's well being?

Response 4: We understand the limitations of this study: we did not research those middle-aged women who were not included in the care of the elderly parents. In order to obtain more complete and accurate data, it is planned to introduce a control group in the nearst study.

The interpretation of findings needs to be more modest. Nothing was "proven."

Response 5: We partially agree, we tried to be more moderate in interpretations.

Response to Reviewer 1 Comments

This is a potentially interesting mixed method (qualitative and quantitative) study of 146 mid-aged women from a small town in Russia who were all looking after aging parents. A number of tests and interviews were done to determine the quality of the relationship with parents and the well being of the women, with an attempt to match the two.

Point1. I use the word "potentially" because I found the article difficult to understand. I believe that shortening it and making it both clearer and more concise would improve the paper considerably.

Response 1: The manuscript has undergone professional English editing in the Journal.

Questions

Point2. How were the women recruited? Were there two waves of recruitment? Did the participants know what was being studied? Attaching the consent form in an appendix would help to understand how the women were "primed."

Response 2: Thank you for the question. Yes, there have been two waves of recruitment.  The changes were made (see 2.1): it has been indicated that the sample consisted of two empirical groups of middle-aged women who gave a informed consent (see Appendix 1) and knew about the aim of the study. The first group took part in Study 1, the second group in Study 2.

Point 3. Were all the women interviewed?

The staging was not clear to me. I think a diagram might help. What I understood is that the first 61 women were divided according to attachment categories and the second stage 85 (all different women?) were divided according to their well being and then attachment categories and well being categories were matched. Is that correct? Were they matched in the same or in different women?

Response 3: Yes, all women were interviewed. The first group of women (N = 61) participated at the first stage, the research of women's attitudes towards their parents. Women were divided according to the attachment style. The criterion for dividing the second group of respondents (N = 85) in study 2 was the style of separation from Mother. These are two independent groups of respondents.

What was the reason for not matching with women who were not looking after parents to see if parental attachment and burden was a significant factor in the women's well being?

Response 4: We understand the limitations of this study: we did not research those middle-aged women who were not included in the care of the elderly parents. In order to obtain more complete and accurate data, it is planned to introduce a control group in the nearst study.

The interpretation of findings needs to be more modest. Nothing was "proven."

Response 5: We partially agree, we tried to be more moderate in interpretations. We consider it more adequate to speak not about the influence of the attachment style and separation type on women`s psychoemotional health, meaning prediction. We assume that a triple load situation under certain conditions can be a personal resource. Consideration of this issue requires further research.

Submission Date

03 November 2020

Date of this review

10 Nov 2020 13:17:56

Response to Reviewer 1 Comments

This is a potentially interesting mixed method (qualitative and quantitative) study of 146 mid-aged women from a small town in Russia who were all looking after aging parents. A number of tests and interviews were done to determine the quality of the relationship with parents and the well being of the women, with an attempt to match the two.

Point1. I use the word "potentially" because I found the article difficult to understand. I believe that shortening it and making it both clearer and more concise would improve the paper considerably.

Response 1: The manuscript has undergone professional English editing in the Journal.

Questions

Point2. How were the women recruited? Were there two waves of recruitment? Did the participants know what was being studied? Attaching the consent form in an appendix would help to understand how the women were "primed."

Response 2: Thank you for the question. Yes, there have been two waves of recruitment.  The changes were made (see 2.1): it has been indicated that the sample consisted of two empirical groups of middle-aged women who gave a informed consent (see Appendix 1) and knew about the aim of the study. The first group took part in Study 1, the second group in Study 2.

Point 3. Were all the women interviewed?

The staging was not clear to me. I think a diagram might help. What I understood is that the first 61 women were divided according to attachment categories and the second stage 85 (all different women?) were divided according to their well being and then attachment categories and well being categories were matched. Is that correct? Were they matched in the same or in different women?

Response 3: Yes, all women were interviewed. The first group of women (N = 61) participated at the first stage, the research of women's attitudes towards their parents. Women were divided according to the attachment style. The criterion for dividing the second group of respondents (N = 85) in study 2 was the style of separation from Mother. These are two independent groups of respondents.

What was the reason for not matching with women who were not looking after parents to see if parental attachment and burden was a significant factor in the women's well being?

Response 4: We understand the limitations of this study: we did not research those middle-aged women who were not included in the care of the elderly parents. In order to obtain more complete and accurate data, it is planned to introduce a control group in the nearst study.

The interpretation of findings needs to be more modest. Nothing was "proven."

Response 5: We partially agree, we tried to be more moderate in interpretations. We consider it more adequate to speak not about the influence of the attachment style and separation type on women`s psychoemotional health, meaning prediction. We assume that a triple load situation under certain conditions can be a personal resource. Consideration of this issue requires further research.

Submission Date

03 November 2020

Date of this review

10 Nov 2020 13:17:56

Response to Reviewer 1 Comments

This is a potentially interesting mixed method (qualitative and quantitative) study of 146 mid-aged women from a small town in Russia who were all looking after aging parents. A number of tests and interviews were done to determine the quality of the relationship with parents and the well being of the women, with an attempt to match the two.

Point1. I use the word "potentially" because I found the article difficult to understand. I believe that shortening it and making it both clearer and more concise would improve the paper considerably.

Response 1: The manuscript has undergone professional English editing in the Journal.

Questions

Point2. How were the women recruited? Were there two waves of recruitment? Did the participants know what was being studied? Attaching the consent form in an appendix would help to understand how the women were "primed."

Response 2: Thank you for the question. Yes, there have been two waves of recruitment.  The changes were made (see 2.1): it has been indicated that the sample consisted of two empirical groups of middle-aged women who gave a informed consent (see Appendix 1) and knew about the aim of the study. The first group took part in Study 1, the second group in Study 2.

Point 3. Were all the women interviewed?

The staging was not clear to me. I think a diagram might help. What I understood is that the first 61 women were divided according to attachment categories and the second stage 85 (all different women?) were divided according to their well being and then attachment categories and well being categories were matched. Is that correct? Were they matched in the same or in different women?

Response 3: Yes, all women were interviewed. The first group of women (N = 61) participated at the first stage, the research of women's attitudes towards their parents. Women were divided according to the attachment style. The criterion for dividing the second group of respondents (N = 85) in study 2 was the style of separation from Mother. These are two independent groups of respondents.

What was the reason for not matching with women who were not looking after parents to see if parental attachment and burden was a significant factor in the women's well being?

Response 4: We understand the limitations of this study: we did not research those middle-aged women who were not included in the care of the elderly parents. In order to obtain more complete and accurate data, it is planned to introduce a control group in the nearst study.

The interpretation of findings needs to be more modest. Nothing was "proven."

Response 5: We partially agree, we tried to be more moderate in interpretations. We consider it more adequate to speak not about the influence of the attachment style and separation type on women`s psychoemotional health, meaning prediction. We assume that a triple load situation under certain conditions can be a personal resource. Consideration of this issue requires further research.

Submission Date

03 November 2020

Date of this review

10 Nov 2020 13:17:56

Response to Reviewer 1 Comments

This is a potentially interesting mixed method (qualitative and quantitative) study of 146 mid-aged women from a small town in Russia who were all looking after aging parents. A number of tests and interviews were done to determine the quality of the relationship with parents and the well being of the women, with an attempt to match the two.

Point1. I use the word "potentially" because I found the article difficult to understand. I believe that shortening it and making it both clearer and more concise would improve the paper considerably.

Response 1: The manuscript has undergone professional English editing in the Journal.

Questions

Point2. How were the women recruited? Were there two waves of recruitment? Did the participants know what was being studied? Attaching the consent form in an appendix would help to understand how the women were "primed."

Response 2: Thank you for the question. Yes, there have been two waves of recruitment.  The changes were made (see 2.1): it has been indicated that the sample consisted of two empirical groups of middle-aged women who gave a informed consent (see Appendix 1) and knew about the aim of the study. The first group took part in Study 1, the second group in Study 2.

Point 3. Were all the women interviewed?

The staging was not clear to me. I think a diagram might help. What I understood is that the first 61 women were divided according to attachment categories and the second stage 85 (all different women?) were divided according to their well being and then attachment categories and well being categories were matched. Is that correct? Were they matched in the same or in different women?

Response 3: Yes, all women were interviewed. The first group of women (N = 61) participated at the first stage, the research of women's attitudes towards their parents. Women were divided according to the attachment style. The criterion for dividing the second group of respondents (N = 85) in study 2 was the style of separation from Mother. These are two independent groups of respondents.

What was the reason for not matching with women who were not looking after parents to see if parental attachment and burden was a significant factor in the women's well being?

Response 4: We understand the limitations of this study: we did not research those middle-aged women who were not included in the care of the elderly parents. In order to obtain more complete and accurate data, it is planned to introduce a control group in the nearst study.

The interpretation of findings needs to be more modest. Nothing was "proven."

Response 5: We partially agree, we tried to be more moderate in interpretations. We consider it more adequate to speak not about the influence of the attachment style and separation type on women`s psychoemotional health, meaning prediction. We assume that a triple load situation under certain conditions can be a personal resource. Consideration of this issue requires further research.

Submission Date

03 November 2020

Date of this review

10 Nov 2020 13:17:56

Response to Reviewer 1 Comments

This is a potentially interesting mixed method (qualitative and quantitative) study of 146 mid-aged women from a small town in Russia who were all looking after aging parents. A number of tests and interviews were done to determine the quality of the relationship with parents and the well being of the women, with an attempt to match the two.

Point1. I use the word "potentially" because I found the article difficult to understand. I believe that shortening it and making it both clearer and more concise would improve the paper considerably.

Response 1: The manuscript has undergone professional English editing in the Journal.

Questions

Point2. How were the women recruited? Were there two waves of recruitment? Did the participants know what was being studied? Attaching the consent form in an appendix would help to understand how the women were "primed."

Response 2: Thank you for the question. Yes, there have been two waves of recruitment.  The changes were made (see 2.1): it has been indicated that the sample consisted of two empirical groups of middle-aged women who gave a informed consent (see Appendix 1) and knew about the aim of the study. The first group took part in Study 1, the second group in Study 2.

Point 3. Were all the women interviewed?

The staging was not clear to me. I think a diagram might help. What I understood is that the first 61 women were divided according to attachment categories and the second stage 85 (all different women?) were divided according to their well being and then attachment categories and well being categories were matched. Is that correct? Were they matched in the same or in different women?

Response 3: Yes, all women were interviewed. The first group of women (N = 61) participated at the first stage, the research of women's attitudes towards their parents. Women were divided according to the attachment style. The criterion for dividing the second group of respondents (N = 85) in study 2 was the style of separation from Mother. These are two independent groups of respondents.

What was the reason for not matching with women who were not looking after parents to see if parental attachment and burden was a significant factor in the women's well being?

Response 4: We understand the limitations of this study: we did not research those middle-aged women who were not included in the care of the elderly parents. In order to obtain more complete and accurate data, it is planned to introduce a control group in the nearst study.

The interpretation of findings needs to be more modest. Nothing was "proven."

Response 5: We partially agree, we tried to be more moderate in interpretations. We consider it more adequate to speak not about the influence of the attachment style and separation type on women`s psychoemotional health, meaning prediction. We assume that a triple load situation under certain conditions can be a personal resource. Consideration of this issue requires further research.

Submission Date

03 November 2020

Date of this review

10 Nov 2020 13:17:56

Response to Reviewer 1 Comments

This is a potentially interesting mixed method (qualitative and quantitative) study of 146 mid-aged women from a small town in Russia who were all looking after aging parents. A number of tests and interviews were done to determine the quality of the relationship with parents and the well being of the women, with an attempt to match the two.

Point1. I use the word "potentially" because I found the article difficult to understand. I believe that shortening it and making it both clearer and more concise would improve the paper considerably.

Response 1: The manuscript has undergone professional English editing in the Journal.

Questions

Point2. How were the women recruited? Were there two waves of recruitment? Did the participants know what was being studied? Attaching the consent form in an appendix would help to understand how the women were "primed."

Response 2: Thank you for the question. Yes, there have been two waves of recruitment.  The changes were made (see 2.1): it has been indicated that the sample consisted of two empirical groups of middle-aged women who gave a informed consent (see Appendix 1) and knew about the aim of the study. The first group took part in Study 1, the second group in Study 2.

Point 3. Were all the women interviewed?

The staging was not clear to me. I think a diagram might help. What I understood is that the first 61 women were divided according to attachment categories and the second stage 85 (all different women?) were divided according to their well being and then attachment categories and well being categories were matched. Is that correct? Were they matched in the same or in different women?

Response 3: Yes, all women were interviewed. The first group of women (N = 61) participated at the first stage, the research of women's attitudes towards their parents. Women were divided according to the attachment style. The criterion for dividing the second group of respondents (N = 85) in study 2 was the style of separation from Mother. These are two independent groups of respondents.

What was the reason for not matching with women who were not looking after parents to see if parental attachment and burden was a significant factor in the women's well being?

Response 4: We understand the limitations of this study: we did not research those middle-aged women who were not included in the care of the elderly parents. In order to obtain more complete and accurate data, it is planned to introduce a control group in the nearst study.

The interpretation of findings needs to be more modest. Nothing was "proven."

Response 5: We partially agree, we tried to be more moderate in interpretations. We consider it more adequate to speak not about the influence of the attachment style and separation type on women`s psychoemotional health, meaning prediction. We assume that a triple load situation under certain conditions can be a personal resource. Consideration of this issue requires further research.

Submission Date

03 November 2020

Date of this review

10 Nov 2020 13:17:56

Response to Reviewer 1 Comments

This is a potentially interesting mixed method (qualitative and quantitative) study of 146 mid-aged women from a small town in Russia who were all looking after aging parents. A number of tests and interviews were done to determine the quality of the relationship with parents and the well being of the women, with an attempt to match the two.

Point1. I use the word "potentially" because I found the article difficult to understand. I believe that shortening it and making it both clearer and more concise would improve the paper considerably.

Response 1: The manuscript has undergone professional English editing in the Journal.

Questions

Point2. How were the women recruited? Were there two waves of recruitment? Did the participants know what was being studied? Attaching the consent form in an appendix would help to understand how the women were "primed."

Response 2: Thank you for the question. Yes, there have been two waves of recruitment.  The changes were made (see 2.1): it has been indicated that the sample consisted of two empirical groups of middle-aged women who gave a informed consent (see Appendix 1) and knew about the aim of the study. The first group took part in Study 1, the second group in Study 2.

Point 3. Were all the women interviewed?

The staging was not clear to me. I think a diagram might help. What I understood is that the first 61 women were divided according to attachment categories and the second stage 85 (all different women?) were divided according to their well being and then attachment categories and well being categories were matched. Is that correct? Were they matched in the same or in different women?

Response 3: Yes, all women were interviewed. The first group of women (N = 61) participated at the first stage, the research of women's attitudes towards their parents. Women were divided according to the attachment style. The criterion for dividing the second group of respondents (N = 85) in study 2 was the style of separation from Mother. These are two independent groups of respondents.

What was the reason for not matching with women who were not looking after parents to see if parental attachment and burden was a significant factor in the women's well being?

Response 4: We understand the limitations of this study: we did not research those middle-aged women who were not included in the care of the elderly parents. In order to obtain more complete and accurate data, it is planned to introduce a control group in the nearst study.

The interpretation of findings needs to be more modest. Nothing was "proven."

Response 5: We partially agree, we tried to be more moderate in interpretations. We consider it more adequate to speak not about the influence of the attachment style and separation type on women`s psychoemotional health, meaning prediction. We assume that a triple load situation under certain conditions can be a personal resource. Consideration of this issue requires further research.

Submission Date

03 November 2020

Date of this review

10 Nov 2020 13:17:56

Response to Reviewer 1 Comments

This is a potentially interesting mixed method (qualitative and quantitative) study of 146 mid-aged women from a small town in Russia who were all looking after aging parents. A number of tests and interviews were done to determine the quality of the relationship with parents and the well being of the women, with an attempt to match the two.

Point1. I use the word "potentially" because I found the article difficult to understand. I believe that shortening it and making it both clearer and more concise would improve the paper considerably.

Response 1: The manuscript has undergone professional English editing in the Journal.

Questions

Point2. How were the women recruited? Were there two waves of recruitment? Did the participants know what was being studied? Attaching the consent form in an appendix would help to understand how the women were "primed."

Response 2: Thank you for the question. Yes, there have been two waves of recruitment.  The changes were made (see 2.1): it has been indicated that the sample consisted of two empirical groups of middle-aged women who gave a informed consent (see Appendix 1) and knew about the aim of the study. The first group took part in Study 1, the second group in Study 2.

Point 3. Were all the women interviewed?

The staging was not clear to me. I think a diagram might help. What I understood is that the first 61 women were divided according to attachment categories and the second stage 85 (all different women?) were divided according to their well being and then attachment categories and well being categories were matched. Is that correct? Were they matched in the same or in different women?

Response 3: Yes, all women were interviewed. The first group of women (N = 61) participated at the first stage, the research of women's attitudes towards their parents. Women were divided according to the attachment style. The criterion for dividing the second group of respondents (N = 85) in study 2 was the style of separation from Mother. These are two independent groups of respondents.

What was the reason for not matching with women who were not looking after parents to see if parental attachment and burden was a significant factor in the women's well being?

Response 4: We understand the limitations of this study: we did not research those middle-aged women who were not included in the care of the elderly parents. In order to obtain more complete and accurate data, it is planned to introduce a control group in the nearst study.

The interpretation of findings needs to be more modest. Nothing was "proven."

Response 5: We partially agree, we tried to be more moderate in interpretations. We consider it more adequate to speak not about the influence of the attachment style and separation type on women`s psychoemotional health, meaning prediction. We assume that a triple load situation under certain conditions can be a personal resource. Consideration of this issue requires further research.

Submission Date

03 November 2020

Date of this review

10 Nov 2020 13:17:56

Response to Reviewer 1 Comments

This is a potentially interesting mixed method (qualitative and quantitative) study of 146 mid-aged women from a small town in Russia who were all looking after aging parents. A number of tests and interviews were done to determine the quality of the relationship with parents and the well being of the women, with an attempt to match the two.

Point1. I use the word "potentially" because I found the article difficult to understand. I believe that shortening it and making it both clearer and more concise would improve the paper considerably.

Response 1: The manuscript has undergone professional English editing in the Journal.

Questions

Point2. How were the women recruited? Were there two waves of recruitment? Did the participants know what was being studied? Attaching the consent form in an appendix would help to understand how the women were "primed."

Response 2: Thank you for the question. Yes, there have been two waves of recruitment.  The changes were made (see 2.1): it has been indicated that the sample consisted of two empirical groups of middle-aged women who gave a informed consent (see Appendix 1) and knew about the aim of the study. The first group took part in Study 1, the second group in Study 2.

Point 3. Were all the women interviewed?

The staging was not clear to me. I think a diagram might help. What I understood is that the first 61 women were divided according to attachment categories and the second stage 85 (all different women?) were divided according to their well being and then attachment categories and well being categories were matched. Is that correct? Were they matched in the same or in different women?

Response 3: Yes, all women were interviewed. The first group of women (N = 61) participated at the first stage, the research of women's attitudes towards their parents. Women were divided according to the attachment style. The criterion for dividing the second group of respondents (N = 85) in study 2 was the style of separation from Mother. These are two independent groups of respondents.

What was the reason for not matching with women who were not looking after parents to see if parental attachment and burden was a significant factor in the women's well being?

Response 4: We understand the limitations of this study: we did not research those middle-aged women who were not included in the care of the elderly parents. In order to obtain more complete and accurate data, it is planned to introduce a control group in the nearst study.

The interpretation of findings needs to be more modest. Nothing was "proven."

Response 5: We partially agree, we tried to be more moderate in interpretations. We consider it more adequate to speak not about the influence of the attachment style and separation type on women`s psychoemotional health, meaning prediction. We assume that a triple load situation under certain conditions can be a personal resource. Consideration of this issue requires further research.

Submission Date

03 November 2020

Date of this review

10 Nov 2020 13:17:56

Response to Reviewer 1 Comments

This is a potentially interesting mixed method (qualitative and quantitative) study of 146 mid-aged women from a small town in Russia who were all looking after aging parents. A number of tests and interviews were done to determine the quality of the relationship with parents and the well being of the women, with an attempt to match the two.

Point1. I use the word "potentially" because I found the article difficult to understand. I believe that shortening it and making it both clearer and more concise would improve the paper considerably.

Response 1: The manuscript has undergone professional English editing in the Journal.

Questions

Point2. How were the women recruited? Were there two waves of recruitment? Did the participants know what was being studied? Attaching the consent form in an appendix would help to understand how the women were "primed."

Response 2: Thank you for the question. Yes, there have been two waves of recruitment.  The changes were made (see 2.1): it has been indicated that the sample consisted of two empirical groups of middle-aged women who gave a informed consent (see Appendix 1) and knew about the aim of the study. The first group took part in Study 1, the second group in Study 2.

Point 3. Were all the women interviewed?

The staging was not clear to me. I think a diagram might help. What I understood is that the first 61 women were divided according to attachment categories and the second stage 85 (all different women?) were divided according to their well being and then attachment categories and well being categories were matched. Is that correct? Were they matched in the same or in different women?

Response 3: Yes, all women were interviewed. The first group of women (N = 61) participated at the first stage, the research of women's attitudes towards their parents. Women were divided according to the attachment style. The criterion for dividing the second group of respondents (N = 85) in study 2 was the style of separation from Mother. These are two independent groups of respondents.

What was the reason for not matching with women who were not looking after parents to see if parental attachment and burden was a significant factor in the women's well being?

Response 4: We understand the limitations of this study: we did not research those middle-aged women who were not included in the care of the elderly parents. In order to obtain more complete and accurate data, it is planned to introduce a control group in the nearst study.

The interpretation of findings needs to be more modest. Nothing was "proven."

Response 5: We partially agree, we tried to be more moderate in interpretations. We consider it more adequate to speak not about the influence of the attachment style and separation type on women`s psychoemotional health, meaning prediction. We assume that a triple load situation under certain conditions can be a personal resource. Consideration of this issue requires further research.

Submission Date

03 November 2020

Date of this review

10 Nov 2020 13:17:56

Response to Reviewer 1 Comments

This is a potentially interesting mixed method (qualitative and quantitative) study of 146 mid-aged women from a small town in Russia who were all looking after aging parents. A number of tests and interviews were done to determine the quality of the relationship with parents and the well being of the women, with an attempt to match the two.

Point1. I use the word "potentially" because I found the article difficult to understand. I believe that shortening it and making it both clearer and more concise would improve the paper considerably.

Response 1: The manuscript has undergone professional English editing in the Journal.

Questions

Point2. How were the women recruited? Were there two waves of recruitment? Did the participants know what was being studied? Attaching the consent form in an appendix would help to understand how the women were "primed."

Response 2: Thank you for the question. Yes, there have been two waves of recruitment.  The changes were made (see 2.1): it has been indicated that the sample consisted of two empirical groups of middle-aged women who gave a informed consent (see Appendix 1) and knew about the aim of the study. The first group took part in Study 1, the second group in Study 2.

Point 3. Were all the women interviewed?

The staging was not clear to me. I think a diagram might help. What I understood is that the first 61 women were divided according to attachment categories and the second stage 85 (all different women?) were divided according to their well being and then attachment categories and well being categories were matched. Is that correct? Were they matched in the same or in different women?

Response 3: Yes, all women were interviewed. The first group of women (N = 61) participated at the first stage, the research of women's attitudes towards their parents. Women were divided according to the attachment style. The criterion for dividing the second group of respondents (N = 85) in study 2 was the style of separation from Mother. These are two independent groups of respondents.

What was the reason for not matching with women who were not looking after parents to see if parental attachment and burden was a significant factor in the women's well being?

Response 4: We understand the limitations of this study: we did not research those middle-aged women who were not included in the care of the elderly parents. In order to obtain more complete and accurate data, it is planned to introduce a control group in the nearst study.

The interpretation of findings needs to be more modest. Nothing was "proven."

Response 5: We partially agree, we tried to be more moderate in interpretations. We consider it more adequate to speak not about the influence of the attachment style and separation type on women`s psychoemotional health, meaning prediction. We assume that a triple load situation under certain conditions can be a personal resource. Consideration of this issue requires further research.

Submission Date

03 November 2020

Date of this review

10 Nov 2020 13:17:56

Response to Reviewer 1 Comments

This is a potentially interesting mixed method (qualitative and quantitative) study of 146 mid-aged women from a small town in Russia who were all looking after aging parents. A number of tests and interviews were done to determine the quality of the relationship with parents and the well being of the women, with an attempt to match the two.

Point1. I use the word "potentially" because I found the article difficult to understand. I believe that shortening it and making it both clearer and more concise would improve the paper considerably.

Response 1: The manuscript has undergone professional English editing in the Journal.

Questions

Point2. How were the women recruited? Were there two waves of recruitment? Did the participants know what was being studied? Attaching the consent form in an appendix would help to understand how the women were "primed."

Response 2: Thank you for the question. Yes, there have been two waves of recruitment.  The changes were made (see 2.1): it has been indicated that the sample consisted of two empirical groups of middle-aged women who gave a informed consent (see Appendix 1) and knew about the aim of the study. The first group took part in Study 1, the second group in Study 2.

Point 3. Were all the women interviewed?

The staging was not clear to me. I think a diagram might help. What I understood is that the first 61 women were divided according to attachment categories and the second stage 85 (all different women?) were divided according to their well being and then attachment categories and well being categories were matched. Is that correct? Were they matched in the same or in different women?

Response 3: Yes, all women were interviewed. The first group of women (N = 61) participated at the first stage, the research of women's attitudes towards their parents. Women were divided according to the attachment style. The criterion for dividing the second group of respondents (N = 85) in study 2 was the style of separation from Mother. These are two independent groups of respondents.

What was the reason for not matching with women who were not looking after parents to see if parental attachment and burden was a significant factor in the women's well being?

Response 4: We understand the limitations of this study: we did not research those middle-aged women who were not included in the care of the elderly parents. In order to obtain more complete and accurate data, it is planned to introduce a control group in the nearst study.

The interpretation of findings needs to be more modest. Nothing was "proven."

Response 5: We partially agree, we tried to be more moderate in interpretations. We consider it more adequate to speak not about the influence of the attachment style and separation type on women`s psychoemotional health, meaning prediction. We assume that a triple load situation under certain conditions can be a personal resource. Consideration of this issue requires further research.

Submission Date

03 November 2020

Date of this review

10 Nov 2020 13:17:56

Response to Reviewer 1 Comments

This is a potentially interesting mixed method (qualitative and quantitative) study of 146 mid-aged women from a small town in Russia who were all looking after aging parents. A number of tests and interviews were done to determine the quality of the relationship with parents and the well being of the women, with an attempt to match the two.

Point1. I use the word "potentially" because I found the article difficult to understand. I believe that shortening it and making it both clearer and more concise would improve the paper considerably.

Response 1: The manuscript has undergone professional English editing in the Journal.

Questions

Point2. How were the women recruited? Were there two waves of recruitment? Did the participants know what was being studied? Attaching the consent form in an appendix would help to understand how the women were "primed."

Response 2: Thank you for the question. Yes, there have been two waves of recruitment.  The changes were made (see 2.1): it has been indicated that the sample consisted of two empirical groups of middle-aged women who gave a informed consent (see Appendix 1) and knew about the aim of the study. The first group took part in Study 1, the second group in Study 2.

Point 3. Were all the women interviewed?

The staging was not clear to me. I think a diagram might help. What I understood is that the first 61 women were divided according to attachment categories and the second stage 85 (all different women?) were divided according to their well being and then attachment categories and well being categories were matched. Is that correct? Were they matched in the same or in different women?

Response 3: Yes, all women were interviewed. The first group of women (N = 61) participated at the first stage, the research of women's attitudes towards their parents. Women were divided according to the attachment style. The criterion for dividing the second group of respondents (N = 85) in study 2 was the style of separation from Mother. These are two independent groups of respondents.

What was the reason for not matching with women who were not looking after parents to see if parental attachment and burden was a significant factor in the women's well being?

Response 4: We understand the limitations of this study: we did not research those middle-aged women who were not included in the care of the elderly parents. In order to obtain more complete and accurate data, it is planned to introduce a control group in the nearst study.

The interpretation of findings needs to be more modest. Nothing was "proven."

Response 5: We partially agree, we tried to be more moderate in interpretations. We consider it more adequate to speak not about the influence of the attachment style and separation type on women`s psychoemotional health, meaning prediction. We assume that a triple load situation under certain conditions can be a personal resource. Consideration of this issue requires further research.

Submission Date

03 November 2020

Date of this review

10 Nov 2020 13:17:56

Response to Reviewer 1 Comments

This is a potentially interesting mixed method (qualitative and quantitative) study of 146 mid-aged women from a small town in Russia who were all looking after aging parents. A number of tests and interviews were done to determine the quality of the relationship with parents and the well being of the women, with an attempt to match the two.

Point1. I use the word "potentially" because I found the article difficult to understand. I believe that shortening it and making it both clearer and more concise would improve the paper considerably.

Response 1: The manuscript has undergone professional English editing in the Journal.

Questions

Point2. How were the women recruited? Were there two waves of recruitment? Did the participants know what was being studied? Attaching the consent form in an appendix would help to understand how the women were "primed."

Response 2: Thank you for the question. Yes, there have been two waves of recruitment.  The changes were made (see 2.1): it has been indicated that the sample consisted of two empirical groups of middle-aged women who gave a informed consent (see Appendix 1) and knew about the aim of the study. The first group took part in Study 1, the second group in Study 2.

Point 3. Were all the women interviewed?

The staging was not clear to me. I think a diagram might help. What I understood is that the first 61 women were divided according to attachment categories and the second stage 85 (all different women?) were divided according to their well being and then attachment categories and well being categories were matched. Is that correct? Were they matched in the same or in different women?

Response 3: Yes, all women were interviewed. The first group of women (N = 61) participated at the first stage, the research of women's attitudes towards their parents. Women were divided according to the attachment style. The criterion for dividing the second group of respondents (N = 85) in study 2 was the style of separation from Mother. These are two independent groups of respondents.

What was the reason for not matching with women who were not looking after parents to see if parental attachment and burden was a significant factor in the women's well being?

Response 4: We understand the limitations of this study: we did not research those middle-aged women who were not included in the care of the elderly parents. In order to obtain more complete and accurate data, it is planned to introduce a control group in the nearst study.

The interpretation of findings needs to be more modest. Nothing was "proven."

Response 5: We partially agree, we tried to be more moderate in interpretations. We consider it more adequate to speak not about the influence of the attachment style and separation type on women`s psychoemotional health, meaning prediction. We assume that a triple load situation under certain conditions can be a personal resource. Consideration of this issue requires further research.

Submission Date

03 November 2020

Date of this review

10 Nov 2020 13:17:56

Response to Reviewer 1 Comments

This is a potentially interesting mixed method (qualitative and quantitative) study of 146 mid-aged women from a small town in Russia who were all looking after aging parents. A number of tests and interviews were done to determine the quality of the relationship with parents and the well being of the women, with an attempt to match the two.

Point1. I use the word "potentially" because I found the article difficult to understand. I believe that shortening it and making it both clearer and more concise would improve the paper considerably.

Response 1: The manuscript has undergone professional English editing in the Journal.

Questions

Point2. How were the women recruited? Were there two waves of recruitment? Did the participants know what was being studied? Attaching the consent form in an appendix would help to understand how the women were "primed."

Response 2: Thank you for the question. Yes, there have been two waves of recruitment.  The changes were made (see 2.1): it has been indicated that the sample consisted of two empirical groups of middle-aged women who gave a informed consent (see Appendix 1) and knew about the aim of the study. The first group took part in Study 1, the second group in Study 2.

Point 3. Were all the women interviewed?

The staging was not clear to me. I think a diagram might help. What I understood is that the first 61 women were divided according to attachment categories and the second stage 85 (all different women?) were divided according to their well being and then attachment categories and well being categories were matched. Is that correct? Were they matched in the same or in different women?

Response 3: Yes, all women were interviewed. The first group of women (N = 61) participated at the first stage, the research of women's attitudes towards their parents. Women were divided according to the attachment style. The criterion for dividing the second group of respondents (N = 85) in study 2 was the style of separation from Mother. These are two independent groups of respondents.

What was the reason for not matching with women who were not looking after parents to see if parental attachment and burden was a significant factor in the women's well being?

Response 4: We understand the limitations of this study: we did not research those middle-aged women who were not included in the care of the elderly parents. In order to obtain more complete and accurate data, it is planned to introduce a control group in the nearst study.

The interpretation of findings needs to be more modest. Nothing was "proven."

Response 5: We partially agree, we tried to be more moderate in interpretations. We consider it more adequate to speak not about the influence of the attachment style and separation type on women`s psychoemotional health, meaning prediction. We assume that a triple load situation under certain conditions can be a personal resource. Consideration of this issue requires further research.

Submission Date

03 November 2020

Date of this review

10 Nov 2020 13:17:56

Response to Reviewer 1 Comments

This is a potentially interesting mixed method (qualitative and quantitative) study of 146 mid-aged women from a small town in Russia who were all looking after aging parents. A number of tests and interviews were done to determine the quality of the relationship with parents and the well being of the women, with an attempt to match the two.

Point1. I use the word "potentially" because I found the article difficult to understand. I believe that shortening it and making it both clearer and more concise would improve the paper considerably.

Response 1: The manuscript has undergone professional English editing in the Journal.

Questions

Point2. How were the women recruited? Were there two waves of recruitment? Did the participants know what was being studied? Attaching the consent form in an appendix would help to understand how the women were "primed."

Response 2: Thank you for the question. Yes, there have been two waves of recruitment.  The changes were made (see 2.1): it has been indicated that the sample consisted of two empirical groups of middle-aged women who gave a informed consent (see Appendix 1) and knew about the aim of the study. The first group took part in Study 1, the second group in Study 2.

Point 3. Were all the women interviewed?

The staging was not clear to me. I think a diagram might help. What I understood is that the first 61 women were divided according to attachment categories and the second stage 85 (all different women?) were divided according to their well being and then attachment categories and well being categories were matched. Is that correct? Were they matched in the same or in different women?

Response 3: Yes, all women were interviewed. The first group of women (N = 61) participated at the first stage, the research of women's attitudes towards their parents. Women were divided according to the attachment style. The criterion for dividing the second group of respondents (N = 85) in study 2 was the style of separation from Mother. These are two independent groups of respondents.

What was the reason for not matching with women who were not looking after parents to see if parental attachment and burden was a significant factor in the women's well being?

Response 4: We understand the limitations of this study: we did not research those middle-aged women who were not included in the care of the elderly parents. In order to obtain more complete and accurate data, it is planned to introduce a control group in the nearst study.

The interpretation of findings needs to be more modest. Nothing was "proven."

Response 5: We partially agree, we tried to be more moderate in interpretations. We consider it more adequate to speak not about the influence of the attachment style and separation type on women`s psychoemotional health, meaning prediction. We assume that a triple load situation under certain conditions can be a personal resource. Consideration of this issue requires further research.

Submission Date

03 November 2020

Date of this review

10 Nov 2020 13:17:56

Response to Reviewer 1 Comments

This is a potentially interesting mixed method (qualitative and quantitative) study of 146 mid-aged women from a small town in Russia who were all looking after aging parents. A number of tests and interviews were done to determine the quality of the relationship with parents and the well being of the women, with an attempt to match the two.

Point1. I use the word "potentially" because I found the article difficult to understand. I believe that shortening it and making it both clearer and more concise would improve the paper considerably.

Response 1: The manuscript has undergone professional English editing in the Journal.

Questions

Point2. How were the women recruited? Were there two waves of recruitment? Did the participants know what was being studied? Attaching the consent form in an appendix would help to understand how the women were "primed."

Response 2: Thank you for the question. Yes, there have been two waves of recruitment.  The changes were made (see 2.1): it has been indicated that the sample consisted of two empirical groups of middle-aged women who gave a informed consent (see Appendix 1) and knew about the aim of the study. The first group took part in Study 1, the second group in Study 2.

Point 3. Were all the women interviewed?

The staging was not clear to me. I think a diagram might help. What I understood is that the first 61 women were divided according to attachment categories and the second stage 85 (all different women?) were divided according to their well being and then attachment categories and well being categories were matched. Is that correct? Were they matched in the same or in different women?

Response 3: Yes, all women were interviewed. The first group of women (N = 61) participated at the first stage, the research of women's attitudes towards their parents. Women were divided according to the attachment style. The criterion for dividing the second group of respondents (N = 85) in study 2 was the style of separation from Mother. These are two independent groups of respondents.

What was the reason for not matching with women who were not looking after parents to see if parental attachment and burden was a significant factor in the women's well being?

Response 4: We understand the limitations of this study: we did not research those middle-aged women who were not included in the care of the elderly parents. In order to obtain more complete and accurate data, it is planned to introduce a control group in the nearst study.

The interpretation of findings needs to be more modest. Nothing was "proven."

Response 5: We partially agree, we tried to be more moderate in interpretations. We consider it more adequate to speak not about the influence of the attachment style and separation type on women`s psychoemotional health, meaning prediction. We assume that a triple load situation under certain conditions can be a personal resource. Consideration of this issue requires further research.

Submission Date

03 November 2020

Date of this review

10 Nov 2020 13:17:56

Response to Reviewer 1 Comments

This is a potentially interesting mixed method (qualitative and quantitative) study of 146 mid-aged women from a small town in Russia who were all looking after aging parents. A number of tests and interviews were done to determine the quality of the relationship with parents and the well being of the women, with an attempt to match the two.

Point1. I use the word "potentially" because I found the article difficult to understand. I believe that shortening it and making it both clearer and more concise would improve the paper considerably.

Response 1: The manuscript has undergone professional English editing in the Journal.

Questions

Point2. How were the women recruited? Were there two waves of recruitment? Did the participants know what was being studied? Attaching the consent form in an appendix would help to understand how the women were "primed."

Response 2: Thank you for the question. Yes, there have been two waves of recruitment.  The changes were made (see 2.1): it has been indicated that the sample consisted of two empirical groups of middle-aged women who gave a informed consent (see Appendix 1) and knew about the aim of the study. The first group took part in Study 1, the second group in Study 2.

Point 3. Were all the women interviewed?

The staging was not clear to me. I think a diagram might help. What I understood is that the first 61 women were divided according to attachment categories and the second stage 85 (all different women?) were divided according to their well being and then attachment categories and well being categories were matched. Is that correct? Were they matched in the same or in different women?

Response 3: Yes, all women were interviewed. The first group of women (N = 61) participated at the first stage, the research of women's attitudes towards their parents. Women were divided according to the attachment style. The criterion for dividing the second group of respondents (N = 85) in study 2 was the style of separation from Mother. These are two independent groups of respondents.

What was the reason for not matching with women who were not looking after parents to see if parental attachment and burden was a significant factor in the women's well being?

Response 4: We understand the limitations of this study: we did not research those middle-aged women who were not included in the care of the elderly parents. In order to obtain more complete and accurate data, it is planned to introduce a control group in the nearst study.

The interpretation of findings needs to be more modest. Nothing was "proven."

Response 5: We partially agree, we tried to be more moderate in interpretations. We consider it more adequate to speak not about the influence of the attachment style and separation type on women`s psychoemotional health, meaning prediction. We assume that a triple load situation under certain conditions can be a personal resource. Consideration of this issue requires further research.

Submission Date

03 November 2020

Date of this review

10 Nov 2020 13:17:56

Response to Reviewer 1 Comments

This is a potentially interesting mixed method (qualitative and quantitative) study of 146 mid-aged women from a small town in Russia who were all looking after aging parents. A number of tests and interviews were done to determine the quality of the relationship with parents and the well being of the women, with an attempt to match the two.

Point1. I use the word "potentially" because I found the article difficult to understand. I believe that shortening it and making it both clearer and more concise would improve the paper considerably.

Response 1: The manuscript has undergone professional English editing in the Journal.

Questions

Point2. How were the women recruited? Were there two waves of recruitment? Did the participants know what was being studied? Attaching the consent form in an appendix would help to understand how the women were "primed."

Response 2: Thank you for the question. Yes, there have been two waves of recruitment.  The changes were made (see 2.1): it has been indicated that the sample consisted of two empirical groups of middle-aged women who gave a informed consent (see Appendix 1) and knew about the aim of the study. The first group took part in Study 1, the second group in Study 2.

Point 3. Were all the women interviewed?

The staging was not clear to me. I think a diagram might help. What I understood is that the first 61 women were divided according to attachment categories and the second stage 85 (all different women?) were divided according to their well being and then attachment categories and well being categories were matched. Is that correct? Were they matched in the same or in different women?

Response 3: Yes, all women were interviewed. The first group of women (N = 61) participated at the first stage, the research of women's attitudes towards their parents. Women were divided according to the attachment style. The criterion for dividing the second group of respondents (N = 85) in study 2 was the style of separation from Mother. These are two independent groups of respondents.

What was the reason for not matching with women who were not looking after parents to see if parental attachment and burden was a significant factor in the women's well being?

Response 4: We understand the limitations of this study: we did not research those middle-aged women who were not included in the care of the elderly parents. In order to obtain more complete and accurate data, it is planned to introduce a control group in the nearst study.

The interpretation of findings needs to be more modest. Nothing was "proven."

Response 5: We partially agree, we tried to be more moderate in interpretations. We consider it more adequate to speak not about the influence of the attachment style and separation type on women`s psychoemotional health, meaning prediction. We assume that a triple load situation under certain conditions can be a personal resource. Consideration of this issue requires further research.

Submission Date

03 November 2020

Date of this review

10 Nov 2020 13:17:56

Response to Reviewer 1 Comments

This is a potentially interesting mixed method (qualitative and quantitative) study of 146 mid-aged women from a small town in Russia who were all looking after aging parents. A number of tests and interviews were done to determine the quality of the relationship with parents and the well being of the women, with an attempt to match the two.

Point1. I use the word "potentially" because I found the article difficult to understand. I believe that shortening it and making it both clearer and more concise would improve the paper considerably.

Response 1: The manuscript has undergone professional English editing in the Journal.

Questions

Point2. How were the women recruited? Were there two waves of recruitment? Did the participants know what was being studied? Attaching the consent form in an appendix would help to understand how the women were "primed."

Response 2: Thank you for the question. Yes, there have been two waves of recruitment.  The changes were made (see 2.1): it has been indicated that the sample consisted of two empirical groups of middle-aged women who gave a informed consent (see Appendix 1) and knew about the aim of the study. The first group took part in Study 1, the second group in Study 2.

Point 3. Were all the women interviewed?

The staging was not clear to me. I think a diagram might help. What I understood is that the first 61 women were divided according to attachment categories and the second stage 85 (all different women?) were divided according to their well being and then attachment categories and well being categories were matched. Is that correct? Were they matched in the same or in different women?

Response 3: Yes, all women were interviewed. The first group of women (N = 61) participated at the first stage, the research of women's attitudes towards their parents. Women were divided according to the attachment style. The criterion for dividing the second group of respondents (N = 85) in study 2 was the style of separation from Mother. These are two independent groups of respondents.

What was the reason for not matching with women who were not looking after parents to see if parental attachment and burden was a significant factor in the women's well being?

Response 4: We understand the limitations of this study: we did not research those middle-aged women who were not included in the care of the elderly parents. In order to obtain more complete and accurate data, it is planned to introduce a control group in the nearst study.

The interpretation of findings needs to be more modest. Nothing was "proven."

Response 5: We partially agree, we tried to be more moderate in interpretations. We consider it more adequate to speak not about the influence of the attachment style and separation type on women`s psychoemotional health, meaning prediction. We assume that a triple load situation under certain conditions can be a personal resource. Consideration of this issue requires further research.

Submission Date

03 November 2020

Date of this review

10 Nov 2020 13:17:56

We consider it more adequate to speak not about the influence of the attachment style and separation type on women`s psychoemotional health, meaning prediction. We assume that a triple load situation under certain conditions can be a personal resource. Consideration of this issue requires further research.

Submission Date

03 November 2020

Date of this review

10 Nov 2020 13:17:56

Reviewer 2 Report

I am really pleased to review the manuscript entitled “A decrease in psycho-emotional health in middleaged Russian women associated with their lifestyle: factors and indicators

The topic of this manuscript falls within the scope of IJERPH.

The paper was methodologically well written, and the authors decided to perform a research on a relevant and debated topic. However, few minor concerns have been underlined while reviewing the manuscript and should be resolved by the authors.

  • All along the manuscript, English language used is, by far, unclear. However, the text needs a major language check by a native English speaker person, in order to improve its readability.
  • Title: Change "middleaged" to "middle-aged"
  • Abstract: No concerns
  • Introduction: I would report only one needed improvement
    • L62-66: This sentence should be at least mitigated, I believe that parenting in the 2020s should no be considered only a woman duty.
  • Materials and methods: The methodology is clear and well described. One only addition needed
    • L104-109: insert an appropriate reference
  • Results: Description of included studies is clear and to the point. Synthesis of results is well written. There are no concerns to be reported.
  • Discussion:Authors should describe, at least briefly, the role of women in the modern family, including the division of home duties with the partner.

Author Response

Response to Reviewer 2 Comments

I am really pleased to review the manuscript entitled “A decrease in psycho-emotional health in middleaged Russian women associated with their lifestyle: factors and indicators

The topic of this manuscript falls within the scope of IJERPH.

The paper was methodologically well written, and the authors decided to perform a research on a relevant and debated topic. However, few minor concerns have been underlined while reviewing the manuscript and should be resolved by the authors.

Point 1. All along the manuscript, English language used is, by far, unclear. However, the text needs a major language check by a native English speaker person, in order to improve its readability.

Title: Change "middleaged" to "middle-aged"

Response 1: The manuscript has undergone professional English editing in the Journal now.

Abstract: No concerns

Point 2. Introduction: I would report only one needed improvement

L 62-66: This sentence should be at least mitigated, I believe that parenting in the 2020 s should not be considered only a woman duty.

Response 2: Changes have been made: it is shown that Russian men predominantly define their main role in the family by the role of breadwinner. Even when fathers are included in the role of caring for and raising children, women traditionally feel and bear responsibility for this in Russian families [2]. A man`s role in a woman's coping and dyadic coping with the stress of a triple load situation, a supporting male role in maintaining the psychoemotional health of a middle-aged woman may be an interesting prospect.

Materials and methods: The methodology is clear and well described. One only addition needed

Point 3. L 104-109: insert an appropriate reference

Response 3: done [2, 3]

Results: Description of included studies is clear and to the point. Synthesis of results is well written. There are no concerns to be reported.

Point 4. Discussion: Authors should describe, at least briefly, the role of women in the modern family, including the division of home duties with the partner.

Response 4: In the current study, the task of studying the distribution of hou

Response to Reviewer 2 Comments

I am really pleased to review the manuscript entitled “A decrease in psycho-emotional health in middleaged Russian women associated with their lifestyle: factors and indicators

The topic of this manuscript falls within the scope of IJERPH.

The paper was methodologically well written, and the authors decided to perform a research on a relevant and debated topic. However, few minor concerns have been underlined while reviewing the manuscript and should be resolved by the authors.

Point 1. All along the manuscript, English language used is, by far, unclear. However, the text needs a major language check by a native English speaker person, in order to improve its readability.

Title: Change "middleaged" to "middle-aged"

Response 1: The manuscript has undergone professional English editing in the Journal now.

Abstract: No concerns

Point 2. Introduction: I would report only one needed improvement

L 62-66: This sentence should be at least mitigated, I believe that parenting in the 2020 s should not be considered only a woman duty.

Response 2: Changes have been made: it is shown that Russian men predominantly define their main role in the family by the role of breadwinner. Even when fathers are included in the role of caring for and raising children, women traditionally feel and bear responsibility for this in Russian families [2]. A man`s role in a woman's coping and dyadic coping with the stress of a triple load situation, a supporting male role in maintaining the psychoemotional health of a middle-aged woman may be an interesting prospect.

Materials and methods: The methodology is clear and well described. One only addition needed

Point 3. L 104-109: insert an appropriate reference

Response 3: done [2, 3]

Results: Description of included studies is clear and to the point. Synthesis of results is well written. There are no concerns to be reported.

Point 4. Discussion: Authors should describe, at least briefly, the role of women in the modern family, including the division of home duties with the partner.

Response 4: In the current study, the task of studying the distribution of household responsibilities in the family hasn`t been set, which does not allow us to make interpretations.

Submission Date

03 November 2020

Date of this review

10 Nov 2020 13:54:44

Response to Reviewer 2 Comments

I am really pleased to review the manuscript entitled “A decrease in psycho-emotional health in middleaged Russian women associated with their lifestyle: factors and indicators

The topic of this manuscript falls within the scope of IJERPH.

The paper was methodologically well written, and the authors decided to perform a research on a relevant and debated topic. However, few minor concerns have been underlined while reviewing the manuscript and should be resolved by the authors.

Point 1. All along the manuscript, English language used is, by far, unclear. However, the text needs a major language check by a native English speaker person, in order to improve its readability.

Title: Change "middleaged" to "middle-aged"

Response 1: The manuscript has undergone professional English editing in the Journal now.

Abstract: No concerns

Point 2. Introduction: I would report only one needed improvement

L 62-66: This sentence should be at least mitigated, I believe that parenting in the 2020 s should not be considered only a woman duty.

Response 2: Changes have been made: it is shown that Russian men predominantly define their main role in the family by the role of breadwinner. Even when fathers are included in the role of caring for and raising children, women traditionally feel and bear responsibility for this in Russian families [2]. A man`s role in a woman's coping and dyadic coping with the stress of a triple load situation, a supporting male role in maintaining the psychoemotional health of a middle-aged woman may be an interesting prospect.

Materials and methods: The methodology is clear and well described. One only addition needed

Point 3. L 104-109: insert an appropriate reference

Response 3: done [2, 3]

Results: Description of included studies is clear and to the point. Synthesis of results is well written. There are no concerns to be reported.

Point 4. Discussion: Authors should describe, at least briefly, the role of women in the modern family, including the division of home duties with the partner.

Response 4: In the current study, the task of studying the distribution of household responsibilities in the family hasn`t been set, which does not allow us to make interpretations.

Submission Date

03 November 2020

Date of this review

10 Nov 2020 13:54:44

sehold responsibilities in the family hasn`t been set, which does not allow us to make interpretations.

Submission Date

03 November 2020

Date of this review

10 Nov 2020 13:54:44

Reviewer 3 Report

The aim of the paper entitled A decrease in psycho-emotional health in middle-aged Russian women associated with their lifestyle: factors and indicators was to establish relationships between objective (understood as a lifestyle) along with subjective factors (attitudes towards parents, attachment styles), and well-being of middle-aged women in Russia. The authors presented two studies based on a qualitative (interview, incomplete sentences, metaphors’ analysis) and a quantitative method (two sets of various scales). In both studies took part 146 women. The authors used content analysis and statistical procedures in analyses of their data. Results revealed that women’s well-being related to some forms of their attitudes toward parents.  

This research addresses the very important issue of a “triple load” faced by women, i.e. meeting demands of one’s professional career, fulfilling family duties, and caring for elderly relatives. Undoubtedly, combining all those duties would make a woman’s life difficult and affect her psychological and physical well-being, which is confirmed in psychological and sociological literature. The focus of the paper is on the third type of load, namely taking on obligations to one’s parents, and factors related to a woman’s well-being in such a situation. The psychological significance of this issue, and its importance for the society I can see as the strength and value of the research presented in this paper. One more strength of the research pertains to collecting a vast amount of data, thanks to using qualitative and quantitative instruments.  

However, many theoretical and methodological weaknesses prevent me from suggesting the publication in a present version of the manuscript. I outline my concerns below.

Introduction 

  • The authors underline that a “triple load” burden is a common problem in the lives of Russian women. This issue is researched widely around the world, showing that mere care for elder parents is not Russian women specificity, but such an experience is shared by almost all women in the West and East alike. But I couldn’t deny that there might be some unique aspects of Russian ways of life or culture which would make this burden heavier or maybe lighter. One can think, for example, about a higher level of cultural collectivism in Russia, reflected in attitudes towards elders, or levels of other cultural dimensions such as power distance, or masculinity. So, the question “Why Russian women” must be answered in the introduction.  
  • Authors declare in the paper's title and in the beginning of the abstract that the study is about life-style, i.e. objective factors and subjective factors related to psycho-emotional health of Russian women. I understand that objective factors pertain to the “triple load” mainly, or perhaps to other circumstances of women’s lives. But none of the objective factors are included in the research design. What is more, it would be not possible to treat a life-style/triple load/caregiving/difficult life situation as a variable of the study, since there is no variation in the sample, each participant does live obviously the same life-style! It is impossible to infer from the fact that everybody in the sample has a feature that this very feature affects anything. You need a control group to compare with those who do not have that feature. Here, it might be a group of women who are not caregivers to their parents. Or at least, you may take into account that caregivers somehow vary in that how they feel about their obligations, how many elders are taken care for, their own parents plus parents-in-law, or how they evaluate stress load in these situations. I understand that the researchers asked about the last issue in their interviews (rows 126, 127), but no data on that has not been provided.
  • Apparently, the research is designed to explore solely the role of subjective factors (relationships with parents). But paradoxically, these factors are not elaborated in the Introduction. They are only mentioned in one paragraph containing the research questions (rows 99-101). There is no theoretical background for introducing particular variables into the research project. For example, why attachment style matters in predicting the well-being of adults? Or a sense of guilt? There are well-known theories that may give good reasons for incorporating these in a research design. One can think also about other possible factors, such as a sense of duty, or empathy, or feeling gratitude, or being a role model for her own children, etc. Therefore, the authors must justify theoretically their choice of variables.  
  • The authors posed four research questions. The first - How does the objectively difficult life situation of middle-aged women, related to their role overload of lifestyle, make their psychological and emotional health worse?–is irrelevant to the study, as I explained above. The second - What is the role of such subjective factors as women’s attitude to parents, their attachment style, and the style of separation from parents in the process?—should get a solid theoretical ground, as well as the third one - Is there the feeling of guilt? The fourth question - What are the negative influences of the objective and subjective factors?—seems to be redundant as the two first questions asked also—more or less—about influence. This question sounds as being biased toward looking for only negative outcomes, while positive consequences seem to be not interesting for researchers. Why? It’s easy to imagine that hard work, especially hard work for others, and loved ones, in particular, might bring positive outcomes for a person, such as an enhancement of self-esteem, self-confidence, agency, gaining new skills, feeling good as a member of a community and many other. Research questions have to be re-formulated.
  • The authors do not articulate hypotheses, which is an acceptable way of proceeding in some studies, but they in further sections of the paper surprisingly refer to non-existing hypotheses. They write: To validate the main research hypothesis…. (row 221), to verify the hypothesis…. (row 242), to verify the research hypothesis, i.e. how the style of separation from mothers influences the indicators of psycho-emotional health in adult daughters (rows 256, 257). I believe, there is an enormous amount of knowledge accumulated in this area to put forward specific hypotheses.
  • There are several theoretical concepts of somewhat similar meanings that the authors use in their paper without defining them explicitly. These are the following constructs: psycho-emotional health, psychological health, emotional health, well-being, psychological well-being. They must be defined clearly.

Materials and methods

  • At the beginning of the section, the authors provide information that the region (where the study took place) occupies a rather inferior position along the quality of life dimension in Russia. I suppose this information applies to the aims of the study, but the authors have not explained its relevance.
  • In this paper, two studies are presented, based on different sets of instruments, and analyzed separately (in the section Results). So, the total sample also should be pooled into two samples and described separately as the “study 1 sample” and the “study 2 sample”. Also, sample descriptions should be based on more detailed demographic data, which has been obtained already. 
  • One of the major flaws of the paper concerns the measurement tools used in both studies. The authors simply listed them, giving no details (but names and references only), such as what variables a tool (scale) is supposed to measure, the number of items, its reliability, or any other essential information. Especially, providing an exhaustive description is even more important with projective methods (incomplete sentences and metaphors), and a modified semantic differential scale. Also, information about how the authors would hold with interview data, how the content analysis would be carried on, is required definitely. Perhaps it might be useful to put some vital information about methods in an appendix (i.e. listing interview questions, incomplete sentences, metaphors, categories of content analysis).  

Results

There are several serious limitations in the results section that should be acknowledged.

  • The results section usually starts with a presentation of psychometric properties of the key research variables put in a table. A table may contain mean values (M), standard deviation values (SD), reliability coefficients (Cronbach Alpha), and possibly intercorrelations between variables. Since the authors stated the data did not meet normality criteria (line 160), it should include also KS criteria or skewness indices. This kind of information is missing in both studies. 
  • STUDY 1. The authors write: The content analysis of the materials got via the interview “The image of parents and my life situation” and the subsequent data analysis enabled to identify four types of attitude a middle-aged woman can have towards her parents. But they didn’t offer any details about their approach of this kind of data handling, and how they come to the point—being able to perform advanced statistical analysis on the qualitative data! If I am not mistaken, the next paragraphs contain the results of factor analysis. One can tell this by reading a phrase such as the first factor accounting for 27% of the dispersion. If really factor analysis has been performed, the authors are obliged to present relevant data.
  • I must admit that it is very hard to follow the results of study 1. The logic of the presentation is rather confusing. First, we have learned that there are four types of attitudes toward parents (operationalized as outcomes of factor analysis): strong attachment, distancing attitude, ambivalent attitude, and normative/healthy closeness. Then, in the table 1 Correlations between attitudes of a middle-aged woman to her parents and irrational forms of her guilt (n=61), we can see different operationalizations, based on Osgood SD and Attachment Style Questionnaire (as the note under the table says – row 206).  Then, we are back to the first types of attitudes—in the table 4  Average values of the psycho-emotional health indicators in groups of women with different attitude types. In meantime, we have to deal with two other tables containing correlational data concerning psycho-emotional health. In order to avoid any confusion with the “attitude” and “attachment” constructs, I suggest providing precise theoretical definitions of the terms and operational ones (showing relevant indicators). 
  • Data displayed in the tables is incomplete. All correlation coefficients should be reported, even insignificant ones. Correlational data should be displayed in tables created in the form of a matrix, which is properly made in the table's 1 case. But the table 2 Correlations between guilt toward parents in middle-aged women and their psycho-emotional health (n=61) should be arranged as the 4 x 8 matrix, that is, 4 rows for 4 forms of guilt and 8 columns for 8 indicators of psycho-emotional health. The same thing should be done with the next table 3: 4 rows for 4 types of attachment and 8 columns for 8 indicators of psycho-emotional health.  As regards table 4, I think it contains mean values, improperly named as average values. As such, standard deviation values should be included. Reporting results of one-way analysis of variance statistic F along with degrees of freedom (df) should be displayed;  the latter is missing (rows 224-226).
  • STUDY 2. The authors used here four instruments. Two of them belong to projective techniques—“The projective method of incomplete sentences” and “The projective method of metaphors’ analysis”. Although the authors didn’t say it explicitly, I assume that data obtained with these techniques have been used for establishing indices of psycho-emotional health, and then employed in statistical procedures. The authors should give an exhaustive explanation of how they transformed projective material into numerical variables. And what are these variables? The sets of variables presented in the tables 5 & 6 are incompatible. 
  • And again, the table with basic psychometric properties of the study 2 variables is needed.
  • It seems the authors use the phrase “psycho-emotional health” as an umbrella term for many different psychological phenomena. These would be autonomy, self-acceptance, curious man as an open system in the study 1 (Tables 2 & 3), also life as a focus of control, anxiety in the description of the future, indicators of resources, possibilities and means, protective aggression—in the study 2 (Table 5 & 6)). It brings a lot of confusion about what are the actual objectives of the research. How such diverse concepts could be put into one bin? It needs to be explained thoroughly. 
  • The authors say that The objective of this research stage was to study how the style of separation from mothers influences the indicators of psycho-emotional health in middle-aged women (rows 239,240). In order to answer this question, they performed two series of analyses. The first one was a series of Mann-Whitney U tests, with the independent dichotomized variable “separation style”. They created this variable by a median split of the continuous variable based on the Psychological Separation Inventory results. The second analysis was a series of simple linear regression analyses, with the independent variable “separation style”, but the authors didn’t disclose whether it was a continuous or nominal/dichotomized variable. In both analyses—so-called—indicators of psycho-emotional health served as dependent variables. Results obtained with both procedures lead to almost the same conclusions about relationships between separation style and other variables (with minor differences). Assuming that indicators of psycho-emotional health are really justifiable and reliable variables (though I am not convinced about this, because they were based on projective material, but the authors didn’t show their ways of dealing with such matters), the correlational analysis would be a default procedure to get to know how the variables relate to each other. But perhaps there is a good reason behind the authors’ approach. In that case, the authors should disclose it. 
  • The authors should take care of using proper statistical terms and symbols. For example, instead of X =41.1, ó=3.5  – M = 41.1; SD = 3.5 (row 110);  instead of 27% of the dispersion – 27% of variance (row 179);  instead of one-factor dispersion analysis – one-way analysis of variance (row 221); instead of average values – mean values (row 229); instead of Dispersion percentage R2 – Percent of variance explained R2 (Table 6); instead of one-factor regression analysis – simple linear regression analysis (rows 257-258).

Discussion

  • Having so many doubts regarding the theoretical background and methodological procedures, I would not feel in a position to discuss results. So, I will skip this part of the manuscript. Just only one comment: limits of the study are not discussed at all.

In conclusion: All things considered, I encourage the authors to pursue their work on this important issue of the well-being of women. However, I would strongly recommend a substantial theoretical and methodological revision.   

Author Response

Response to Reviewer 3 Comments

The aim of the paper entitled A decrease in psycho-emotional health in middle-aged Russian women associated with their lifestyle: factors and indicators was to establish relationships between objective (understood as a lifestyle) along with subjective factors (attitudes towards parents, attachment styles), and well-being of middle-aged women in Russia. The authors presented two studies based on a qualitative (interview, incomplete sentences, metaphors’ analysis) and a quantitative method (two sets of various scales). In both studies took part 146 women. The authors used content analysis and statistical procedures in analyses of their data. Results revealed that women’s well-being related to some forms of their attitudes toward parents.  

This research addresses the very important issue of a “triple load” faced by women, i.e. meeting demands of one’s professional career, fulfilling family duties, and caring for elderly relatives. Undoubtedly, combining all those duties would make a woman’s life difficult and affect her psychological and physical well-being, which is confirmed in psychological and sociological literature. The focus of the paper is on the third type of load, namely taking on obligations to one’s parents, and factors related to a woman’s well-being in such a situation. The psychological significance of this issue, and its importance for the society I can see as the strength and value of the research presented in this paper. One more strength of the research pertains to collecting a vast amount of data, thanks to using qualitative and quantitative instruments.  

However, many theoretical and methodological weaknesses prevent me from suggesting the publication in a present version of the manuscript. I outline my concerns below.

Introduction 

Point 1. The authors underline that a “triple load” burden is a common problem in the lives of Russian women. This issue is researched widely around the world, showing that mere care for elder parents is not Russian women specificity, but such an experience is shared by almost all women in the West and East alike. But I couldn’t deny that there might be some unique aspects of Russian ways of life or culture which would make this burden heavier or maybe lighter. One can think, for example, about a higher level of cultural collectivism in Russia, reflected in attitudes towards elders, or levels of other cultural dimensions such as power distance, or masculinity. So, the question “Why Russian women” must be answered in the introduction.  

Response 1: The authors agree with the reviewer significantly: it is important, indeed, to show the grounds for considering the specifics of the situation of the triple load of Russian women.Corrections in the text have been made: it`s shown that middle-aged Russian women and their parents are to a great extent representatives of collectivist culture, its values` holders. Therefore, caring for parents is not only a necessity, but also a woman`s important personal need associated with values and also developmental tasks [see links 16; 9]

Point 2. Authors declare in the paper's title and in the beginning of the abstract that the study is about life-style, i.e. objective factors and subjective factors related to psycho-emotional health of Russian women. I understand that objective factors pertain to the “triple load” mainly, or perhaps to other circumstances of women’s lives. But none of the objective factors are included in the research design.

Response 2: The variable "triple" role loading, as an objective factor, is controlled while building the sample (section 2.1.) and is currently analyzed in the introduction.

Point 3. What is more, it would be not possible to treat a life-style/triple load/caregiving/difficult life situation as a variable of the study, since there is no variation in the sample, each participant does live obviously the same life-style! It is impossible to infer from the fact that everybody in the sample has a feature that this very feature affects anything. You need a control group to compare with those who do not have that feature. Here, it might be a group of women who are not caregivers to their parents.

Response 3: We`re aware that it is impossible to introduce a control group at this stage, as the research is aleady completed. However, this is an undisputed limitation of this study and requires further development in the nearest future. We pointed this out in the discussion.

Point 4. Or at least, you may take into account that caregivers somehow vary in that how they feel about their obligations, how many elders are taken care for, their own parents plus parents-in-law, or how they evaluate stress load in these situations. I understand that the researchers asked about the last issue in their interviews (rows 126, 127), but no data on that has not been provided.

Response 4: We consider it important to clarify that, in a face-to-face interview, each respondent was asked to describe his current life situation in a free form, to evaluate subjectively the amount of everyday stress loads (high, medium, low stress levels). 100% of respondents in both groups report high and medium levels of daily stress. These data is given in section 2.1.

Point 5. Apparently, the research is designed to explore solely the role of subjective factors (relationships with parents). But paradoxically, these factors are not elaborated in the Introduction. They are only mentioned in one paragraph containing the research questions (rows 99 -101). There is no theoretical background for introducing particular variables into the research project. For example, why attachment style matters in predicting the well-being of adults? Or a sense of guilt? There are well-known theories that may give good reasons for incorporating these in a research design. One can think also about other possible factors, such as a sense of duty, or empathy, or feeling gratitude, or being a role model for her own children, etc. Therefore, the authors must justify theoretically their choice of variables. 

Response 5: Thanks for the question, the authors edited the Introduction: the role of a subjective factor (style of attachment, separation type, the feeling of guilt) in predicting the state of psychoemotional health of middle-aged women is described.

Point 6. The authors posed four research questions. The first - How does the objectively difficult life situation of middle-aged women, related to their role overload of lifestyle, make their psychological and emotional health worse?–is irrelevant to the study, as I explained above. The second - What is the role of such subjective factors as women’s attitude to parents, their attachment style, and the style of separation from parents in the process?—should get a solid theoretical ground, as well as the third one - Is there the feeling of guilt? The fourth question - What are the negative influences of the objective and subjective factors?—seems to be redundant as the two first questions asked also—more or less—about influence. This question sounds as being biased toward looking for only negative outcomes, while positive consequences seem to be not interesting for researchers. Why? It’s easy to imagine that hard work, especially hard work for others, and loved ones, in particular, might bring positive outcomes for a person, such as an enhancement of self-esteem, self-confidence, agency, gaining new skills, feeling good as a member of a community and many other. Research questions have to be re-formulated.

Response 6: Research questions have been refined and reformulated. Now Research questions are: What is the specificity of attitudes towards parents among middle-aged women included in the daily care of their parents? What is the role of the subjective factor (their attachment style, the feeling of guilt to parents, the style of separation from parents) in their psycho-emotional health and well-being?

The authors do not articulate hypotheses, which is an acceptable way of proceeding in some studies, but they in further sections of the paper surprisingly refer to non-existing hypotheses. They write: To validate the main research hypothesis…. (row 221), to verify the hypothesis…. (row 242), to verify the research hypothesis, i.e. how the style of separation from mothers influences the indicators of psycho-emotional health in adult daughters (rows 256, 257). I believe, there is an enormous amount of knowledge accumulated in this area to put forward specific hypotheses.

Point 7. There are several theoretical concepts of somewhat similar meanings that the authors use in their paper without defining them explicitly. These are the following constructs: psycho-emotional health, psychological health, emotional health, well-being, psychological well-being. They must be defined clearly.

Response 7: The text has been revised theoretically and linguistically.

Materials and methods

Point 8. At the beginning of the section, the authors provide information that the region (where the study took place) occupies a rather inferior position along the quality of life dimension in Russia. I suppose this information applies to the aims of the study, but the authors have not explained its relevance.

Response 8: The relevance of research has been deepened and expanded

Point 9. In this paper, two studies are presented, based on different sets of instruments, and analyzed separately (in the section Results). So, the total sample also should be pooled into two samples and described separately as the “study 1 sample” and the “study 2 sample”. Also, sample descriptions should be based on more detailed demographic data, which has been obtained already. 

Response 9: 2 empirical groups are described more specifically.The nature of the perception of life hardship associated with a triple load situation and socio-demographic data is described separately for each group (n1=61; n2=85): age, gender, level of education; marital status, age of children; professional status, experience and duration of caregiving for elderly parents.

Point 10. One of the major flaws of the paper concerns the measurement tools used in both studies. The authors simply listed them, giving no details (but names and references only), such as what variables a tool (scale) is supposed to measure, the number of items, its reliability, or any other essential information. Especially, providing an exhaustive description is even more important with projective methods (incomplete sentences and metaphors), and a modified semantic differential scale. Also, information about how the authors would hold with interview data, how the content analysis would be carried on, is required definitely. Perhaps it might be useful to put some vital information about methods in an appendix (i.e. listing interview questions, incomplete sentences, metaphors, categories of content analysis). 

Response 10: R: We thank the reviewer for the important comment, which allowed to describe the methods according to the required standards.

Results

There are several serious limitations in the results section that should be acknowledged.

Point 11. The results section usually starts with a presentation of psychometric properties of the key research variables put in a table. A table may contain mean values (M), standard deviation values (SD), reliability coefficients (Cronbach Alpha), and possibly intercorrelations between variables. Since the authors stated the data did not meet normality criteria (line 160), it should include also KS criteria or skewness indices. This kind of information is missing in both studies.

Response 11: Descriptive statistics is given in Appendix â„– 3, 4, 5, 6.

Point 12.

STUDY 1. The authors write: The content analysis of the materials got via the interview “The image of parents and my life situation” and the subsequent data analysis enabled to identify four types of attitude a middle-aged woman can have towards her parents. But they didn’t offer any details about their approach of this kind of data handling, and how they come to the point—being able to perform advanced statistical analysis on the qualitative data! If I am not mistaken, the next paragraphs contain the results of factor analysis. One can tell this by reading a phrase such as the first factor accounting for 27% of the dispersion. If really factor analysis has been performed, the authors are obliged to present relevant data.

Response 12: Revision has been done: 1. the categories of content analysis indicated; 2. the reasons for including the interview results into statistical processing described; 3.factor loadings indicated.

Point 13. I must admit that it is very hard to follow the results of study 1. The logic of the presentation is rather confusing. First, we have learned that there are four types of attitudes toward parents (operationalized as outcomes of factor analysis): strong attachment, distancing attitude, ambivalent attitude, and normative/healthy closeness. Then, in the table 1 Correlations between attitudes of a middle-aged woman to her parents and irrational forms of her guilt (n=61), we can see different operationalizations, based on Osgood SD and Attachment Style Questionnaire (as the note under the table says – row 206).  Then, we are back to the first types of attitudes—in the table 4  Average values of the psycho-emotional health indicators in groups of women with different attitude types. In meantime, we have to deal with two other tables containing correlational data concerning psycho-emotional health. In order to avoid any confusion with the “attitude” and “attachment” constructs, I suggest providing precise theoretical definitions of the terms and operational ones (showing relevant indicators). 

Response 13: Due to the large number of variables, the study design is really difficult to embrace and read. To understand the logic of the variables` selection and their operationalization, there was a table given in Appendix 2.

Point 14. Data displayed in the tables is incomplete. All correlation coefficients should be reported, even insignificant ones. Correlational data should be displayed in tables created in the form of a matrix, which is properly made in the table's 1 case. But the table 2 Correlations between guilt toward parents in middle-aged women and their psycho-emotional health (n=61) should be arranged as the 4 x 8 matrix, that is, 4 rows for 4 forms of guilt and 8 columns for 8 indicators of psycho-emotional health. The same thing should be done with the next table 3: 4 rows for 4 types of attachment and 8 columns for 8 indicators of psycho-emotional health.  As regards table 4, I think it contains mean values, improperly named as average values. As such, standard deviation values should be included. Reporting results of one-way analysis of variance statistic F along with degrees of freedom (df) should be displayed;  the latter is missing (rows 224-226).

Response 14: Revision has been made: the data in the tables have become more complete, missing indicators have been added.

Point 15.  STUDY 2. The authors used here four instruments. Two of them belong to projective techniques—“The projective method of incomplete sentences” and “The projective method of metaphors’ analysis”. Although the authors didn’t say it explicitly, I assume that data obtained with these techniques have been used for establishing indices of psycho-emotional health, and then employed in statistical procedures. The authors should give an exhaustive explanation of how they transformed projective material into numerical variables. And what are these variables? The sets of variables presented in the tables 5 & 6 are incompatible.

Response 15: Revision has been made: 1. categories of content analysis are indicated; 2. the grounds for including projective methods` results into statistical processing are described; 3. tables` titles (â„–5 and â„–6) are unified, made alike.

Point 16.  And again, the table with basic psychometric properties of the study 2 variables is needed.

Response 16: Descriptive statistics is given in Appendix No. 3, No. 4, No. 5, No.6

Point 17. It seems the authors use the phrase “psycho-emotional health” as an umbrella term for many different psychological phenomena. These would be autonomy, self-acceptance, curious man as an open system in the study 1 (Tables 2 & 3), also life as a focus of control, anxiety in the description of the future, indicators of resources, possibilities and means, protective aggression—in the study 2 (Table 5 & 6)). It brings a lot of confusion about what are the actual objectives of the research. How such diverse concepts could be put into one bin? It needs to be explained thoroughly.

Response 17: The reviewer is right. In the Introduction, the definition of psychoemotional health is clarified, its indicators are given. Table 2 with the operationalization of variables is presented in Appendix.

The authors say that The objective of this research stage was to study how the style of separation from mothers influences the indicators of psycho-emotional health in middle-aged women (rows 239,240). In order to answer this question, they performed two series of analyses. The first one was a series of Mann-Whitney U tests, with the independent dichotomized variable “separation style”. They created this variable by a median split of the continuous variable based on the Psychological Separation Inventory results. The second analysis was a series of simple linear regression analyses, with the independent variable “separation style”, but the authors didn’t disclose whether it was a continuous or nominal/dichotomized variable. In both analyses—so-called—indicators of psycho-emotional health served as dependent variables. Results obtained with both procedures lead to almost the same conclusions about relationships between separation style and other variables (with minor differences).

Point 18. Assuming that indicators of psycho-emotional health are really justifiable and reliable variables (though I am not convinced about this, because they were based on projective material, but the authors didn’t show their ways of dealing with such matters), the correlational analysis would be a default procedure to get to know how the variables relate to each other. But perhaps there is a good reason behind the authors’ approach. In that case, the authors should disclose it.

Response 18: The variable Separation style at the first stage of Study 2 is dichotomous to identify differences. At the second stage, it is metric, continuous, which gives grounds to include it in simple linear regression analysis. The data of Mann-Whitney U-test is used to compare two independent groups. And the data of the simple linear regression analysis made it possible to reveal the research idea reflected in the articl title better.  Point 19. The authors should take care of using proper statistical terms and symbols. For example, instead of X =41.1, ó=3.5  – M = 41.1; SD = 3.5 (row 110);  instead of 27% of the dispersion – 27% of variance (row 179);  instead of one-factor dispersion analysis – one-way analysis of variance (row 221); instead of average values – mean values (row 229); instead of Dispersion percentage R2 – Percent of variance explained R2 (Table 6); instead of one-factor regression analysis – simple linear regression analysis (rows 257-258). Response 19: The text has been corrected, the terms revised.

Discussion

Point 20. Having so many doubts regarding the theoretical background and methodological procedures, I would not feel in a position to discuss results. So, I will skip this part of the manuscript. Just only one comment: limits of the study are not discussed at all.

In conclusion: All things considered, I encourage the authors to pursue their work on this important issue of the well-being of women. However, I would strongly recommend a substantial theoretical and methodological revision.   

Response 20: The authors thank the distinguished reviewer for reading the article so carefully, which has helped a lot.

Submission Date

03 November 2020

Date of this review

10 Dec 2020 17:04

Reviewer 4 Report

However, in order to better understand and disseminate the study, it should be improved according these points:

  1. Link the research questions (97-101), the presentation of the results in studies 1 (174-176) and 2 (239-240), discussion and the conclusions in the same logical order and sequence. For example, right now the aim of the study 1 (174-176), presented in results part differ from the reasearch questions (97-101) in the end of the introduction.
  2. To describe the research methodologies in more detail (144-154): to present the number of items, the nature of their evaluation (Likert type or other...), reliability coefficients (Cronbach α or other coefficient).
  3. In the discussion section, analyze the results at the level of phenomena by discarding the presented p (299-304 and so on)
  4. Let the conclusions correspond to the questions raised and answer them in the results section, or let there be more research questions.
  5. In the text it was used hypothesis term, but this research raised questions, not hypothesis (221,256). So you need to eliminate the usage of hypothesis or formulate hypothesis in introduction part.
  6. Correlation didn‘t indicate influences (222-223)

Author Response

Response to Reviewer 4 Comments

However, in order to better understand and disseminate the study, it should be improved according these points:

  1. Point 1. Link the research questions (97-101), the presentation of the results in studies 1 (174-176) and 2 (239-240), discussion and the conclusions in the same logical order and sequence. For example, right now the aim of the study 1 (174-176), presented in results part differ from the reasearch questions (97-101) in the end of the introduction.

Response 1: Thanks to the reviewer1s suggestion, necessary corrections were made at the beginning and at the end of the article. Research questions have been refined and reformulated. Now Research questions are: What is the specificity of attitudes towards parents among middle-aged women included in the daily care of their parents? What is the role of the subjective factor (their attachment style, the feeling of guilt to parents, the style of separation from parents) in their psycho-emotional health and well-being? The logic of the presentation of the results is brought in accordance with the questions asked.

  1. Point 2. To describe the research methodologies in more detail (144-154): to present the number of items, the nature of their evaluation (Likert type or other...), reliability coefficients (Cronbach α or other coefficient).

Response 2: We are grateful to the reviewer for an important comment that helped bring methods description (2. 2) to the required standards. We`ve described the techniques in more detail, indicating number of items, a Likert scale, reliability coefficient (Cronbach alpha),  evaluation examples in Russian samples. 

  1. Point 3. In the discussion section, analyze the results at the level of phenomena by discarding the presented p (299-304 and so on)

Response 3: In the discussion section, p values ​​have been excluded and the results are presented at the level of phenomena, focusing on main ideas.

  1. Point 4. Let the conclusions correspond to the questions raised and answer them in the results section, or let there be more research questions.

Response 4: Once edited, the conclusions are consistent with the research questions (see section 5).

  1. Point 5. In Point 5the text it was used hypothesis term, but this research raised questions, not hypothesis (221, 256). So you need to eliminate the usage of hypothesis or formulate hypothesis in introduction part.

Response 5: The Text has been re-edited, the term hypothesis  omitted.

  1. Point 6. Correlation didn‘t indicate influences (222-223) 

Response 6: It is true, the term "influence" has been replaced by “prediction”. In fact the use of simple linear regression analysis allowed to draw appropriate conclusions.

Submission Date

03 November 2020

Date of this review

01 Dec 2020 19:03:48

Response to Reviewer 4 Comments

However, in order to better understand and disseminate the study, it should be improved according these points:

  1. Point 1. Link the research questions (97-101), the presentation of the results in studies 1 (174-176) and 2 (239-240), discussion and the conclusions in the same logical order and sequence. For example, right now the aim of the study 1 (174-176), presented in results part differ from the reasearch questions (97-101) in the end of the introduction.

Response 1: Thanks to the reviewer1s suggestion, necessary corrections were made at the beginning and at the end of the article. Research questions have been refined and reformulated. Now Research questions are: What is the specificity of attitudes towards parents among middle-aged women included in the daily care of their parents? What is the role of the subjective factor (their attachment style, the feeling of guilt to parents, the style of separation from parents) in their psycho-emotional health and well-being? The logic of the presentation of the results is brought in accordance with the questions asked.

  1. Point 2. To describe the research methodologies in more detail (144-154): to present the number of items, the nature of their evaluation (Likert type or other...), reliability coefficients (Cronbach α or other coefficient).

Response 2: We are grateful to the reviewer for an important comment that helped bring methods description (2. 2) to the required standards. We`ve described the techniques in more detail, indicating number of items, a Likert scale, reliability coefficient (Cronbach alpha),  evaluation examples in Russian samples. 

  1. Point 3. In the discussion section, analyze the results at the level of phenomena by discarding the presented p (299-304 and so on)

Response 3: In the discussion section, p values ​​have been excluded and the results are presented at the level of phenomena, focusing on main ideas.

  1. Point 4. Let the conclusions correspond to the questions raised and answer them in the results section, or let there be more research questions.

Response 4: Once edited, the conclusions are consistent with the research questions (see section 5).

  1. Point 5. In Point 5the text it was used hypothesis term, but this research raised questions, not hypothesis (221, 256). So you need to eliminate the usage of hypothesis or formulate hypothesis in introduction part.

Response 5: The Text has been re-edited, the term hypothesis  omitted.

  1. Point 6. Correlation didn‘t indicate influences (222-223) 

Response 6: It is true, the term "influence" has been replaced by “prediction”. In fact the use of simple linear regression analysis allowed to draw appropriate conclusions.

Submission Date

03 November 2020

Date of this review

01 Dec 2020 19:03:48

Response to Reviewer 4 Comments

However, in order to better understand and disseminate the study, it should be improved according these points:

  1. Point 1. Link the research questions (97-101), the presentation of the results in studies 1 (174-176) and 2 (239-240), discussion and the conclusions in the same logical order and sequence. For example, right now the aim of the study 1 (174-176), presented in results part differ from the reasearch questions (97-101) in the end of the introduction.

Response 1: Thanks to the reviewer1s suggestion, necessary corrections were made at the beginning and at the end of the article. Research questions have been refined and reformulated. Now Research questions are: What is the specificity of attitudes towards parents among middle-aged women included in the daily care of their parents? What is the role of the subjective factor (their attachment style, the feeling of guilt to parents, the style of separation from parents) in their psycho-emotional health and well-being? The logic of the presentation of the results is brought in accordance with the questions asked.

  1. Point 2. To describe the research methodologies in more detail (144-154): to present the number of items, the nature of their evaluation (Likert type or other...), reliability coefficients (Cronbach α or other coefficient).

Response 2: We are grateful to the reviewer for an important comment that helped bring methods description (2. 2) to the required standards. We`ve described the techniques in more detail, indicating number of items, a Likert scale, reliability coefficient (Cronbach alpha),  evaluation examples in Russian samples. 

  1. Point 3. In the discussion section, analyze the results at the level of phenomena by discarding the presented p (299-304 and so on)

Response 3: In the discussion section, p values ​​have been excluded and the results are presented at the level of phenomena, focusing on main ideas.

  1. Point 4. Let the conclusions correspond to the questions raised and answer them in the results section, or let there be more research questions.

Response 4: Once edited, the conclusions are consistent with the research questions (see section 5).

  1. Point 5. In Point 5the text it was used hypothesis term, but this research raised questions, not hypothesis (221, 256). So you need to eliminate the usage of hypothesis or formulate hypothesis in introduction part.

Response 5: The Text has been re-edited, the term hypothesis  omitted.

  1. Point 6. Correlation didn‘t indicate influences (222-223) 

Response 6: It is true, the term "influence" has been replaced by “prediction”. In fact the use of simple linear regression analysis allowed to draw appropriate conclusions.

Submission Date

03 November 2020

Date of this review

01 Dec 2020 19:03:48

Response to Reviewer 4 Comments

However, in order to better understand and disseminate the study, it should be improved according these points:

  1. Point 1. Link the research questions (97-101), the presentation of the results in studies 1 (174-176) and 2 (239-240), discussion and the conclusions in the same logical order and sequence. For example, right now the aim of the study 1 (174-176), presented in results part differ from the reasearch questions (97-101) in the end of the introduction.

Response 1: Thanks to the reviewer1s suggestion, necessary corrections were made at the beginning and at the end of the article. Research questions have been refined and reformulated. Now Research questions are: What is the specificity of attitudes towards parents among middle-aged women included in the daily care of their parents? What is the role of the subjective factor (their attachment style, the feeling of guilt to parents, the style of separation from parents) in their psycho-emotional health and well-being? The logic of the presentation of the results is brought in accordance with the questions asked.

  1. Point 2. To describe the research methodologies in more detail (144-154): to present the number of items, the nature of their evaluation (Likert type or other...), reliability coefficients (Cronbach α or other coefficient).

Response 2: We are grateful to the reviewer for an important comment that helped bring methods description (2. 2) to the required standards. We`ve described the techniques in more detail, indicating number of items, a Likert scale, reliability coefficient (Cronbach alpha),  evaluation examples in Russian samples. 

  1. Point 3. In the discussion section, analyze the results at the level of phenomena by discarding the presented p (299-304 and so on)

Response 3: In the discussion section, p values ​​have been excluded and the results are presented at the level of phenomena, focusing on main ideas.

  1. Point 4. Let the conclusions correspond to the questions raised and answer them in the results section, or let there be more research questions.

Response 4: Once edited, the conclusions are consistent with the research questions (see section 5).

  1. Point 5. In Point 5the text it was used hypothesis term, but this research raised questions, not hypothesis (221, 256). So you need to eliminate the usage of hypothesis or formulate hypothesis in introduction part.

Response 5: The Text has been re-edited, the term hypothesis  omitted.

  1. Point 6. Correlation didn‘t indicate influences (222-223) 

Response 6: It is true, the term "influence" has been replaced by “prediction”. In fact the use of simple linear regression analysis allowed to draw appropriate conclusions.

Submission Date

03 November 2020

Date of this review

01 Dec 2020 19:03:48

Response to Reviewer 4 Comments

However, in order to better understand and disseminate the study, it should be improved according these points:

  1. Point 1. Link the research questions (97-101), the presentation of the results in studies 1 (174-176) and 2 (239-240), discussion and the conclusions in the same logical order and sequence. For example, right now the aim of the study 1 (174-176), presented in results part differ from the reasearch questions (97-101) in the end of the introduction.

Response 1: Thanks to the reviewer1s suggestion, necessary corrections were made at the beginning and at the end of the article. Research questions have been refined and reformulated. Now Research questions are: What is the specificity of attitudes towards parents among middle-aged women included in the daily care of their parents? What is the role of the subjective factor (their attachment style, the feeling of guilt to parents, the style of separation from parents) in their psycho-emotional health and well-being? The logic of the presentation of the results is brought in accordance with the questions asked.

  1. Point 2. To describe the research methodologies in more detail (144-154): to present the number of items, the nature of their evaluation (Likert type or other...), reliability coefficients (Cronbach α or other coefficient).

Response 2: We are grateful to the reviewer for an important comment that helped bring methods description (2. 2) to the required standards. We`ve described the techniques in more detail, indicating number of items, a Likert scale, reliability coefficient (Cronbach alpha),  evaluation examples in Russian samples. 

  1. Point 3. In the discussion section, analyze the results at the level of phenomena by discarding the presented p (299-304 and so on)

Response 3: In the discussion section, p values ​​have been excluded and the results are presented at the level of phenomena, focusing on main ideas.

  1. Point 4. Let the conclusions correspond to the questions raised and answer them in the results section, or let there be more research questions.

Response 4: Once edited, the conclusions are consistent with the research questions (see section 5).

  1. Point 5. In Point 5the text it was used hypothesis term, but this research raised questions, not hypothesis (221, 256). So you need to eliminate the usage of hypothesis or formulate hypothesis in introduction part.

Response 5: The Text has been re-edited, the term hypothesis  omitted.

  1. Point 6. Correlation didn‘t indicate influences (222-223) 

Response 6: It is true, the term "influence" has been replaced by “prediction”. In fact the use of simple linear regression analysis allowed to draw appropriate conclusions.

Submission Date

03 November 2020

Date of this review

01 Dec 2020 19:03:48

Response to Reviewer 4 Comments

However, in order to better understand and disseminate the study, it should be improved according these points:

  1. Point 1. Link the research questions (97-101), the presentation of the results in studies 1 (174-176) and 2 (239-240), discussion and the conclusions in the same logical order and sequence. For example, right now the aim of the study 1 (174-176), presented in results part differ from the reasearch questions (97-101) in the end of the introduction.

Response 1: Thanks to the reviewer1s suggestion, necessary corrections were made at the beginning and at the end of the article. Research questions have been refined and reformulated. Now Research questions are: What is the specificity of attitudes towards parents among middle-aged women included in the daily care of their parents? What is the role of the subjective factor (their attachment style, the feeling of guilt to parents, the style of separation from parents) in their psycho-emotional health and well-being? The logic of the presentation of the results is brought in accordance with the questions asked.

  1. Point 2. To describe the research methodologies in more detail (144-154): to present the number of items, the nature of their evaluation (Likert type or other...), reliability coefficients (Cronbach α or other coefficient).

Response 2: We are grateful to the reviewer for an important comment that helped bring methods description (2. 2) to the required standards. We`ve described the techniques in more detail, indicating number of items, a Likert scale, reliability coefficient (Cronbach alpha),  evaluation examples in Russian samples. 

  1. Point 3. In the discussion section, analyze the results at the level of phenomena by discarding the presented p (299-304 and so on)

Response 3: In the discussion section, p values ​​have been excluded and the results are presented at the level of phenomena, focusing on main ideas.

  1. Point 4. Let the conclusions correspond to the questions raised and answer them in the results section, or let there be more research questions.

Response 4: Once edited, the conclusions are consistent with the research questions (see section 5).

  1. Point 5. In Point 5the text it was used hypothesis term, but this research raised questions, not hypothesis (221, 256). So you need to eliminate the usage of hypothesis or formulate hypothesis in introduction part.

Response 5: The Text has been re-edited, the term hypothesis  omitted.

  1. Point 6. Correlation didn‘t indicate influences (222-223) 

Response 6: It is true, the term "influence" has been replaced by “prediction”. In fact the use of simple linear regression analysis allowed to draw appropriate conclusions.

Submission Date

03 November 2020

Date of this review

01 Dec 2020 19:03:48

Response to Reviewer 4 Comments

However, in order to better understand and disseminate the study, it should be improved according these points:

  1. Point 1. Link the research questions (97-101), the presentation of the results in studies 1 (174-176) and 2 (239-240), discussion and the conclusions in the same logical order and sequence. For example, right now the aim of the study 1 (174-176), presented in results part differ from the reasearch questions (97-101) in the end of the introduction.

Response 1: Thanks to the reviewer1s suggestion, necessary corrections were made at the beginning and at the end of the article. Research questions have been refined and reformulated. Now Research questions are: What is the specificity of attitudes towards parents among middle-aged women included in the daily care of their parents? What is the role of the subjective factor (their attachment style, the feeling of guilt to parents, the style of separation from parents) in their psycho-emotional health and well-being? The logic of the presentation of the results is brought in accordance with the questions asked.

  1. Point 2. To describe the research methodologies in more detail (144-154): to present the number of items, the nature of their evaluation (Likert type or other...), reliability coefficients (Cronbach α or other coefficient).

Response 2: We are grateful to the reviewer for an important comment that helped bring methods description (2. 2) to the required standards. We`ve described the techniques in more detail, indicating number of items, a Likert scale, reliability coefficient (Cronbach alpha),  evaluation examples in Russian samples. 

  1. Point 3. In the discussion section, analyze the results at the level of phenomena by discarding the presented p (299-304 and so on)

Response 3: In the discussion section, p values ​​have been excluded and the results are presented at the level of phenomena, focusing on main ideas.

  1. Point 4. Let the conclusions correspond to the questions raised and answer them in the results section, or let there be more research questions.

Response 4: Once edited, the conclusions are consistent with the research questions (see section 5).

  1. Point 5. In Point 5the text it was used hypothesis term, but this research raised questions, not hypothesis (221, 256). So you need to eliminate the usage of hypothesis or formulate hypothesis in introduction part.

Response 5: The Text has been re-edited, the term hypothesis  omitted.

  1. Point 6. Correlation didn‘t indicate influences (222-223) 

Response 6: It is true, the term "influence" has been replaced by “prediction”. In fact the use of simple linear regression analysis allowed to draw appropriate conclusions.

Submission Date

03 November 2020

Date of this review

01 Dec 2020 19:03:48

Response to Reviewer 4 Comments

However, in order to better understand and disseminate the study, it should be improved according these points:

  1. Point 1. Link the research questions (97-101), the presentation of the results in studies 1 (174-176) and 2 (239-240), discussion and the conclusions in the same logical order and sequence. For example, right now the aim of the study 1 (174-176), presented in results part differ from the reasearch questions (97-101) in the end of the introduction.

Response 1: Thanks to the reviewer1s suggestion, necessary corrections were made at the beginning and at the end of the article. Research questions have been refined and reformulated. Now Research questions are: What is the specificity of attitudes towards parents among middle-aged women included in the daily care of their parents? What is the role of the subjective factor (their attachment style, the feeling of guilt to parents, the style of separation from parents) in their psycho-emotional health and well-being? The logic of the presentation of the results is brought in accordance with the questions asked.

  1. Point 2. To describe the research methodologies in more detail (144-154): to present the number of items, the nature of their evaluation (Likert type or other...), reliability coefficients (Cronbach α or other coefficient).

Response 2: We are grateful to the reviewer for an important comment that helped bring methods description (2. 2) to the required standards. We`ve described the techniques in more detail, indicating number of items, a Likert scale, reliability coefficient (Cronbach alpha),  evaluation examples in Russian samples. 

  1. Point 3. In the discussion section, analyze the results at the level of phenomena by discarding the presented p (299-304 and so on)

Response 3: In the discussion section, p values ​​have been excluded and the results are presented at the level of phenomena, focusing on main ideas.

  1. Point 4. Let the conclusions correspond to the questions raised and answer them in the results section, or let there be more research questions.

Response 4: Once edited, the conclusions are consistent with the research questions (see section 5).

  1. Point 5. In Point 5the text it was used hypothesis term, but this research raised questions, not hypothesis (221, 256). So you need to eliminate the usage of hypothesis or formulate hypothesis in introduction part.

Response 5: The Text has been re-edited, the term hypothesis  omitted.

  1. Point 6. Correlation didn‘t indicate influences (222-223) 

Response 6: It is true, the term "influence" has been replaced by “prediction”. In fact the use of simple linear regression analysis allowed to draw appropriate conclusions.

Submission Date

03 November 2020

Date of this review

01 Dec 2020 19:03:48

Response to Reviewer 4 Comments

However, in order to better understand and disseminate the study, it should be improved according these points:

  1. Point 1. Link the research questions (97-101), the presentation of the results in studies 1 (174-176) and 2 (239-240), discussion and the conclusions in the same logical order and sequence. For example, right now the aim of the study 1 (174-176), presented in results part differ from the reasearch questions (97-101) in the end of the introduction.

Response 1: Thanks to the reviewer1s suggestion, necessary corrections were made at the beginning and at the end of the article. Research questions have been refined and reformulated. Now Research questions are: What is the specificity of attitudes towards parents among middle-aged women included in the daily care of their parents? What is the role of the subjective factor (their attachment style, the feeling of guilt to parents, the style of separation from parents) in their psycho-emotional health and well-being? The logic of the presentation of the results is brought in accordance with the questions asked.

  1. Point 2. To describe the research methodologies in more detail (144-154): to present the number of items, the nature of their evaluation (Likert type or other...), reliability coefficients (Cronbach α or other coefficient).

Response 2: We are grateful to the reviewer for an important comment that helped bring methods description (2. 2) to the required standards. We`ve described the techniques in more detail, indicating number of items, a Likert scale, reliability coefficient (Cronbach alpha),  evaluation examples in Russian samples. 

  1. Point 3. In the discussion section, analyze the results at the level of phenomena by discarding the presented p (299-304 and so on)

Response 3: In the discussion section, p values ​​have been excluded and the results are presented at the level of phenomena, focusing on main ideas.

  1. Point 4. Let the conclusions correspond to the questions raised and answer them in the results section, or let there be more research questions.

Response 4: Once edited, the conclusions are consistent with the research questions (see section 5).

  1. Point 5. In Point 5the text it was used hypothesis term, but this research raised questions, not hypothesis (221, 256). So you need to eliminate the usage of hypothesis or formulate hypothesis in introduction part.

Response 5: The Text has been re-edited, the term hypothesis  omitted.

  1. Point 6. Correlation didn‘t indicate influences (222-223) 

Response 6: It is true, the term "influence" has been replaced by “prediction”. In fact the use of simple linear regression analysis allowed to draw appropriate conclusions.

Submission Date

03 November 2020

Date of this review

01 Dec 2020 19:03:48

Response to Reviewer 4 Comments

However, in order to better understand and disseminate the study, it should be improved according these points:

  1. Point 1. Link the research questions (97-101), the presentation of the results in studies 1 (174-176) and 2 (239-240), discussion and the conclusions in the same logical order and sequence. For example, right now the aim of the study 1 (174-176), presented in results part differ from the reasearch questions (97-101) in the end of the introduction.

Response 1: Thanks to the reviewer1s suggestion, necessary corrections were made at the beginning and at the end of the article. Research questions have been refined and reformulated. Now Research questions are: What is the specificity of attitudes towards parents among middle-aged women included in the daily care of their parents? What is the role of the subjective factor (their attachment style, the feeling of guilt to parents, the style of separation from parents) in their psycho-emotional health and well-being? The logic of the presentation of the results is brought in accordance with the questions asked.

  1. Point 2. To describe the research methodologies in more detail (144-154): to present the number of items, the nature of their evaluation (Likert type or other...), reliability coefficients (Cronbach α or other coefficient).

Response 2: We are grateful to the reviewer for an important comment that helped bring methods description (2. 2) to the required standards. We`ve described the techniques in more detail, indicating number of items, a Likert scale, reliability coefficient (Cronbach alpha),  evaluation examples in Russian samples. 

  1. Point 3. In the discussion section, analyze the results at the level of phenomena by discarding the presented p (299-304 and so on)

Response 3: In the discussion section, p values ​​have been excluded and the results are presented at the level of phenomena, focusing on main ideas.

  1. Point 4. Let the conclusions correspond to the questions raised and answer them in the results section, or let there be more research questions.

Response 4: Once edited, the conclusions are consistent with the research questions (see section 5).

  1. Point 5. In Point 5the text it was used hypothesis term, but this research raised questions, not hypothesis (221, 256). So you need to eliminate the usage of hypothesis or formulate hypothesis in introduction part.

Response 5: The Text has been re-edited, the term hypothesis  omitted.

  1. Point 6. Correlation didn‘t indicate influences (222-223) 

Response 6: It is true, the term "influence" has been replaced by “prediction”. In fact the use of simple linear regression analysis allowed to draw appropriate conclusions.

Submission Date

03 November 2020

Date of this review

01 Dec 2020 19:03:48

Response to Reviewer 4 Comments

However, in order to better understand and disseminate the study, it should be improved according these points:

  1. Point 1. Link the research questions (97-101), the presentation of the results in studies 1 (174-176) and 2 (239-240), discussion and the conclusions in the same logical order and sequence. For example, right now the aim of the study 1 (174-176), presented in results part differ from the reasearch questions (97-101) in the end of the introduction.

Response 1: Thanks to the reviewer1s suggestion, necessary corrections were made at the beginning and at the end of the article. Research questions have been refined and reformulated. Now Research questions are: What is the specificity of attitudes towards parents among middle-aged women included in the daily care of their parents? What is the role of the subjective factor (their attachment style, the feeling of guilt to parents, the style of separation from parents) in their psycho-emotional health and well-being? The logic of the presentation of the results is brought in accordance with the questions asked.

  1. Point 2. To describe the research methodologies in more detail (144-154): to present the number of items, the nature of their evaluation (Likert type or other...), reliability coefficients (Cronbach α or other coefficient).

Response 2: We are grateful to the reviewer for an important comment that helped bring methods description (2. 2) to the required standards. We`ve described the techniques in more detail, indicating number of items, a Likert scale, reliability coefficient (Cronbach alpha),  evaluation examples in Russian samples. 

  1. Point 3. In the discussion section, analyze the results at the level of phenomena by discarding the presented p (299-304 and so on)

Response 3: In the discussion section, p values ​​have been excluded and the results are presented at the level of phenomena, focusing on main ideas.

  1. Point 4. Let the conclusions correspond to the questions raised and answer them in the results section, or let there be more research questions.

Response 4: Once edited, the conclusions are consistent with the research questions (see section 5).

  1. Point 5. In Point 5the text it was used hypothesis term, but this research raised questions, not hypothesis (221, 256). So you need to eliminate the usage of hypothesis or formulate hypothesis in introduction part.

Response 5: The Text has been re-edited, the term hypothesis  omitted.

  1. Point 6. Correlation didn‘t indicate influences (222-223) 

Response 6: It is true, the term "influence" has been replaced by “prediction”. In fact the use of simple linear regression analysis allowed to draw appropriate conclusions.

Submission Date

03 November 2020

Date of this review

01 Dec 2020 19:03:48

Response to Reviewer 4 Comments

However, in order to better understand and disseminate the study, it should be improved according these points:

  1. Point 1. Link the research questions (97-101), the presentation of the results in studies 1 (174-176) and 2 (239-240), discussion and the conclusions in the same logical order and sequence. For example, right now the aim of the study 1 (174-176), presented in results part differ from the reasearch questions (97-101) in the end of the introduction.

Response 1: Thanks to the reviewer1s suggestion, necessary corrections were made at the beginning and at the end of the article. Research questions have been refined and reformulated. Now Research questions are: What is the specificity of attitudes towards parents among middle-aged women included in the daily care of their parents? What is the role of the subjective factor (their attachment style, the feeling of guilt to parents, the style of separation from parents) in their psycho-emotional health and well-being? The logic of the presentation of the results is brought in accordance with the questions asked.

  1. Point 2. To describe the research methodologies in more detail (144-154): to present the number of items, the nature of their evaluation (Likert type or other...), reliability coefficients (Cronbach α or other coefficient).

Response 2: We are grateful to the reviewer for an important comment that helped bring methods description (2. 2) to the required standards. We`ve described the techniques in more detail, indicating number of items, a Likert scale, reliability coefficient (Cronbach alpha),  evaluation examples in Russian samples. 

  1. Point 3. In the discussion section, analyze the results at the level of phenomena by discarding the presented p (299-304 and so on)

Response 3: In the discussion section, p values ​​have been excluded and the results are presented at the level of phenomena, focusing on main ideas.

  1. Point 4. Let the conclusions correspond to the questions raised and answer them in the results section, or let there be more research questions.

Response 4: Once edited, the conclusions are consistent with the research questions (see section 5).

  1. Point 5. In Point 5the text it was used hypothesis term, but this research raised questions, not hypothesis (221, 256). So you need to eliminate the usage of hypothesis or formulate hypothesis in introduction part.

Response 5: The Text has been re-edited, the term hypothesis  omitted.

  1. Point 6. Correlation didn‘t indicate influences (222-223) 

Response 6: It is true, the term "influence" has been replaced by “prediction”. In fact the use of simple linear regression analysis allowed to draw appropriate conclusions.

Submission Date

03 November 2020

Date of this review

01 Dec 2020 19:03:48

Response to Reviewer 4 Comments

However, in order to better understand and disseminate the study, it should be improved according these points:

  1. Point 1. Link the research questions (97-101), the presentation of the results in studies 1 (174-176) and 2 (239-240), discussion and the conclusions in the same logical order and sequence. For example, right now the aim of the study 1 (174-176), presented in results part differ from the reasearch questions (97-101) in the end of the introduction.

Response 1: Thanks to the reviewer1s suggestion, necessary corrections were made at the beginning and at the end of the article. Research questions have been refined and reformulated. Now Research questions are: What is the specificity of attitudes towards parents among middle-aged women included in the daily care of their parents? What is the role of the subjective factor (their attachment style, the feeling of guilt to parents, the style of separation from parents) in their psycho-emotional health and well-being? The logic of the presentation of the results is brought in accordance with the questions asked.

  1. Point 2. To describe the research methodologies in more detail (144-154): to present the number of items, the nature of their evaluation (Likert type or other...), reliability coefficients (Cronbach α or other coefficient).

Response 2: We are grateful to the reviewer for an important comment that helped bring methods description (2. 2) to the required standards. We`ve described the techniques in more detail, indicating number of items, a Likert scale, reliability coefficient (Cronbach alpha),  evaluation examples in Russian samples. 

  1. Point 3. In the discussion section, analyze the results at the level of phenomena by discarding the presented p (299-304 and so on)

Response 3: In the discussion section, p values ​​have been excluded and the results are presented at the level of phenomena, focusing on main ideas.

  1. Point 4. Let the conclusions correspond to the questions raised and answer them in the results section, or let there be more research questions.

Response 4: Once edited, the conclusions are consistent with the research questions (see section 5).

  1. Point 5. In Point 5the text it was used hypothesis term, but this research raised questions, not hypothesis (221, 256). So you need to eliminate the usage of hypothesis or formulate hypothesis in introduction part.

Response 5: The Text has been re-edited, the term hypothesis  omitted.

  1. Point 6. Correlation didn‘t indicate influences (222-223) 

Response 6: It is true, the term "influence" has been replaced by “prediction”. In fact the use of simple linear regression analysis allowed to draw appropriate conclusions.

Submission Date

03 November 2020

Date of this review

01 Dec 2020 19:03:48

Response to Reviewer 4 Comments

However, in order to better understand and disseminate the study, it should be improved according these points:

  1. Point 1. Link the research questions (97-101), the presentation of the results in studies 1 (174-176) and 2 (239-240), discussion and the conclusions in the same logical order and sequence. For example, right now the aim of the study 1 (174-176), presented in results part differ from the reasearch questions (97-101) in the end of the introduction.

Response 1: Thanks to the reviewer1s suggestion, necessary corrections were made at the beginning and at the end of the article. Research questions have been refined and reformulated. Now Research questions are: What is the specificity of attitudes towards parents among middle-aged women included in the daily care of their parents? What is the role of the subjective factor (their attachment style, the feeling of guilt to parents, the style of separation from parents) in their psycho-emotional health and well-being? The logic of the presentation of the results is brought in accordance with the questions asked.

  1. Point 2. To describe the research methodologies in more detail (144-154): to present the number of items, the nature of their evaluation (Likert type or other...), reliability coefficients (Cronbach α or other coefficient).

Response 2: We are grateful to the reviewer for an important comment that helped bring methods description (2. 2) to the required standards. We`ve described the techniques in more detail, indicating number of items, a Likert scale, reliability coefficient (Cronbach alpha),  evaluation examples in Russian samples. 

  1. Point 3. In the discussion section, analyze the results at the level of phenomena by discarding the presented p (299-304 and so on)

Response 3: In the discussion section, p values ​​have been excluded and the results are presented at the level of phenomena, focusing on main ideas.

  1. Point 4. Let the conclusions correspond to the questions raised and answer them in the results section, or let there be more research questions.

Response 4: Once edited, the conclusions are consistent with the research questions (see section 5).

  1. Point 5. In Point 5the text it was used hypothesis term, but this research raised questions, not hypothesis (221, 256). So you need to eliminate the usage of hypothesis or formulate hypothesis in introduction part.

Response 5: The Text has been re-edited, the term hypothesis  omitted.

  1. Point 6. Correlation didn‘t indicate influences (222-223) 

Response 6: It is true, the term "influence" has been replaced by “prediction”. In fact the use of simple linear regression analysis allowed to draw appropriate conclusions.

Submission Date

03 November 2020

Date of this review

01 Dec 2020 19:03:48

Response to Reviewer 4 Comments

However, in order to better understand and disseminate the study, it should be improved according these points:

  1. Point 1. Link the research questions (97-101), the presentation of the results in studies 1 (174-176) and 2 (239-240), discussion and the conclusions in the same logical order and sequence. For example, right now the aim of the study 1 (174-176), presented in results part differ from the reasearch questions (97-101) in the end of the introduction.

Response 1: Thanks to the reviewer1s suggestion, necessary corrections were made at the beginning and at the end of the article. Research questions have been refined and reformulated. Now Research questions are: What is the specificity of attitudes towards parents among middle-aged women included in the daily care of their parents? What is the role of the subjective factor (their attachment style, the feeling of guilt to parents, the style of separation from parents) in their psycho-emotional health and well-being? The logic of the presentation of the results is brought in accordance with the questions asked.

  1. Point 2. To describe the research methodologies in more detail (144-154): to present the number of items, the nature of their evaluation (Likert type or other...), reliability coefficients (Cronbach α or other coefficient).

Response 2: We are grateful to the reviewer for an important comment that helped bring methods description (2. 2) to the required standards. We`ve described the techniques in more detail, indicating number of items, a Likert scale, reliability coefficient (Cronbach alpha),  evaluation examples in Russian samples. 

  1. Point 3. In the discussion section, analyze the results at the level of phenomena by discarding the presented p (299-304 and so on)

Response 3: In the discussion section, p values ​​have been excluded and the results are presented at the level of phenomena, focusing on main ideas.

  1. Point 4. Let the conclusions correspond to the questions raised and answer them in the results section, or let there be more research questions.

Response 4: Once edited, the conclusions are consistent with the research questions (see section 5).

  1. Point 5. In Point 5the text it was used hypothesis term, but this research raised questions, not hypothesis (221, 256). So you need to eliminate the usage of hypothesis or formulate hypothesis in introduction part.

Response 5: The Text has been re-edited, the term hypothesis  omitted.

  1. Point 6. Correlation didn‘t indicate influences (222-223) 

Response 6: It is true, the term "influence" has been replaced by “prediction”. In fact the use of simple linear regression analysis allowed to draw appropriate conclusions.

Submission Date

03 November 2020

Date of this review

01 Dec 2020 19:03:48

Response to Reviewer 4 Comments

However, in order to better understand and disseminate the study, it should be improved according these points:

  1. Point 1. Link the research questions (97-101), the presentation of the results in studies 1 (174-176) and 2 (239-240), discussion and the conclusions in the same logical order and sequence. For example, right now the aim of the study 1 (174-176), presented in results part differ from the reasearch questions (97-101) in the end of the introduction.

Response 1: Thanks to the reviewer1s suggestion, necessary corrections were made at the beginning and at the end of the article. Research questions have been refined and reformulated. Now Research questions are: What is the specificity of attitudes towards parents among middle-aged women included in the daily care of their parents? What is the role of the subjective factor (their attachment style, the feeling of guilt to parents, the style of separation from parents) in their psycho-emotional health and well-being? The logic of the presentation of the results is brought in accordance with the questions asked.

  1. Point 2. To describe the research methodologies in more detail (144-154): to present the number of items, the nature of their evaluation (Likert type or other...), reliability coefficients (Cronbach α or other coefficient).

Response 2: We are grateful to the reviewer for an important comment that helped bring methods description (2. 2) to the required standards. We`ve described the techniques in more detail, indicating number of items, a Likert scale, reliability coefficient (Cronbach alpha),  evaluation examples in Russian samples. 

  1. Point 3. In the discussion section, analyze the results at the level of phenomena by discarding the presented p (299-304 and so on)

Response 3: In the discussion section, p values ​​have been excluded and the results are presented at the level of phenomena, focusing on main ideas.

  1. Point 4. Let the conclusions correspond to the questions raised and answer them in the results section, or let there be more research questions.

Response 4: Once edited, the conclusions are consistent with the research questions (see section 5).

  1. Point 5. In Point 5the text it was used hypothesis term, but this research raised questions, not hypothesis (221, 256). So you need to eliminate the usage of hypothesis or formulate hypothesis in introduction part.

Response 5: The Text has been re-edited, the term hypothesis  omitted.

  1. Point 6. Correlation didn‘t indicate influences (222-223) 

Response 6: It is true, the term "influence" has been replaced by “prediction”. In fact the use of simple linear regression analysis allowed to draw appropriate conclusions.

Submission Date

03 November 2020

Date of this review

01 Dec 2020 19:03:48

Response to Reviewer 4 Comments

However, in order to better understand and disseminate the study, it should be improved according these points:

  1. Point 1. Link the research questions (97-101), the presentation of the results in studies 1 (174-176) and 2 (239-240), discussion and the conclusions in the same logical order and sequence. For example, right now the aim of the study 1 (174-176), presented in results part differ from the reasearch questions (97-101) in the end of the introduction.

Response 1: Thanks to the reviewer1s suggestion, necessary corrections were made at the beginning and at the end of the article. Research questions have been refined and reformulated. Now Research questions are: What is the specificity of attitudes towards parents among middle-aged women included in the daily care of their parents? What is the role of the subjective factor (their attachment style, the feeling of guilt to parents, the style of separation from parents) in their psycho-emotional health and well-being? The logic of the presentation of the results is brought in accordance with the questions asked.

  1. Point 2. To describe the research methodologies in more detail (144-154): to present the number of items, the nature of their evaluation (Likert type or other...), reliability coefficients (Cronbach α or other coefficient).

Response 2: We are grateful to the reviewer for an important comment that helped bring methods description (2. 2) to the required standards. We`ve described the techniques in more detail, indicating number of items, a Likert scale, reliability coefficient (Cronbach alpha),  evaluation examples in Russian samples. 

  1. Point 3. In the discussion section, analyze the results at the level of phenomena by discarding the presented p (299-304 and so on)

Response 3: In the discussion section, p values ​​have been excluded and the results are presented at the level of phenomena, focusing on main ideas.

  1. Point 4. Let the conclusions correspond to the questions raised and answer them in the results section, or let there be more research questions.

Response 4: Once edited, the conclusions are consistent with the research questions (see section 5).

  1. Point 5. In Point 5the text it was used hypothesis term, but this research raised questions, not hypothesis (221, 256). So you need to eliminate the usage of hypothesis or formulate hypothesis in introduction part.

Response 5: The Text has been re-edited, the term hypothesis  omitted.

  1. Point 6. Correlation didn‘t indicate influences (222-223) 

Response 6: It is true, the term "influence" has been replaced by “prediction”. In fact the use of simple linear regression analysis allowed to draw appropriate conclusions.

Submission Date

03 November 2020

Date of this review

01 Dec 2020 19:03:48

Response to Reviewer 4 Comments

However, in order to better understand and disseminate the study, it should be improved according these points:

  1. Point 1. Link the research questions (97-101), the presentation of the results in studies 1 (174-176) and 2 (239-240), discussion and the conclusions in the same logical order and sequence. For example, right now the aim of the study 1 (174-176), presented in results part differ from the reasearch questions (97-101) in the end of the introduction.

Response 1: Thanks to the reviewer1s suggestion, necessary corrections were made at the beginning and at the end of the article. Research questions have been refined and reformulated. Now Research questions are: What is the specificity of attitudes towards parents among middle-aged women included in the daily care of their parents? What is the role of the subjective factor (their attachment style, the feeling of guilt to parents, the style of separation from parents) in their psycho-emotional health and well-being? The logic of the presentation of the results is brought in accordance with the questions asked.

  1. Point 2. To describe the research methodologies in more detail (144-154): to present the number of items, the nature of their evaluation (Likert type or other...), reliability coefficients (Cronbach α or other coefficient).

Response 2: We are grateful to the reviewer for an important comment that helped bring methods description (2. 2) to the required standards. We`ve described the techniques in more detail, indicating number of items, a Likert scale, reliability coefficient (Cronbach alpha),  evaluation examples in Russian samples. 

  1. Point 3. In the discussion section, analyze the results at the level of phenomena by discarding the presented p (299-304 and so on)

Response 3: In the discussion section, p values ​​have been excluded and the results are presented at the level of phenomena, focusing on main ideas.

  1. Point 4. Let the conclusions correspond to the questions raised and answer them in the results section, or let there be more research questions.

Response 4: Once edited, the conclusions are consistent with the research questions (see section 5).

  1. Point 5. In Point 5the text it was used hypothesis term, but this research raised questions, not hypothesis (221, 256). So you need to eliminate the usage of hypothesis or formulate hypothesis in introduction part.

Response 5: The Text has been re-edited, the term hypothesis  omitted.

  1. Point 6. Correlation didn‘t indicate influences (222-223) 

Response 6: It is true, the term "influence" has been replaced by “prediction”. In fact the use of simple linear regression analysis allowed to draw appropriate conclusions.

Submission Date

03 November 2020

Date of this review

01 Dec 2020 19:03:48

Response to Reviewer 4 Comments

However, in order to better understand and disseminate the study, it should be improved according these points:

  1. Point 1. Link the research questions (97-101), the presentation of the results in studies 1 (174-176) and 2 (239-240), discussion and the conclusions in the same logical order and sequence. For example, right now the aim of the study 1 (174-176), presented in results part differ from the reasearch questions (97-101) in the end of the introduction.

Response 1: Thanks to the reviewer1s suggestion, necessary corrections were made at the beginning and at the end of the article. Research questions have been refined and reformulated. Now Research questions are: What is the specificity of attitudes towards parents among middle-aged women included in the daily care of their parents? What is the role of the subjective factor (their attachment style, the feeling of guilt to parents, the style of separation from parents) in their psycho-emotional health and well-being? The logic of the presentation of the results is brought in accordance with the questions asked.

  1. Point 2. To describe the research methodologies in more detail (144-154): to present the number of items, the nature of their evaluation (Likert type or other...), reliability coefficients (Cronbach α or other coefficient).

Response 2: We are grateful to the reviewer for an important comment that helped bring methods description (2. 2) to the required standards. We`ve described the techniques in more detail, indicating number of items, a Likert scale, reliability coefficient (Cronbach alpha),  evaluation examples in Russian samples. 

  1. Point 3. In the discussion section, analyze the results at the level of phenomena by discarding the presented p (299-304 and so on)

Response 3: In the discussion section, p values ​​have been excluded and the results are presented at the level of phenomena, focusing on main ideas.

  1. Point 4. Let the conclusions correspond to the questions raised and answer them in the results section, or let there be more research questions.

Response 4: Once edited, the conclusions are consistent with the research questions (see section 5).

  1. Point 5. In Point 5the text it was used hypothesis term, but this research raised questions, not hypothesis (221, 256). So you need to eliminate the usage of hypothesis or formulate hypothesis in introduction part.

Response 5: The Text has been re-edited, the term hypothesis  omitted.

  1. Point 6. Correlation didn‘t indicate influences (222-223) 

Response 6: It is true, the term "influence" has been replaced by “prediction”. In fact the use of simple linear regression analysis allowed to draw appropriate conclusions.

Submission Date

03 November 2020

Date of this review

01 Dec 2020 19:03:48

Response to Reviewer 4 Comments

However, in order to better understand and disseminate the study, it should be improved according these points:

  1. Point 1. Link the research questions (97-101), the presentation of the results in studies 1 (174-176) and 2 (239-240), discussion and the conclusions in the same logical order and sequence. For example, right now the aim of the study 1 (174-176), presented in results part differ from the reasearch questions (97-101) in the end of the introduction.

Response 1: Thanks to the reviewer1s suggestion, necessary corrections were made at the beginning and at the end of the article. Research questions have been refined and reformulated. Now Research questions are: What is the specificity of attitudes towards parents among middle-aged women included in the daily care of their parents? What is the role of the subjective factor (their attachment style, the feeling of guilt to parents, the style of separation from parents) in their psycho-emotional health and well-being? The logic of the presentation of the results is brought in accordance with the questions asked.

  1. Point 2. To describe the research methodologies in more detail (144-154): to present the number of items, the nature of their evaluation (Likert type or other...), reliability coefficients (Cronbach α or other coefficient).

Response 2: We are grateful to the reviewer for an important comment that helped bring methods description (2. 2) to the required standards. We`ve described the techniques in more detail, indicating number of items, a Likert scale, reliability coefficient (Cronbach alpha),  evaluation examples in Russian samples. 

  1. Point 3. In the discussion section, analyze the results at the level of phenomena by discarding the presented p (299-304 and so on)

Response 3: In the discussion section, p values ​​have been excluded and the results are presented at the level of phenomena, focusing on main ideas.

  1. Point 4. Let the conclusions correspond to the questions raised and answer them in the results section, or let there be more research questions.

Response 4: Once edited, the conclusions are consistent with the research questions (see section 5).

  1. Point 5. In Point 5the text it was used hypothesis term, but this research raised questions, not hypothesis (221, 256). So you need to eliminate the usage of hypothesis or formulate hypothesis in introduction part.

Response 5: The Text has been re-edited, the term hypothesis  omitted.

  1. Point 6. Correlation didn‘t indicate influences (222-223) 

Response 6: It is true, the term "influence" has been replaced by “prediction”. In fact the use of simple linear regression analysis allowed to draw appropriate conclusions.

Submission Date

03 November 2020

Date of this review

01 Dec 2020 19:03:48

Response to Reviewer 4 Comments

However, in order to better understand and disseminate the study, it should be improved according these points:

  1. Point 1. Link the research questions (97-101), the presentation of the results in studies 1 (174-176) and 2 (239-240), discussion and the conclusions in the same logical order and sequence. For example, right now the aim of the study 1 (174-176), presented in results part differ from the reasearch questions (97-101) in the end of the introduction.

Response 1: Thanks to the reviewer1s suggestion, necessary corrections were made at the beginning and at the end of the article. Research questions have been refined and reformulated. Now Research questions are: What is the specificity of attitudes towards parents among middle-aged women included in the daily care of their parents? What is the role of the subjective factor (their attachment style, the feeling of guilt to parents, the style of separation from parents) in their psycho-emotional health and well-being? The logic of the presentation of the results is brought in accordance with the questions asked.

  1. Point 2. To describe the research methodologies in more detail (144-154): to present the number of items, the nature of their evaluation (Likert type or other...), reliability coefficients (Cronbach α or other coefficient).

Response 2: We are grateful to the reviewer for an important comment that helped bring methods description (2. 2) to the required standards. We`ve described the techniques in more detail, indicating number of items, a Likert scale, reliability coefficient (Cronbach alpha),  evaluation examples in Russian samples. 

  1. Point 3. In the discussion section, analyze the results at the level of phenomena by discarding the presented p (299-304 and so on)

Response 3: In the discussion section, p values ​​have been excluded and the results are presented at the level of phenomena, focusing on main ideas.

  1. Point 4. Let the conclusions correspond to the questions raised and answer them in the results section, or let there be more research questions.

Response 4: Once edited, the conclusions are consistent with the research questions (see section 5).

  1. Point 5. In Point 5the text it was used hypothesis term, but this research raised questions, not hypothesis (221, 256). So you need to eliminate the usage of hypothesis or formulate hypothesis in introduction part.

Response 5: The Text has been re-edited, the term hypothesis  omitted.

  1. Point 6. Correlation didn‘t indicate influences (222-223) 

Response 6: It is true, the term "influence" has been replaced by “prediction”. In fact the use of simple linear regression analysis allowed to draw appropriate conclusions.

Submission Date

03 November 2020

Date of this review

01 Dec 2020 19:03:48

Response to Reviewer 4 Comments

However, in order to better understand and disseminate the study, it should be improved according these points:

  1. Point 1. Link the research questions (97-101), the presentation of the results in studies 1 (174-176) and 2 (239-240), discussion and the conclusions in the same logical order and sequence. For example, right now the aim of the study 1 (174-176), presented in results part differ from the reasearch questions (97-101) in the end of the introduction.

Response 1: Thanks to the reviewer1s suggestion, necessary corrections were made at the beginning and at the end of the article. Research questions have been refined and reformulated. Now Research questions are: What is the specificity of attitudes towards parents among middle-aged women included in the daily care of their parents? What is the role of the subjective factor (their attachment style, the feeling of guilt to parents, the style of separation from parents) in their psycho-emotional health and well-being? The logic of the presentation of the results is brought in accordance with the questions asked.

  1. Point 2. To describe the research methodologies in more detail (144-154): to present the number of items, the nature of their evaluation (Likert type or other...), reliability coefficients (Cronbach α or other coefficient).

Response 2: We are grateful to the reviewer for an important comment that helped bring methods description (2. 2) to the required standards. We`ve described the techniques in more detail, indicating number of items, a Likert scale, reliability coefficient (Cronbach alpha),  evaluation examples in Russian samples. 

  1. Point 3. In the discussion section, analyze the results at the level of phenomena by discarding the presented p (299-304 and so on)

Response 3: In the discussion section, p values ​​have been excluded and the results are presented at the level of phenomena, focusing on main ideas.

  1. Point 4. Let the conclusions correspond to the questions raised and answer them in the results section, or let there be more research questions.

Response 4: Once edited, the conclusions are consistent with the research questions (see section 5).

  1. Point 5. In Point 5the text it was used hypothesis term, but this research raised questions, not hypothesis (221, 256). So you need to eliminate the usage of hypothesis or formulate hypothesis in introduction part.

Response 5: The Text has been re-edited, the term hypothesis  omitted.

  1. Point 6. Correlation didn‘t indicate influences (222-223) 

Response 6: It is true, the term "influence" has been replaced by “prediction”. In fact the use of simple linear regression analysis allowed to draw appropriate conclusions.

Submission Date

03 November 2020

Date of this review

01 Dec 2020 19:03:48

Response to Reviewer 4 Comments

However, in order to better understand and disseminate the study, it should be improved according these points:

  1. Point 1. Link the research questions (97-101), the presentation of the results in studies 1 (174-176) and 2 (239-240), discussion and the conclusions in the same logical order and sequence. For example, right now the aim of the study 1 (174-176), presented in results part differ from the reasearch questions (97-101) in the end of the introduction.

Response 1: Thanks to the reviewer1s suggestion, necessary corrections were made at the beginning and at the end of the article. Research questions have been refined and reformulated. Now Research questions are: What is the specificity of attitudes towards parents among middle-aged women included in the daily care of their parents? What is the role of the subjective factor (their attachment style, the feeling of guilt to parents, the style of separation from parents) in their psycho-emotional health and well-being? The logic of the presentation of the results is brought in accordance with the questions asked.

  1. Point 2. To describe the research methodologies in more detail (144-154): to present the number of items, the nature of their evaluation (Likert type or other...), reliability coefficients (Cronbach α or other coefficient).

Response 2: We are grateful to the reviewer for an important comment that helped bring methods description (2. 2) to the required standards. We`ve described the techniques in more detail, indicating number of items, a Likert scale, reliability coefficient (Cronbach alpha),  evaluation examples in Russian samples. 

  1. Point 3. In the discussion section, analyze the results at the level of phenomena by discarding the presented p (299-304 and so on)

Response 3: In the discussion section, p values ​​have been excluded and the results are presented at the level of phenomena, focusing on main ideas.

  1. Point 4. Let the conclusions correspond to the questions raised and answer them in the results section, or let there be more research questions.

Response 4: Once edited, the conclusions are consistent with the research questions (see section 5).

  1. Point 5. In Point 5the text it was used hypothesis term, but this research raised questions, not hypothesis (221, 256). So you need to eliminate the usage of hypothesis or formulate hypothesis in introduction part.

Response 5: The Text has been re-edited, the term hypothesis  omitted.

  1. Point 6. Correlation didn‘t indicate influences (222-223) 

Response 6: It is true, the term "influence" has been replaced by “prediction”. In fact the use of simple linear regression analysis allowed to draw appropriate conclusions.

Submission Date

03 November 2020

Date of this review

01 Dec 2020 19:03:48

Response to Reviewer 4 Comments

However, in order to better understand and disseminate the study, it should be improved according these points:

  1. Point 1. Link the research questions (97-101), the presentation of the results in studies 1 (174-176) and 2 (239-240), discussion and the conclusions in the same logical order and sequence. For example, right now the aim of the study 1 (174-176), presented in results part differ from the reasearch questions (97-101) in the end of the introduction.

Response 1: Thanks to the reviewer1s suggestion, necessary corrections were made at the beginning and at the end of the article. Research questions have been refined and reformulated. Now Research questions are: What is the specificity of attitudes towards parents among middle-aged women included in the daily care of their parents? What is the role of the subjective factor (their attachment style, the feeling of guilt to parents, the style of separation from parents) in their psycho-emotional health and well-being? The logic of the presentation of the results is brought in accordance with the questions asked.

  1. Point 2. To describe the research methodologies in more detail (144-154): to present the number of items, the nature of their evaluation (Likert type or other...), reliability coefficients (Cronbach α or other coefficient).

Response 2: We are grateful to the reviewer for an important comment that helped bring methods description (2. 2) to the required standards. We`ve described the techniques in more detail, indicating number of items, a Likert scale, reliability coefficient (Cronbach alpha),  evaluation examples in Russian samples. 

  1. Point 3. In the discussion section, analyze the results at the level of phenomena by discarding the presented p (299-304 and so on)

Response 3: In the discussion section, p values ​​have been excluded and the results are presented at the level of phenomena, focusing on main ideas.

  1. Point 4. Let the conclusions correspond to the questions raised and answer them in the results section, or let there be more research questions.

Response 4: Once edited, the conclusions are consistent with the research questions (see section 5).

  1. Point 5. In Point 5the text it was used hypothesis term, but this research raised questions, not hypothesis (221, 256). So you need to eliminate the usage of hypothesis or formulate hypothesis in introduction part.

Response 5: The Text has been re-edited, the term hypothesis  omitted.

  1. Point 6. Correlation didn‘t indicate influences (222-223) 

Response 6: It is true, the term "influence" has been replaced by “prediction”. In fact the use of simple linear regression analysis allowed to draw appropriate conclusions.

Submission Date

03 November 2020

Date of this review

01 Dec 2020 19:03:48

Response to Reviewer 4 Comments

However, in order to better understand and disseminate the study, it should be improved according these points:

  1. Point 1. Link the research questions (97-101), the presentation of the results in studies 1 (174-176) and 2 (239-240), discussion and the conclusions in the same logical order and sequence. For example, right now the aim of the study 1 (174-176), presented in results part differ from the reasearch questions (97-101) in the end of the introduction.

Response 1: Thanks to the reviewer1s suggestion, necessary corrections were made at the beginning and at the end of the article. Research questions have been refined and reformulated. Now Research questions are: What is the specificity of attitudes towards parents among middle-aged women included in the daily care of their parents? What is the role of the subjective factor (their attachment style, the feeling of guilt to parents, the style of separation from parents) in their psycho-emotional health and well-being? The logic of the presentation of the results is brought in accordance with the questions asked.

  1. Point 2. To describe the research methodologies in more detail (144-154): to present the number of items, the nature of their evaluation (Likert type or other...), reliability coefficients (Cronbach α or other coefficient).

Response 2: We are grateful to the reviewer for an important comment that helped bring methods description (2. 2) to the required standards. We`ve described the techniques in more detail, indicating number of items, a Likert scale, reliability coefficient (Cronbach alpha),  evaluation examples in Russian samples. 

  1. Point 3. In the discussion section, analyze the results at the level of phenomena by discarding the presented p (299-304 and so on)

Response 3: In the discussion section, p values ​​have been excluded and the results are presented at the level of phenomena, focusing on main ideas.

  1. Point 4. Let the conclusions correspond to the questions raised and answer them in the results section, or let there be more research questions.

Response 4: Once edited, the conclusions are consistent with the research questions (see section 5).

  1. Point 5. In Point 5the text it was used hypothesis term, but this research raised questions, not hypothesis (221, 256). So you need to eliminate the usage of hypothesis or formulate hypothesis in introduction part.

Response 5: The Text has been re-edited, the term hypothesis  omitted.

  1. Point 6. Correlation didn‘t indicate influences (222-223) 

Response 6: It is true, the term "influence" has been replaced by “prediction”. In fact the use of simple linear regression analysis allowed to draw appropriate conclusions.

Submission Date

03 November 2020

Date of this review

01 Dec 2020 19:03:48

Response to Reviewer 4 Comments

However, in order to better understand and disseminate the study, it should be improved according these points:

  1. Point 1. Link the research questions (97-101), the presentation of the results in studies 1 (174-176) and 2 (239-240), discussion and the conclusions in the same logical order and sequence. For example, right now the aim of the study 1 (174-176), presented in results part differ from the reasearch questions (97-101) in the end of the introduction.

Response 1: Thanks to the reviewer1s suggestion, necessary corrections were made at the beginning and at the end of the article. Research questions have been refined and reformulated. Now Research questions are: What is the specificity of attitudes towards parents among middle-aged women included in the daily care of their parents? What is the role of the subjective factor (their attachment style, the feeling of guilt to parents, the style of separation from parents) in their psycho-emotional health and well-being? The logic of the presentation of the results is brought in accordance with the questions asked.

  1. Point 2. To describe the research methodologies in more detail (144-154): to present the number of items, the nature of their evaluation (Likert type or other...), reliability coefficients (Cronbach α or other coefficient).

Response 2: We are grateful to the reviewer for an important comment that helped bring methods description (2. 2) to the required standards. We`ve described the techniques in more detail, indicating number of items, a Likert scale, reliability coefficient (Cronbach alpha),  evaluation examples in Russian samples. 

  1. Point 3. In the discussion section, analyze the results at the level of phenomena by discarding the presented p (299-304 and so on)

Response 3: In the discussion section, p values ​​have been excluded and the results are presented at the level of phenomena, focusing on main ideas.

  1. Point 4. Let the conclusions correspond to the questions raised and answer them in the results section, or let there be more research questions.

Response 4: Once edited, the conclusions are consistent with the research questions (see section 5).

  1. Point 5. In Point 5the text it was used hypothesis term, but this research raised questions, not hypothesis (221, 256). So you need to eliminate the usage of hypothesis or formulate hypothesis in introduction part.

Response 5: The Text has been re-edited, the term hypothesis  omitted.

  1. Point 6. Correlation didn‘t indicate influences (222-223) 

Response 6: It is true, the term "influence" has been replaced by “prediction”. In fact the use of simple linear regression analysis allowed to draw appropriate conclusions.

Submission Date

03 November 2020

Date of this review

01 Dec 2020 19:03:48

Response to Reviewer 4 Comments

However, in order to better understand and disseminate the study, it should be improved according these points:

  1. Point 1. Link the research questions (97-101), the presentation of the results in studies 1 (174-176) and 2 (239-240), discussion and the conclusions in the same logical order and sequence. For example, right now the aim of the study 1 (174-176), presented in results part differ from the reasearch questions (97-101) in the end of the introduction.

Response 1: Thanks to the reviewer1s suggestion, necessary corrections were made at the beginning and at the end of the article. Research questions have been refined and reformulated. Now Research questions are: What is the specificity of attitudes towards parents among middle-aged women included in the daily care of their parents? What is the role of the subjective factor (their attachment style, the feeling of guilt to parents, the style of separation from parents) in their psycho-emotional health and well-being? The logic of the presentation of the results is brought in accordance with the questions asked.

  1. Point 2. To describe the research methodologies in more detail (144-154): to present the number of items, the nature of their evaluation (Likert type or other...), reliability coefficients (Cronbach α or other coefficient).

Response 2: We are grateful to the reviewer for an important comment that helped bring methods description (2. 2) to the required standards. We`ve described the techniques in more detail, indicating number of items, a Likert scale, reliability coefficient (Cronbach alpha),  evaluation examples in Russian samples. 

  1. Point 3. In the discussion section, analyze the results at the level of phenomena by discarding the presented p (299-304 and so on)

Response 3: In the discussion section, p values ​​have been excluded and the results are presented at the level of phenomena, focusing on main ideas.

  1. Point 4. Let the conclusions correspond to the questions raised and answer them in the results section, or let there be more research questions.

Response 4: Once edited, the conclusions are consistent with the research questions (see section 5).

  1. Point 5. In Point 5the text it was used hypothesis term, but this research raised questions, not hypothesis (221, 256). So you need to eliminate the usage of hypothesis or formulate hypothesis in introduction part.

Response 5: The Text has been re-edited, the term hypothesis  omitted.

  1. Point 6. Correlation didn‘t indicate influences (222-223) 

Response 6: It is true, the term "influence" has been replaced by “prediction”. In fact the use of simple linear regression analysis allowed to draw appropriate conclusions.

Submission Date

03 November 2020

Date of this review

01 Dec 2020 19:03:48

Response to Reviewer 4 Comments

However, in order to better understand and disseminate the study, it should be improved according these points:

  1. Point 1. Link the research questions (97-101), the presentation of the results in studies 1 (174-176) and 2 (239-240), discussion and the conclusions in the same logical order and sequence. For example, right now the aim of the study 1 (174-176), presented in results part differ from the reasearch questions (97-101) in the end of the introduction.

Response 1: Thanks to the reviewer1s suggestion, necessary corrections were made at the beginning and at the end of the article. Research questions have been refined and reformulated. Now Research questions are: What is the specificity of attitudes towards parents among middle-aged women included in the daily care of their parents? What is the role of the subjective factor (their attachment style, the feeling of guilt to parents, the style of separation from parents) in their psycho-emotional health and well-being? The logic of the presentation of the results is brought in accordance with the questions asked.

  1. Point 2. To describe the research methodologies in more detail (144-154): to present the number of items, the nature of their evaluation (Likert type or other...), reliability coefficients (Cronbach α or other coefficient).

Response 2: We are grateful to the reviewer for an important comment that helped bring methods description (2. 2) to the required standards. We`ve described the techniques in more detail, indicating number of items, a Likert scale, reliability coefficient (Cronbach alpha),  evaluation examples in Russian samples. 

  1. Point 3. In the discussion section, analyze the results at the level of phenomena by discarding the presented p (299-304 and so on)

Response 3: In the discussion section, p values ​​have been excluded and the results are presented at the level of phenomena, focusing on main ideas.

  1. Point 4. Let the conclusions correspond to the questions raised and answer them in the results section, or let there be more research questions.

Response 4: Once edited, the conclusions are consistent with the research questions (see section 5).

  1. Point 5. In Point 5the text it was used hypothesis term, but this research raised questions, not hypothesis (221, 256). So you need to eliminate the usage of hypothesis or formulate hypothesis in introduction part.

Response 5: The Text has been re-edited, the term hypothesis  omitted.

  1. Point 6. Correlation didn‘t indicate influences (222-223) 

Response 6: It is true, the term "influence" has been replaced by “prediction”. In fact the use of simple linear regression analysis allowed to draw appropriate conclusions.

Submission Date

03 November 2020

Date of this review

01 Dec 2020 19:03:48

Response to Reviewer 4 Comments

However, in order to better understand and disseminate the study, it should be improved according these points:

  1. Point 1. Link the research questions (97-101), the presentation of the results in studies 1 (174-176) and 2 (239-240), discussion and the conclusions in the same logical order and sequence. For example, right now the aim of the study 1 (174-176), presented in results part differ from the reasearch questions (97-101) in the end of the introduction.

Response 1: Thanks to the reviewer1s suggestion, necessary corrections were made at the beginning and at the end of the article. Research questions have been refined and reformulated. Now Research questions are: What is the specificity of attitudes towards parents among middle-aged women included in the daily care of their parents? What is the role of the subjective factor (their attachment style, the feeling of guilt to parents, the style of separation from parents) in their psycho-emotional health and well-being? The logic of the presentation of the results is brought in accordance with the questions asked.

  1. Point 2. To describe the research methodologies in more detail (144-154): to present the number of items, the nature of their evaluation (Likert type or other...), reliability coefficients (Cronbach α or other coefficient).

Response 2: We are grateful to the reviewer for an important comment that helped bring methods description (2. 2) to the required standards. We`ve described the techniques in more detail, indicating number of items, a Likert scale, reliability coefficient (Cronbach alpha),  evaluation examples in Russian samples. 

  1. Point 3. In the discussion section, analyze the results at the level of phenomena by discarding the presented p (299-304 and so on)

Response 3: In the discussion section, p values ​​have been excluded and the results are presented at the level of phenomena, focusing on main ideas.

  1. Point 4. Let the conclusions correspond to the questions raised and answer them in the results section, or let there be more research questions.

Response 4: Once edited, the conclusions are consistent with the research questions (see section 5).

  1. Point 5. In Point 5the text it was used hypothesis term, but this research raised questions, not hypothesis (221, 256). So you need to eliminate the usage of hypothesis or formulate hypothesis in introduction part.

Response 5: The Text has been re-edited, the term hypothesis  omitted.

  1. Point 6. Correlation didn‘t indicate influences (222-223) 

Response 6: It is true, the term "influence" has been replaced by “prediction”. In fact the use of simple linear regression analysis allowed to draw appropriate conclusions.

Submission Date

03 November 2020

Date of this review

01 Dec 2020 19:03:48

Response to Reviewer 4 Comments

However, in order to better understand and disseminate the study, it should be improved according these points:

  1. Point 1. Link the research questions (97-101), the presentation of the results in studies 1 (174-176) and 2 (239-240), discussion and the conclusions in the same logical order and sequence. For example, right now the aim of the study 1 (174-176), presented in results part differ from the reasearch questions (97-101) in the end of the introduction.

Response 1: Thanks to the reviewer1s suggestion, necessary corrections were made at the beginning and at the end of the article. Research questions have been refined and reformulated. Now Research questions are: What is the specificity of attitudes towards parents among middle-aged women included in the daily care of their parents? What is the role of the subjective factor (their attachment style, the feeling of guilt to parents, the style of separation from parents) in their psycho-emotional health and well-being? The logic of the presentation of the results is brought in accordance with the questions asked.

  1. Point 2. To describe the research methodologies in more detail (144-154): to present the number of items, the nature of their evaluation (Likert type or other...), reliability coefficients (Cronbach α or other coefficient).

Response 2: We are grateful to the reviewer for an important comment that helped bring methods description (2. 2) to the required standards. We`ve described the techniques in more detail, indicating number of items, a Likert scale, reliability coefficient (Cronbach alpha),  evaluation examples in Russian samples. 

  1. Point 3. In the discussion section, analyze the results at the level of phenomena by discarding the presented p (299-304 and so on)

Response 3: In the discussion section, p values ​​have been excluded and the results are presented at the level of phenomena, focusing on main ideas.

  1. Point 4. Let the conclusions correspond to the questions raised and answer them in the results section, or let there be more research questions.

Response 4: Once edited, the conclusions are consistent with the research questions (see section 5).

  1. Point 5. In Point 5the text it was used hypothesis term, but this research raised questions, not hypothesis (221, 256). So you need to eliminate the usage of hypothesis or formulate hypothesis in introduction part.

Response 5: The Text has been re-edited, the term hypothesis  omitted.

  1. Point 6. Correlation didn‘t indicate influences (222-223) 

Response 6: It is true, the term "influence" has been replaced by “prediction”. In fact the use of simple linear regression analysis allowed to draw appropriate conclusions.

Submission Date

03 November 2020

Date of this review

01 Dec 2020 19:03:48

Response to Reviewer 4 Comments

However, in order to better understand and disseminate the study, it should be improved according these points:

  1. Point 1. Link the research questions (97-101), the presentation of the results in studies 1 (174-176) and 2 (239-240), discussion and the conclusions in the same logical order and sequence. For example, right now the aim of the study 1 (174-176), presented in results part differ from the reasearch questions (97-101) in the end of the introduction.

Response 1: Thanks to the reviewer1s suggestion, necessary corrections were made at the beginning and at the end of the article. Research questions have been refined and reformulated. Now Research questions are: What is the specificity of attitudes towards parents among middle-aged women included in the daily care of their parents? What is the role of the subjective factor (their attachment style, the feeling of guilt to parents, the style of separation from parents) in their psycho-emotional health and well-being? The logic of the presentation of the results is brought in accordance with the questions asked.

  1. Point 2. To describe the research methodologies in more detail (144-154): to present the number of items, the nature of their evaluation (Likert type or other...), reliability coefficients (Cronbach α or other coefficient).

Response 2: We are grateful to the reviewer for an important comment that helped bring methods description (2. 2) to the required standards. We`ve described the techniques in more detail, indicating number of items, a Likert scale, reliability coefficient (Cronbach alpha),  evaluation examples in Russian samples. 

  1. Point 3. In the discussion section, analyze the results at the level of phenomena by discarding the presented p (299-304 and so on)

Response 3: In the discussion section, p values ​​have been excluded and the results are presented at the level of phenomena, focusing on main ideas.

  1. Point 4. Let the conclusions correspond to the questions raised and answer them in the results section, or let there be more research questions.

Response 4: Once edited, the conclusions are consistent with the research questions (see section 5).

  1. Point 5. In Point 5the text it was used hypothesis term, but this research raised questions, not hypothesis (221, 256). So you need to eliminate the usage of hypothesis or formulate hypothesis in introduction part.

Response 5: The Text has been re-edited, the term hypothesis  omitted.

  1. Point 6. Correlation didn‘t indicate influences (222-223) 

Response 6: It is true, the term "influence" has been replaced by “prediction”. In fact the use of simple linear regression analysis allowed to draw appropriate conclusions.

Submission Date

03 November 2020

Date of this review

01 Dec 2020 19:03:48

Response to Reviewer 4 Comments

However, in order to better understand and disseminate the study, it should be improved according these points:

  1. Point 1. Link the research questions (97-101), the presentation of the results in studies 1 (174-176) and 2 (239-240), discussion and the conclusions in the same logical order and sequence. For example, right now the aim of the study 1 (174-176), presented in results part differ from the reasearch questions (97-101) in the end of the introduction.

Response 1: Thanks to the reviewer1s suggestion, necessary corrections were made at the beginning and at the end of the article. Research questions have been refined and reformulated. Now Research questions are: What is the specificity of attitudes towards parents among middle-aged women included in the daily care of their parents? What is the role of the subjective factor (their attachment style, the feeling of guilt to parents, the style of separation from parents) in their psycho-emotional health and well-being? The logic of the presentation of the results is brought in accordance with the questions asked.

  1. Point 2. To describe the research methodologies in more detail (144-154): to present the number of items, the nature of their evaluation (Likert type or other...), reliability coefficients (Cronbach α or other coefficient).

Response 2: We are grateful to the reviewer for an important comment that helped bring methods description (2. 2) to the required standards. We`ve described the techniques in more detail, indicating number of items, a Likert scale, reliability coefficient (Cronbach alpha),  evaluation examples in Russian samples. 

  1. Point 3. In the discussion section, analyze the results at the level of phenomena by discarding the presented p (299-304 and so on)

Response 3: In the discussion section, p values ​​have been excluded and the results are presented at the level of phenomena, focusing on main ideas.

  1. Point 4. Let the conclusions correspond to the questions raised and answer them in the results section, or let there be more research questions.

Response 4: Once edited, the conclusions are consistent with the research questions (see section 5).

  1. Point 5. In Point 5the text it was used hypothesis term, but this research raised questions, not hypothesis (221, 256). So you need to eliminate the usage of hypothesis or formulate hypothesis in introduction part.

Response 5: The Text has been re-edited, the term hypothesis  omitted.

  1. Point 6. Correlation didn‘t indicate influences (222-223) 

Response 6: It is true, the term "influence" has been replaced by “prediction”. In fact the use of simple linear regression analysis allowed to draw appropriate conclusions.

Submission Date

03 November 2020

Date of this review

01 Dec 2020 19:03:48

Response to Reviewer 4 Comments

However, in order to better understand and disseminate the study, it should be improved according these points:

  1. Point 1. Link the research questions (97-101), the presentation of the results in studies 1 (174-176) and 2 (239-240), discussion and the conclusions in the same logical order and sequence. For example, right now the aim of the study 1 (174-176), presented in results part differ from the reasearch questions (97-101) in the end of the introduction.

Response 1: Thanks to the reviewer1s suggestion, necessary corrections were made at the beginning and at the end of the article. Research questions have been refined and reformulated. Now Research questions are: What is the specificity of attitudes towards parents among middle-aged women included in the daily care of their parents? What is the role of the subjective factor (their attachment style, the feeling of guilt to parents, the style of separation from parents) in their psycho-emotional health and well-being? The logic of the presentation of the results is brought in accordance with the questions asked.

  1. Point 2. To describe the research methodologies in more detail (144-154): to present the number of items, the nature of their evaluation (Likert type or other...), reliability coefficients (Cronbach α or other coefficient).

Response 2: We are grateful to the reviewer for an important comment that helped bring methods description (2. 2) to the required standards. We`ve described the techniques in more detail, indicating number of items, a Likert scale, reliability coefficient (Cronbach alpha),  evaluation examples in Russian samples. 

  1. Point 3. In the discussion section, analyze the results at the level of phenomena by discarding the presented p (299-304 and so on)

Response 3: In the discussion section, p values ​​have been excluded and the results are presented at the level of phenomena, focusing on main ideas.

  1. Point 4. Let the conclusions correspond to the questions raised and answer them in the results section, or let there be more research questions.

Response 4: Once edited, the conclusions are consistent with the research questions (see section 5).

  1. Point 5. In Point 5the text it was used hypothesis term, but this research raised questions, not hypothesis (221, 256). So you need to eliminate the usage of hypothesis or formulate hypothesis in introduction part.

Response 5: The Text has been re-edited, the term hypothesis  omitted.

  1. Point 6. Correlation didn‘t indicate influences (222-223) 

Response 6: It is true, the term "influence" has been replaced by “prediction”. In fact the use of simple linear regression analysis allowed to draw appropriate conclusions.

Submission Date

03 November 2020

Date of this review

01 Dec 2020 19:03:48

Response to Reviewer 4 Comments

However, in order to better understand and disseminate the study, it should be improved according these points:

  1. Point 1. Link the research questions (97-101), the presentation of the results in studies 1 (174-176) and 2 (239-240), discussion and the conclusions in the same logical order and sequence. For example, right now the aim of the study 1 (174-176), presented in results part differ from the reasearch questions (97-101) in the end of the introduction.

Response 1: Thanks to the reviewer1s suggestion, necessary corrections were made at the beginning and at the end of the article. Research questions have been refined and reformulated. Now Research questions are: What is the specificity of attitudes towards parents among middle-aged women included in the daily care of their parents? What is the role of the subjective factor (their attachment style, the feeling of guilt to parents, the style of separation from parents) in their psycho-emotional health and well-being? The logic of the presentation of the results is brought in accordance with the questions asked.

  1. Point 2. To describe the research methodologies in more detail (144-154): to present the number of items, the nature of their evaluation (Likert type or other...), reliability coefficients (Cronbach α or other coefficient).

Response 2: We are grateful to the reviewer for an important comment that helped bring methods description (2. 2) to the required standards. We`ve described the techniques in more detail, indicating number of items, a Likert scale, reliability coefficient (Cronbach alpha),  evaluation examples in Russian samples. 

  1. Point 3. In the discussion section, analyze the results at the level of phenomena by discarding the presented p (299-304 and so on)

Response 3: In the discussion section, p values ​​have been excluded and the results are presented at the level of phenomena, focusing on main ideas.

  1. Point 4. Let the conclusions correspond to the questions raised and answer them in the results section, or let there be more research questions.

Response 4: Once edited, the conclusions are consistent with the research questions (see section 5).

  1. Point 5. In Point 5the text it was used hypothesis term, but this research raised questions, not hypothesis (221, 256). So you need to eliminate the usage of hypothesis or formulate hypothesis in introduction part.

Response 5: The Text has been re-edited, the term hypothesis  omitted.

  1. Point 6. Correlation didn‘t indicate influences (222-223) 

Response 6: It is true, the term "influence" has been replaced by “prediction”. In fact the use of simple linear regression analysis allowed to draw appropriate conclusions.

Submission Date

03 November 2020

Date of this review

01 Dec 2020 19:03:48

Response to Reviewer 4 Comments

However, in order to better understand and disseminate the study, it should be improved according these points:

  1. Point 1. Link the research questions (97-101), the presentation of the results in studies 1 (174-176) and 2 (239-240), discussion and the conclusions in the same logical order and sequence. For example, right now the aim of the study 1 (174-176), presented in results part differ from the reasearch questions (97-101) in the end of the introduction.

Response 1: Thanks to the reviewer1s suggestion, necessary corrections were made at the beginning and at the end of the article. Research questions have been refined and reformulated. Now Research questions are: What is the specificity of attitudes towards parents among middle-aged women included in the daily care of their parents? What is the role of the subjective factor (their attachment style, the feeling of guilt to parents, the style of separation from parents) in their psycho-emotional health and well-being? The logic of the presentation of the results is brought in accordance with the questions asked.

  1. Point 2. To describe the research methodologies in more detail (144-154): to present the number of items, the nature of their evaluation (Likert type or other...), reliability coefficients (Cronbach α or other coefficient).

Response 2: We are grateful to the reviewer for an important comment that helped bring methods description (2. 2) to the required standards. We`ve described the techniques in more detail, indicating number of items, a Likert scale, reliability coefficient (Cronbach alpha),  evaluation examples in Russian samples. 

  1. Point 3. In the discussion section, analyze the results at the level of phenomena by discarding the presented p (299-304 and so on)

Response 3: In the discussion section, p values ​​have been excluded and the results are presented at the level of phenomena, focusing on main ideas.

  1. Point 4. Let the conclusions correspond to the questions raised and answer them in the results section, or let there be more research questions.

Response 4: Once edited, the conclusions are consistent with the research questions (see section 5).

  1. Point 5. In Point 5the text it was used hypothesis term, but this research raised questions, not hypothesis (221, 256). So you need to eliminate the usage of hypothesis or formulate hypothesis in introduction part.

Response 5: The Text has been re-edited, the term hypothesis  omitted.

  1. Point 6. Correlation didn‘t indicate influences (222-223) 

Response 6: It is true, the term "influence" has been replaced by “prediction”. In fact the use of simple linear regression analysis allowed to draw appropriate conclusions.

Submission Date

03 November 2020

Date of this review

01 Dec 2020 19:03:48

Round 2

Reviewer 1 Report

This report about the burden of women in Russia looking after elderly parents has been revised and I appreciate the revisions.

However, for the sake of readers, this paper has to be made clearer. It needs to be considerably more concise and attention is still needed to English.

I would recommend

Cutting out everything that is not absolutely required.

Making it absolutely clear from the beginning that there are two studies and formatting the paper in a way that makes this clear.

Discuss the overload on this sample of women in two parts a) the overload itself - how it potentially applies to all women in these circumstances

b) anything that is special and perhaps unique to women in Russia.

This is a good paper but it will not be read unless it is shortened and written in a clearer fashion.

Reviewer 3 Report

I accept the authors' responses to my comments related to the theoretical introduction which has been improved highly. However, their explanations concerning methodological and statistical issues couldn't be approved. What is more, some of the explanations disclosed even more serious faults. For example, they apply the same instruments to measure independent variables (in FACTOR ANALYSIS) and dependent variables (in ONE-WAY ANOVA). Therefore, the results of the study are inconclusive.